# DISTRIBUTED ALGORITHMS FOR EUCLIDEAN CLUSTERING

**Vincent Cohen-Addad**
Google Research
cohenaddad@google.com

**Liudeng Wang**
Texas A&M University
eureka@tamu.edu

**David P. Woodruff**
Carnegie Mellon University
dwoodruf@andrew.cmu.edu

**Samson Zhou**
Texas A&M University
samsonzhou@gmail.com

## ABSTRACT

We study the problem of constructing $(1 + \varepsilon)$-coresets for Euclidean $(k, z)$-clustering in the distributed setting, where $n$ data points are partitioned across $s$ sites. We focus on two prominent communication models: the coordinator model and the blackboard model. In the coordinator model, we design a protocol that achieves a $(1 + \varepsilon)$-strong coreset with total communication complexity $\tilde{O}\left(sk + \frac{dk}{\min(\varepsilon^4, \varepsilon^{2+z})} + dk \log(n\Delta)\right)$ bits, improving upon prior work (Chen et al., NeurIPS 2016) by eliminating the need to communicate explicit point coordinates in-the-clear across all servers. In the blackboard model, we further reduce the communication complexity to $\tilde{O}\left(s \log(n\Delta) + dk \log(n\Delta) + \frac{dk}{\min(\varepsilon^4, \varepsilon^{2+z})}\right)$ bits, achieving better bounds than previous approaches while upgrading from constant-factor to $(1 + \varepsilon)$-approximation guarantees. Our techniques combine new strategies for constant-factor approximation with efficient coreset constructions and compact encoding schemes, leading to optimal protocols that match both the communication costs of the best-known offline coreset constructions and existing lower bounds (Chen et al., NeurIPS 2016, Huang et. al., STOC 2024), up to polylogarithmic factors.

## 1 INTRODUCTION

Clustering is the process of partitioning a dataset by grouping points with similar properties and separating those with differing properties. The study of clustering dates back to the 1950s (Steinhaus et al., 1956; MacQueen, 1967), and its many variants find applications in fields such as bioinformatics, combinatorial optimization, computational geometry, computer graphics, data science, and machine learning. In the Euclidean $(k, z)$-clustering problem, the input consists of a set $X$ of $n$ points $x_1, \ldots, x_n \in \mathbb{R}^d$, along with a cluster count $k > 0$ and an exponent $z > 0$. The objective is to find a set $\mathcal{C}$ of at most $k$ centers that minimizes the clustering cost:

$$\min_{\mathcal{C} \subset \mathbb{R}^d, |\mathcal{C}| \le k} \mathrm{cost}(X, \mathcal{C}) := \min_{\mathcal{C} \subset \mathbb{R}^d, |\mathcal{C}| \le k} \sum_{i=1}^{n} \min_{c \in \mathcal{C}} \|x_i - c\|_2^z.$$

When $z = 1$ and $z = 2$, the $(k, z)$-clustering problem reduces to the classical $k$-median and $k$-means clustering problems, respectively. These models remain the most widely utilized formulations in practice, with significant research dedicated to their algorithmic foundations (Charikar et al., 1997; 1999; Charikar & Guha, 1999; Meyerson, 2001; Charikar et al., 2003; Feldman & Langberg, 2011; Cohen-Addad et al., 2020; Braverman et al., 2021; Cohen-Addad et al., 2021; Braverman et al., 2022; Cohen-Addad et al., 2022a; Ergun et al., 2022; Tukan et al., 2022; Bucarelli et al., 2023; Cohen-Addad et al., 2023a; Woodruff et al., 2023; Braverman et al., 2025; Cohen-Addad et al., 2025a;b; Karthik C. S. et al., 2025).

Due to the substantial growth in modern datasets, focus has shifted toward large-scale computational models that can process data across multiple machines without requiring centralized access to the

full dataset. The distributed model of computation has become a popular framework for handling such large-scale data. In the distributed setting, the points $x_1, \ldots, x_n$ of $X$ are partitioned across $s$ different machines, and given an input accuracy parameter $\varepsilon > 0$, the goal is for the machines to collectively find a clustering $\mathcal{C}$ of $X$ with cost that is a $(1 + \varepsilon)$-multiplicative approximation of the optimal clustering of $X$, while minimizing the total communication between machines. As in other models, a set $\mathcal{C}$ of $k$ centers implicitly defines the clustering, since each point is assigned to its nearest center. Since transmitting the entire dataset or an explicit cluster label for each point would incur communication linear in $n = |X|$, it is more feasible to design protocols that exchange only succinct summaries, such as small sets of representative points or coresets. We also note that because of finite precision, input points are assumed to lie within the grid $\{1, \ldots, \Delta\}^d$, which can be communicated efficiently using a small number of bits per coordinate.

A standard strategy for efficient distributed clustering is to have each machine construct a small weighted subset of its local points, i.e., a coreset, that preserves the clustering cost for any choice of $k$ centers. These local coresets can then be merged at a coordinator or hierarchically aggregated to approximate the clustering objective over the full dataset. Naturally, smaller coresets correspond to lower communication costs and faster centralized processing. In the offline setting, where the full dataset $X$ is available on a single machine without resource constraints, coreset constructions are known that select $\tilde{\mathcal{O}}\left(\min\left(\frac{1}{\varepsilon^2} \cdot k^{2-\frac{z}{z+2}}, \frac{1}{\min(\varepsilon^4, \varepsilon^{2+z})} \cdot k\right)\right)$ weighted points of $X$ (Cohen-Addad et al., 2021; 2022a;b; Huang et al., 2024)[1][2]. As is common, we assume $\Delta = \text{poly}(n)$ so that each coordinate can be stored in $\mathcal{O}(\log(nd\Delta))$ bits, allowing efficient communication and storage. Thus, offline coreset constructions have size independent of $n$, an essential feature given that modern datasets often contain hundreds of millions of points.

## 1.1 OUR CONTRIBUTIONS

We study the construction of $(1+\varepsilon)$-coresets for Euclidean $(k, z)$-clustering in the distributed setting, focusing on the coordinator and blackboard communication models.

**The coordinator model.** In the coordinator (message passing) model, the $s$ sites, i.e., servers, communicate only with the coordinator over private channels, using private randomness. Protocols are assumed without loss of generality to be sequential, round-based, and self-delimiting, i.e., in each round the coordinator speaks to some number of players and awaits their responses before initiating the next round and all parties know when each message has been completely sent. Given $\varepsilon > 0$, the goal is a protocol $\Pi$ where the coordinator outputs a $(1 + \varepsilon)$-coreset for Euclidean $(k, z)$-clustering that minimizes the communication cost, i.e., the total number of bits exchanged in the worst case. We achieve optimal communication protocols for clustering in this model, matching known lower bounds from Chen et al. (2016); Huang et al. (2024):

**Theorem 1.1** (Communication-optimal clustering in the coordinator model, informal). *Given accuracy parameter $\varepsilon \in (0, 1)$, there exists a protocol on $n$ points distributed across $s$ sites that produces a $(1 + \varepsilon)$-strong coreset for $(k, z)$-clustering that uses $\tilde{\mathcal{O}}\left(sk + \frac{dk}{\min(\varepsilon^4, \varepsilon^{2+z})} + dk\log(n\Delta)\right)$ total bits of communication in the coordinator model.*

We note that our results also yield the optimal bounds for $k$-median, matching Huang et al. (2024); Bansal et al. (2024); for clarity, we omit this special case in the remainder of the paper. By comparison, the protocol of Balcan et al. (2013), combined with more recent coreset constructions such as Cohen-Addad et al. (2021; 2022a); Bansal et al. (2024), achieves a $(1 + \varepsilon)$-strong coreset using $\mathcal{O}\left(\frac{skd}{\min(\varepsilon^4, \varepsilon^{2+z})}\log(n\Delta)\right)$ bits of communication. At first glance, one might expect our bounds to follow by straightforward generalizations of these approaches; however, as we explain in Appendix A, this is not the case. For example, our result shows that no points need to be communicated "in the clear" across sites—there is no $\mathcal{O}(sd\log(n\Delta))$ dependence in the communication. This is surprising, since one might expect the coordinates of a constant-factor approximation to be broadcast to all users. Moreover, the $\frac{1}{\varepsilon}$ factors in our bounds do not multiply either the number of sites $s$ or the $\log(n\Delta)$ cost of transmitting a single coordinate. Finally, our results extend to arbitrary

---

[1]We write $\tilde{\mathcal{O}}\left(f\left(n, d, k, \Delta, \frac{1}{\varepsilon}\right)\right)$ to denote $\mathcal{O}\left(f\left(n, d, k, \Delta, \frac{1}{\varepsilon}\right)\right) \cdot \text{polylog}\left(f\left(n, d, k, \Delta, \frac{1}{\varepsilon}\right)\right)$.

[2]The bound further refines to $\tilde{\mathcal{O}}\left(\min\left(\frac{k^{4/3}}{\varepsilon^2}, \frac{k}{\varepsilon^3}\right)\right)$ for $k$-median.

connected communication topologies, even when certain pairs of sites are not allowed to interact. The formal statement is deferred to Theorem E.7 in Appendix E.

| Coordinator model | Communication cost (bits) |
| --- | --- |
| Merge-and-reduce, with (Bansal et al., 2024) | $\tilde{\mathcal{O}}\left(\frac{skd}{\min(\varepsilon^4,\varepsilon^{2+z})}\log(n\Delta)\right)$ |
| (Balcan et al., 2013), $z=2$ | $\mathcal{O}\left(\frac{d^2k}{\varepsilon^4}\log(n\Delta) + sdk\log(sk)\log(n\Delta)\right)$ |
| (Balcan et al., 2013) in conjunction with (Bansal et al., 2024) | $\mathcal{O}\left(\frac{dk}{\min(\varepsilon^4,\varepsilon^{2+z})}\log(n\Delta) + sdk\log(sk)\log(n\Delta)\right)$ |
| Theorem 1.1 (this work) | $\tilde{\mathcal{O}}\left(sk + \frac{dk}{\min(\varepsilon^4,\varepsilon^{2+z})} + dk\log(n\Delta)\right)$ |

| Blackboard model | Communication cost (bits) |
| --- | --- |
| (Chen et al., 2016) | $\tilde{\mathcal{O}}\left((s+dk)\log^2(n\Delta)\right)$ |
| Theorem 1.2 (this work) | $\tilde{\mathcal{O}}\left(s\log(n\Delta) + dk\log(n\Delta) + \frac{dk}{\min(\varepsilon^4,\varepsilon^{2+z})}\right)$ |

Fig. 1: Table of $(k,z)$-clustering algorithms in the distributed setting. We remark that (Chen et al., 2016) only achieves a constant-factor-approximation, whereas we achieve a $(1+\varepsilon)$-approximation.

**The blackboard model.** Next, we consider the blackboard model of communication, where each of the $s$ sites can broadcast messages that are visible to all of the other sites. Formally in the blackboard model, communication occurs through a shared public blackboard that is visible to all servers. Each server again has access to private sources of randomness. Unlike the coordinator model, there is no designated coordinator to relay messages; instead, any server may write messages directly onto the blackboard. All servers can immediately observe the entire contents of the blackboard at any point in time. As before, we assume without loss of generality that the protocol is sequential and round-based, meaning that in each round, one or more servers write to the blackboard, and all servers can then read the newly posted messages before the next round begins. Accordingly, messages must be self-delimiting so that all servers can correctly parse when a message has been fully posted.

Given an accuracy parameter $\varepsilon$, the objective is to perform a protocol $\Pi$ such that a $(1+\varepsilon)$-coreset for Euclidean $(k,z)$-clustering is explicitly written onto the blackboard. The communication cost of $\Pi$ is defined as the total number of bits written to the blackboard over the course of the protocol, measured in the worst case. We note that the blackboard model can sometimes yield lower communication costs compared to the coordinator model, as messages need only be written once to be accessible to all servers simultaneously. Indeed we achieve a distributed protocol for the blackboard model with substantially less communication than our protocol for the coordinator model, tight with existing lower bounds (Chen et al., 2016; Huang et al., 2024).

**Theorem 1.2** (Communication-optimal clustering in the blackboard model, informal). *There exists a protocol on $n$ points distributed across $s$ sites that produces a $(1+\varepsilon)$-strong coreset for $(k,z)$-clustering that uses $\tilde{\mathcal{O}}\left(s\log(n\Delta) + dk\log(n\Delta) + \frac{dk}{\min(\varepsilon^4,\varepsilon^{2+z})}\right)$ total bits of communication in the blackboard model.*

By comparison, the state-of-the-art protocol achieves a constant factor approximation using $\tilde{\mathcal{O}}\left((s+dk)\log^2(n\Delta)\right)$ total communication (Chen et al., 2016). Thus compared to the work of Chen et al. (2016), not only do we achieve a $(1+\varepsilon)$-coreset construction in the blackboard setting, but also we remove extraneous $\log(n\Delta)$ factors. Conceptually, the message of Theorem 1.2 is similar to that of Theorem 1.1: some amount of exact coordinates need to be communicated (perhaps only to a small number of sites in the coordinator setting) for a constant-factor communication, but no further overhead is necessary for improving to a $(1+\varepsilon)$-approximation. Finally, we remark that for both the blackboard model and the coordinator model, our coresets have size $\tilde{\mathcal{O}}\left(\frac{k}{\varepsilon^4}\right)$ for the important cases of $k$-median and $k$-median due to sensitivity sampling (Bansal et al., 2024) and in general not only match known lower bounds (Huang et al., 2024) but also can be further optimized in certain regimes of $n$ and $k$ (Cohen-Addad et al., 2021; 2022a).

**Technical and algorithmic novelties.** We remark that our work introduces several algorithmic and technical innovations that may be of independent interest. For instance, we describe a number of existing approaches and why they do not work in Appendix A.

Traditional sensitivity sampling selects entire data points with probability proportional to their importance. While effective in centralized settings, this approach does not translate efficiently to dis-

tributed environments: transmitting full points or high-dimensional centers incurs prohibitive communication costs, and naive adaptive sampling requires frequent updates from all sites.

To address these challenges, we introduce several novel techniques. First, in the blackboard model, we show that adaptive sampling is robust to outdated information, leading to a "lazy" adaptive sampling protocol where sites only update the blackboard when their local weight estimates change significantly. Sampling from this outdated distribution still guarantees a constant-factor approximation, reducing both the number of transmitting sites per round and the total number of rounds. An additional $L_1$ sampling subroutine further detects significant changes in global weight without querying all sites, improving communication efficiency.

In the coordinator model, we introduce a communication-efficient subroutine based on coordinate-wise sampling. Rather than sending full high-dimensional centers, the coordinator and a site perform a distributed binary search on the site's sorted coordinates to find the closest match. Only a small offset is transmitted, decoupling communication from the dimension $d$.

Overall, we combine these techniques with coordinate-wise sensitivity sampling: each point is decomposed along its coordinates, and dimensions are sampled based on their significance. This allows the coordinator to send compact summaries to each site, with servers requesting additional information only when necessary. However, the reconstructed samples may not correspond to any actual point in the dataset, requiring careful analysis to show that the overall clustering costs are not significantly distorted. We believe these techniques could also benefit other distributed settings, such as regression and low-rank approximation. This fits into a general body of work on methods for quantizing data for better memory and communication efficiency, which is often a bottleneck for large language and other models.

## 2 DISTRIBUTED CLUSTERING PROTOCOLS IN THE BLACKBOARD MODEL

Recall that in the blackboard model of communication, each of the $s$ sites has access to private randomness and can directly broadcast to a public platform in sequential, round-based steps, with self-delimiting messages immediately visible to all. Throughout this section, we assume without loss of generality that there is a central coordinator managing the process. Our goal is to design efficient protocols for $(k, z)$-clustering in this setting.

### 2.1 CONSTANT-FACTOR BICRITERIA ALGORITHM

In this section, we present a new algorithm that achieves an $(\mathcal{O}(1), \mathcal{O}(1))$-bicriteria approximation; we will use it to construct a $(1 + \varepsilon)$-coreset in Section 2.2. The resulting scheme yields a $(1 + \varepsilon)$-approximation with $\tilde{\mathcal{O}}\left(s \log n + dk \log n + \frac{dk}{\min(\varepsilon^4, \varepsilon^{2+z})}\right)$ bits of communication and $\mathcal{O}(\log n \log k)$ rounds, with additional optimizations for the case $k = \mathcal{O}(\log n)$ given in the appendix. Existing bicriteria algorithms in the blackboard model suffer from communication bottlenecks, e.g., the classical Mettu–Plaxton protocol (Mettu & Plaxton, 2004) and the subsequent adaptations (Chen et al., 2016) sample $\mathcal{O}(k)$ points in each of $\mathcal{O}(\log n)$ rounds, incurring $\mathcal{O}(dk \log^2 n)$ bits of communication. These costs are prohibitive for our target guarantees.

To overcome these barriers, we adapt the adaptive sampling framework originally developed for $k$-median in the centralized setting (Aggarwal et al., 2009; Balcan et al., 2013). The procedure repeatedly samples points in a manner reminiscent of kmeans++ (Arthur & Vassilvitskii, 2007; Bahmani et al., 2012), but now distributed across $s$ servers. In each iteration, we first sample a server $j$ with probability proportional to $D_j$, the sum of the $z$-th power of the distances from its points to the current sample, and then sample points within server $j$ according to the adaptive distribution. A naïve implementation requires reporting each $D_j$ after every iteration, leading to $\mathcal{O}(sk \log n)$ communication, which is too expensive. To implement this step efficiently, we design the subroutine LAZYSAMPLING, which draws from the adaptive distribution using only approximate values $\widetilde{D_j}$ maintained on the blackboard. Our key innovation is to update these estimates lazily: each site reports a new value only when its true $D_j$ changes by more than a constant factor, which we show suffices for the purposes of adaptive sampling.

**Lemma 2.1.** *There exists an algorithm* LAZYSAMPLING *that samples from the adaptive sampling distribution with probability at least* $0.99$ *provided that* $\sum D_j \leq \sum \widetilde{D}_j < \lambda \sum D_j$ *for a fixed constant* $\lambda > 1$. *The algorithm uses* $\tilde{\mathcal{O}} \left( \log s + d \log n \right)$ *bits of communication.*

The LAZYSAMPLING algorithm is a communication-efficient subroutine for adaptive sampling in the blackboard model, formally described in Algorithm 4 in Appendix B. Each server $j$ maintains an approximate weight $\widetilde{D}_j$ for its local dataset, satisfying $D_j \leq \widetilde{D}_j \leq \lambda D_j$, where $D_j$ is the sum of the $z$-th powers of distances from its points to the current sample. The coordinator first selects a server $j$ with probability proportional to $\widetilde{D}_j / \sum_i \widetilde{D}_i$, then requests a point $y \in X_j$, sampled with probability $d_y / \widetilde{D}_j$. By updating the $\widetilde{D}_j$ lazily, i.e., only when $D_j$ changes significantly, LAZYSAMPLING ensures points are drawn close to the true adaptive distribution while drastically reducing communication. The algorithm either returns a sampled point or $\perp$, and uses only $\tilde{\mathcal{O}} \left( \log s + d \log n \right)$ bits per round, enabling scalable execution of the bicriteria algorithm across $s$ servers. This ensures that each site communicates only $\mathcal{O} \left( \log n \right)$ updates, so the total communication for the constant-factor approximation is reduced to $\tilde{\mathcal{O}} \left( s \log n + kd \log n \right)$, since each server can only update $\widetilde{D}_j$ a total of $\mathcal{O} \left( \log n \right)$ times assuming all points lie in a grid with side length $\text{poly}(n)$. Because lazy updates rely on approximate values, the protocol must occasionally verify whether the aggregate estimate $\sum_j \widetilde{D}_j$ is still close to the true sum $\sum_j D_j$. For this purpose, we use the subroutine L1SAMPLING, which tests whether the two sums differ by more than a constant factor.

**Lemma 2.2.** *There exists an algorithm* L1SAMPLING *that takes input* $\{\widetilde{D}_j\}_{j \in [s]}$ *so that* $\widetilde{D}_j \geq D_j$ *for all* $j \in [s]$. *Let* $\widetilde{D} = \sum_{j \in [s]} \widetilde{D}_j$ *and* $D = \sum_{j \in [s]} D_j$ *and let* $\mu > 1$ *be a parameter. If* $\mu^2 D \leq \widetilde{D}$, *then* L1SAMPLING *returns* True *with probability at least* $1 - \delta$. *If* $\mu D > \widetilde{D}$, *it returns* False *with probability at least* $1 - \delta$. *The algorithm uses* $\tilde{\mathcal{O}} \left( \left( \log s + \log \log n \right) \log \frac{1}{\delta} \right)$ *bits of communication.*

L1SAMPLING checks whether the sum of approximate site costs $\widetilde{D}$ is within a constant factor of the true total $D$ by sampling a few sites and aggregating the rescaled sampled values. If the aggregate is sufficiently close to $\widetilde{D}$, the algorithm returns True; otherwise, it returns False, providing a high-probability guarantee that the lazy estimates remain accurate; the algorithm is presented as Algorithm 9 in Appendix D.2. While this lazy strategy could require up to $\mathcal{O} \left( s \log n \right)$ rounds, we further reduce the round complexity by delaying all updates until the global sum $\sum_j D_j$ decreases by a constant factor. This event is naturally detected when no new point is sampled in a round, and synchronizing updates in this way brings the total number of rounds down to $\mathcal{O} \left( \log n \log k \right)$. Finally, to reduce communication further, we use the subroutine POWERAPPROX, which encodes each $D_j$ to within a constant factor using only $\mathcal{O} \left( \log \log n \right)$ bits:

**Theorem 2.3.** *Given* $m = \text{poly}(n)$ *and a constant* $\lambda > 1$, *there exists an algorithm* POWERAPPROX$(m, \lambda)$ *that outputs* $\widetilde{m}$ *encoded in* $\mathcal{O} \left( \log \log n \right)$ *bits, such that* $m \leq \widetilde{m} < \lambda m$.

Given $m$ and a base $\lambda > 1$, POWERAPPROX computes the smallest integer $i$ such that $\lambda^i$ approximates $m$ from above, i.e., $m \leq \lambda^i < \lambda m$, and returns $i$. This allows each site to communicate a concise representation of its cost using only $\mathcal{O} \left( \log \log n \right)$ bits, which can then be decoded to obtain a constant-factor approximation of the original value; we give the full details in Algorithm 5 in Appendix B. Combined with LAZYSAMPLING and L1SAMPLING, this ensures accuracy while keeping communication near-linear. The protocol proceeds by sampling points with LAZYSAMPLING, verifying accuracy with L1SAMPLING, and refreshing estimates via POWERAPPROX only when necessary. To limit round complexity, sites synchronize updates by reporting only when the global sum $\sum_j D_j$ decreases by a constant factor, detected whenever a round fails to sample a new point. This reduces the number of rounds from $\mathcal{O} \left( s \log n \right)$ to $\mathcal{O} \left( \log n \log k \right)$. Figure 2 informally summarizes the procedure, with the formal procedure appearing in Appendix D.3. Altogether, this yields the first $\left( \mathcal{O} \left( 1 \right), \mathcal{O} \left( 1 \right) \right)$-bicriteria approximation for $(k, z)$-clustering in the blackboard model with near-linear communication and polylogarithmic rounds:

**Lemma 2.4.** *There exists an algorithm that outputs a set* $S$ *such that* $|S| = \mathcal{O} \left( k \right)$ *and* $\text{cost}(S, X) \leq \mathcal{O} \left( 1 \right) \cdot \text{cost}(C_{OPT}, X)$ *with probability at least* $0.98$, *where* $C_{OPT}$ *is the optimal* $(k, z)$-*clustering of* $X$. *The algorithm uses* $\tilde{\mathcal{O}} \left( s \log n + kd \log n \right)$ *bits of communication and* $\mathcal{O} \left( \log n \log k \right)$ *rounds of communication with probability at least* $0.99$.

---

**Algorithm: Bicriteria Approximation for $(k, z)$-Clustering (Simplified)**

1. **Input:** Dataset $X_i$ for each site $i \in [s]$.

2. **Output:** Set $S$ that is an $(\mathcal{O}(1), \mathcal{O}(1))$-bicriteria approximation.

3. Initialize:

   - Sample one point into $S$ and compute approximate distances $\{\widetilde{D_j}\}$ for each site.
   - Set counters: $N = \mathcal{O}(k)$, $M = 0$.

4. **While $M < N$ (sample roughly $k$ points):**

   - Sample new points into $S$ using LAZYSAMPLING.
   - Check accuracy of approximate distances $\{\widetilde{D_j}\}$ with L1SAMPLING.
   - If distances are accurate, sample more aggressively.
   - Otherwise, refine approximate distances $\{\widetilde{D_j}\}$ using POWERAPPROX.
   - Update $M$ with the number of points successfully added.

5. **Return** $S$.

---

Fig. 2: Informal version of bicriteria approximation through adaptive sampling.

## 2.2 $(1 + \varepsilon)$-CORESET CONSTRUCTION

To achieve a $(1 + \varepsilon)$-coreset for $(k, z)$-clustering on an input dataset $X$ given an $(\mathcal{O}(1), \mathcal{O}(1))$-bicriteria approximation $S$, we use the following notion of sensitivity sampling. For each center $s_j \in S$, let $C_j \subset X$ be the cluster centered at $s_j$. For a point $x \in C_j$, let $\Delta_p := \text{cost}(C_j, S)/|C_j|$ denote the average cost of $C_j$. For $x \in C_j$, we define

$$\mu(x) := \frac{1}{4} \cdot \left( \frac{1}{k|C_j|} + \frac{\text{cost}(x, S)}{k \, \text{cost}(C_j, S)} + \frac{\text{cost}(x, S)}{\text{cost}(X, S)} + \frac{\Delta_x}{\text{cost}(X, S)} \right).$$

We define sensitivity sampling to be the process where each point $x$ is sampled with probability proportional to a constant-factor approximation to $\mu(x)$. Then we have the following guarantees:

**Theorem 2.5.** *(Bansal et al., 2024) Sampling $\tilde{\mathcal{O}}\left( \frac{k}{\min(\varepsilon^4, \varepsilon^{2+z})} \right)$ points from constant-factor approximations to the sensitivity sampling probability distribution and then reweighting provides a $(1 + \varepsilon)$-coreset for Euclidean $(k, z)$-clustering with probability at least $0.99$.*

We remark that Bansal et al. (2024) obtained optimal bounds for $k$-median, which immediately extend to our framework as well; we omit further discussion as it naturally generalizes. To apply the sensitivity sampling framework, we require constant-factor approximations of the cluster sizes $|C_j|$ and costs $\text{cost}(C_j, S)$, given a bicriteria solution $S$. However, directly uploading these quantities from $s$ servers for $\mathcal{O}(k)$ clusters would cost $\mathcal{O}(sk \log \log n)$ bits, which is too high. Instead, we adapt Morris counters (Morris, 1978) for the purposes of distributed approximate counting:

**Lemma 2.6.** *Suppose each server $i$ holds $k$ numbers $n_{i,j}$ and we have $|N_j| = \sum_{i=1}^{s} n_{i,j} = \text{poly}(n)$. There exists an algorithm DISTMORRIS that outputs $\{\widehat{N_j}\}$ such that $\widetilde{N}_j \in [\frac{3}{4} N_j, \frac{5}{4} N_j]$ for all $j \in [k]$, with probability $0.99$ using $\tilde{\mathcal{O}}(s + k \log n)$ bits of communication.*

Our Morris counter protocol collectively maintains a counter $r_j$ and increments it with probability $\frac{1}{2^r}$ for each item in a cluster $C_j$. This provides a constant approximation to the cluster size $|C_j|$. Crucially, if a site does not change the global counters $\{r_j\}$, then it only needs to send a single bit to signal no update. Otherwise, each of the $\mathcal{O}(k)$ counters can only increase at most $\mathcal{O}(\log n)$ times, so the total upload cost for these approximations is $\mathcal{O}(k \log n)$ bits. We can perform a similar protocol to approximate the cost of each cluster $\text{cost}(C_j, S)$, so that overall, the total communication for these approximations is $\tilde{\mathcal{O}}(s + k \log n)$ bits. Given these approximations, the servers can then perform sensitivity sampling locally. Finally, we require an efficient encoding of each point $x$ sampled by sensitivity sampling. Informally, each point $x$ is encoded as $x' = \pi_S(x) + y'$, where $\pi_S(x)$ is the nearest center in $S$ to $x$, and $y'$ is the offset vector $x - \pi_S(x)$ whose coordinates are rounded to

the nearest power of $(1 + \varepsilon')$, where $\varepsilon' = \text{poly}\left(\varepsilon, \frac{1}{d}, \frac{1}{\log(n\Delta)}\right)$. The formal details are given in Appendix B. Putting everything together, Algorithm 1 achieves the guarantees in Theorem 1.2.

---

**Algorithm 1** $(1 + \varepsilon)$-coreset for the blackboard model

---

**Input:** A bicriteria set of centers $S$ with constant-factor approximation and $|S| = \mathcal{O}(k)$
**Output:** A $(1 + \varepsilon)$-coreset $A$
1: Use DISTMORRIS to get $\mathcal{O}(1)$-approximation for $|C_j|$ and $\text{cost}(C_j, S)$ for all $j \in [k]$ on the blackboard
2: $m \leftarrow \tilde{\mathcal{O}}\left(\frac{k}{\varepsilon^2}\min\{\varepsilon^{-2}, \varepsilon^{-z}\}\right)$                                                 ▷Set coreset size
3: **for** $i \leftarrow 1$ to $s$ \\ Send local approximations to sensitivities **do**
4:     $A_i \leftarrow \emptyset$
5:     Compute $\widetilde{\mu}(x)$ as an $\mathcal{O}(1)$-approximation of $\mu(x)$ locally for all $x \in X_i$
6:     Upload $\widetilde{\mu}(X_i) = \sum_{x \in X_i} \widetilde{\mu}(x)$ to blackboard
7: **end for**
8: Sample site $i$ with probability $\frac{\widetilde{\mu}(X_i)}{\sum_{i=1}^s \widetilde{\mu}(X_i)}$ independently for $m$ times.     ▷Sensitivity sampling
9: Let $m_i$ be the number of times site $i$ is sampled and write $m_i$ on blackboard
10: **for** $i \leftarrow 1$ to $s$ \\ Iterate through sites to produce samples **do**
11:     $A_i \leftarrow \emptyset$
12:     **for** $j \in [m_i]$ \\ Sample $m_i$ points from site $i$ **do**
13:         Sample $x$ with probability $p_x = \frac{\widetilde{\mu}(x)}{\widetilde{\mu}(X_i)}$
14:         **if** $x$ is sampled \\ Efficiently encode each sample **then**
15:             Let $x'$ be $x$ efficiently encoded by $S$ and accuracy $\varepsilon' = \text{poly}(\varepsilon)$
16:             $A \leftarrow A_i \cup \{(x', \frac{1}{m\widehat{\mu}(x)})\}$, where $\widehat{\mu}(x)$ is a $(1 + \frac{\varepsilon}{2})$-approximation of $\widetilde{\mu}(x)$
17:         **end if**
18:     **end for**
19:     Upload $A_i$ to the blackboard
20: **end for**
21: $A \leftarrow \cup_{i=1}^s A_i$
22: **return** $A$

---

## 3    DISTRIBUTED CLUSTERING PROTOCOLS IN THE COORDINATOR MODEL

We now turn to the coordinator (message passing) model, where each server communicates only with the coordinator over private channels. A direct simulation of our blackboard protocol would require $\mathcal{O}(dsk \log n)$ bits, since $\mathcal{O}(k)$ rounds of adaptive sampling would need to be executed across $s$ servers. To avoid this prohibitive cost, we design a protocol that simulates adaptive sampling without sending points explicitly to all sites. We first apply a Johnson–Lindenstrauss transformation to reduce the dimension to $d' = \mathcal{O}(\log(sk))$, preserving pairwise distances up to $(1 \pm \varepsilon)$. We can then perform adaptive sampling in the projected space, so that when a point $s$ is selected, only its projection $\pi(s)$ is communicated. To efficiently approximate such locations, we introduce the subroutine EFFICIENTCOMMUNICATION, which transmits an approximate version $\widetilde{y}$ of any point $y$ using only $d \log k \, \text{polylog}\left(\log n, \frac{1}{\varepsilon}, \frac{1}{\delta}\right)$ bits, rather than the $\mathcal{O}(d \log n)$ bits required for exact communication:

**Lemma 3.1.** *Given a point $y$ and a dataset $X$, there exists an algorithm* EFFICIENTCOMMUNICATION *that uses $d \log k \, \text{polylog}(\log n, \frac{1}{\varepsilon}, \frac{1}{\delta})$ bits of communication and sends $\widetilde{y}$ such that $\|y - \widetilde{y}\|_2 \leq \min_{x \in X} \varepsilon \|x - y\|_2$ with probability at least $1 - \delta$.*

Intuitively, EFFICIENTCOMMUNICATION allows a site to locate an approximate version of the coordinator's point $y$ using very little communication. For each coordinate $i$, the site first identifies the closest local value $x_{i_s}^{(i)}$ to $y^{(i)}$ via a binary search using the HIGHPROBGREATERTHAN protocol. If $y^{(i)}$ does not exactly match, an exponential search determines a small offset $\Delta y^{(i)}$ so that $x_{i_s}^{(i)} + \Delta y^{(i)}$ approximates $y^{(i)}$ within a factor of $(1 + \varepsilon)$. By doing this coordinate-wise, the site can efficiently reconstruct a point $\widetilde{y}$ that is close to $y$, guaranteeing $|y - \widetilde{y}|_2 \leq \varepsilon |x - y|_2$ for any local point $x \in X$, while sending only a small number of bits. This approach combines the intuition of

searching for the "nearest neighbor" along each coordinate with controlled, approximate refinement, as illustrated in Figure 3.

---

**Algorithm: EFFICIENTCOMMUNICATION$(X, y, \varepsilon, \delta)$ (informal)**

1. **Input:**
   - A set of points $X = \{x_1, \ldots, x_l\}$ owned by one site.
   - A point $y$ from the coordinator.
   - Accuracy parameter $\varepsilon \in (0, 1)$ and failure probability $\delta$.

2. **Goal:** Send an approximate location $\widetilde{y}$ to the site such that $\|y - \widetilde{y}\|_2 \leq \varepsilon \|x - y\|_2$ for any $x \in X$ with probability $\geq 1 - \delta$.

3. For each coordinate $i = 1$ to $d$:
   - Sort points in $X$ by their $i$-th coordinate.
   - Use a local binary search via HIGHPROBGREATERTHAN to find the closest point $x_{i_s}^{(i)}$ to $y^{(i)}$.
   - If $y^{(i)}$ equals $x_{i_s}^{(i)}$, set $\Delta y^{(i)} = 0$.
   - Otherwise:
     - Determine the direction $\gamma = \text{sign}(y^{(i)} - x_{i_s}^{(i)})$.
     - Use exponential search with HIGHPROBGREATERTHAN to find a value $\Delta y^{(i)}$ so that $x_{i_s}^{(i)} + \Delta y^{(i)}$ approximates $y^{(i)}$ within factor $(1 + \varepsilon)$.
   - Set $\widetilde{y}^{(i)} = x_{i_s}^{(i)} + \Delta y^{(i)}$.

4. **Return** $\widetilde{y} = (\widetilde{y}^{(1)}, \ldots, \widetilde{y}^{(d)})$.

---

Fig. 3: Informal version of efficient communication in the message-passing algorithm. For full algorithm, see Algorithm 15.

Given a bicriteria approximation $S$ obtained from the above efficient implementation of adaptive sampling, we next perform sensitivity sampling to construct a $(1 + \varepsilon)$-coreset. Unfortunately, each server can now assign a point $x$ to another center $s'$ instead of the closest center $s$ if $\text{cost}(x, s')$ is very close to $\text{cost}(x, s)$. Consequently, the sizes $|C_j|$ and costs $\text{cost}(C_j, S)$ for the purposes of sensitivity sampling may be incorrectly computed by the servers. However, this does not compromise the correctness: the sensitivity analysis of Bansal et al. (2024) only requires that each point be assigned to a center whose clustering cost is within a constant factor of the optimal assignment. Thus, our procedure still achieves a $(1 + \frac{\varepsilon}{4})$-coreset by sensitivity sampling. Once points are sampled, each site encodes the coordinates of its sampled points using the same efficient encoding scheme as in the blackboard model, c.f., Lemma B.13, ensuring that only a compact representation is sent back to the coordinator. This combination of approximate center assignments and efficient encoding preserves both accuracy and communication efficiency. We give the algorithm in full in Figure 4, which achieves the guarantees of Theorem 1.1, deferring full details to Appendix E.

## 4 EMPIRICAL EVALUATIONS

In this section, we present a number of simple experimental results on both synthetic and real-world datasets that complement our theoretical guarantees. We consider $k$-means clustering in the blackboard setting, using the previous algorithm of Chen et al. (2016) based on Mettu-Plaxton, denoted `MP`, as a baseline. We also implement two versions of our distributed protocol, with varying complexity. We first implement our constant-factor approximation algorithm based on adaptive sampling, denoted `AS`. Additionally, we implement our constant-factor approximation algorithm based on our compact encoding after adaptive sampling, denoted `EAS`. All experiments were conducted on a Dell OptiPlex 7010 Tower desktop equipped with an Intel Core i7-3770 3.40 GHz quad-core processor and 16 GB of RAM. We provide all code at https://github.com/samsonzhou/distributed-cluster.

---

**Algorithm:** $(1+\varepsilon)$**-coreset for the coordinator model (informal)**

(1) Each site computes a local $(1+\varepsilon/2)$-coreset $P_i$.

(2) Coordinator broadcasts a JL transform $\pi$ to all sites.

(3) Initialize solution set $S = \{s_0\}$ with a random point.

(4) For $i = 1$ to $i = \mathcal{O}(k)$ iterations (for bicriteria solution):

- Using EFFICIENTCOMMUNICATION, send approximation $\widetilde{s}_{i-1}^{(j)}$ of center $s_i$ to site $j$.
- Sites compute approximate costs $\widetilde{D}_j$ and send to coordinator.
- Coordinator selects next center $s_i$ into $S$ using LAZYSAMPLING.

(5) Sites compute cluster sizes and costs, send constant approximations to coordinator.

(6) Coordinator computes total approximations and broadcasts to sites.

(7) Sites compute approximate sensitivities $\widetilde{\mu}(x)$, send total to coordinator.

(8) Coordinator samples $m = \tilde{\mathcal{O}}\left(\frac{k}{\min(\varepsilon^4, \varepsilon^{2+z})}\right)$ points across sites based on sensitivities; sites sample points and encode with EFFICIENTCOMMUNICATION.

(9) Merge sampled points into final coreset $A'$ and return.

Fig. 4: Informal version of message-passing algorithm. For full algorithm, see Algorithm 16.

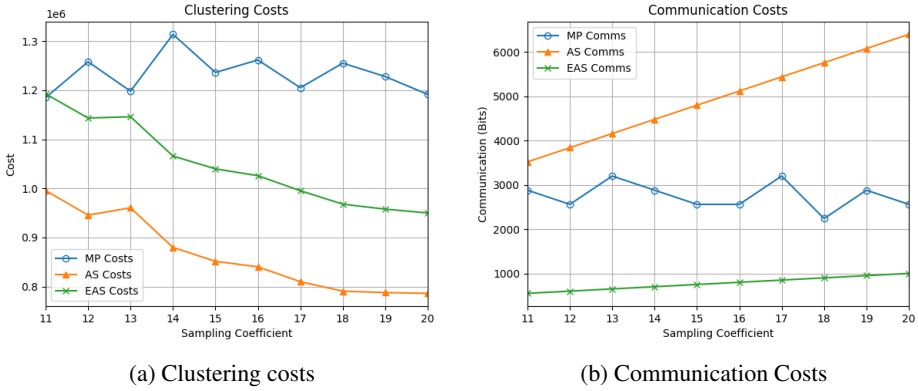

(a) Clustering costs

(b) Communication Costs

Fig. 5: Experiments for clustering costs and communication costs on DIGITS dataset

## 4.1 REAL-WORLD DATASET

To evaluate our algorithms, we conducted our $k$-means clustering algorithms on the DIGITS dataset (Alpaydin & Kaynak, 1998), which consists of 1,797 images of handwritten digits (0-9) and thus naturally associates with $k = 10$. Each image has dimension $8 \times 8$, represented by 64 features, corresponding to the pixel intensities. This dataset is available both through scikit-learn and the UCI repository, and is a popular choice for clustering tasks due to its moderate size and well-defined classes, allowing for a clear evaluation of the clustering performance.

For a parameter $c$, our baseline MP uniformly selects $k$ initial points and then iteratively discards candidates over $c \log_2 n$ rounds. Our AS algorithm iteratively samples a single center in each of $ck$ rounds, with probability proportional to its distance from the previously sampled centers. Finally, EAS quantizes the coordinates of each chosen center of AS to the nearest power of two, mimicking low-precision communication. We compare the clustering costs of the algorithms in Figure 5a and their total communication costs in Figure 5b, both across $c \in \{11, 12, \ldots, 20\}$.

Our results indicate that although adaptive sampling (AS) always outperforms Mettu-Plaxton (MP), when the number of samples is small, the gap is relatively small, so that the rounding error incurred by our efficient encoding in EAS has similar clustering costs for $c = 11$ in Figure 5a. Surprisingly,

the communication cost of MP did not seem to increase with $c$, indicating that all possible points have already been removed and no further samples are possible. Nevertheless, for all $c > 11$, our algorithm in EAS clearly outperforms MP and therefore the previous work of Chen et al. (2016) for both clustering cost and communication cost, c.f., Figure 5b. Our algorithm EAS also exhibits clear tradeoffs in the clustering cost and the communication cost compared to our algorithm AS, as the former is simply a rounding of the latter.

## 4.2 SYNTHETIC DATASET

To facilitate visualization, we generated synthetic datasets consisting of two-dimensional Gaussian mixtures, where the low dimensionality (2D) was chosen to enable visualization of the resulting clusters. Specifically, we created $k = 5$ Gaussian clusters, each containing $n = 100 \times 2^{10}$ points, for a total of $512,000$ data points. Each cluster was sampled from a distinct Gaussian distribution with a randomly selected mean in the range $[-10, 10]^2$ and a randomly generated positive-definite covariance matrix to ensure diverse cluster shapes.

We implemented the baseline MP by sampling $k$ points uniformly and pruning candidates based on a distance threshold that doubles each round, across $c \log_2 n$ rounds, where $c$ is a hyperparameter. We implemented our algorithm AS by iteratively sampling one center per round across $ck$ rounds, selecting points with probability proportional to their distance from existing centers. EAS then modifies AS by rounding each selected center's coordinates to the nearest power of $q = 2^{0.25}$, simulating low-precision communication.

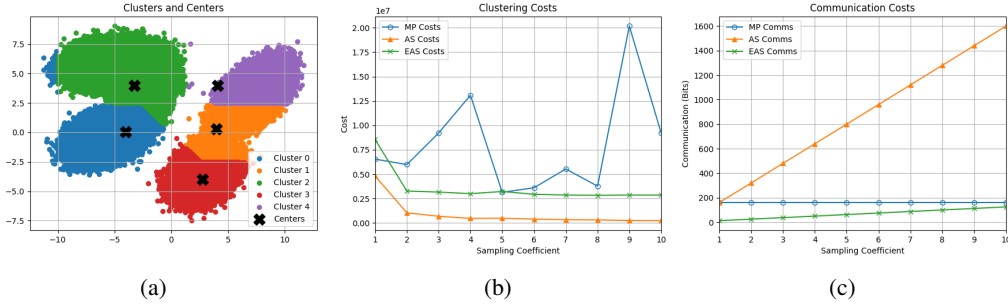

(a)                    (b)                    (c)

Fig. 6: Experiments for clustering costs and communication costs on synthetic dataset

We varied the sampling coefficient $c \in \{11, \ldots, 20\}$ and measured the $k$-means clustering cost (i.e., the total squared distance from each point to its assigned center). The results, shown in Figure 6, highlight the performance trade-offs between sampling strategies as $c$ increases. Communication costs were also computed (though not plotted), assuming 32 bits per coordinate for MP and AS, and 5 bits per coordinate for EAS due to quantization.

Our results for synthetic data echo the trends for the DIGITS dataset. Namely, for all $c > 2$, our algorithm in EAS clearly outperforms MP for both clustering cost and communication cost, c.f., Figure 6c, while also demonstrating clear tradeoffs in the clustering cost and the communication cost compared to our algorithm AS. We plot the resulting clustering by EAS in Figure 6a.

## ACKNOWLEDGMENTS

David P. Woodruff is supported in part Office of Naval Research award number N000142112647, and a Simons Investigator Award. Samson Zhou is supported in part by NSF CCF-2335411. Samson Zhou gratefully acknowledges funding provided by the Oak Ridge Associated Universities (ORAU) Ralph E. Powe Junior Faculty Enhancement Award.

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

## A  TECHNICAL OVERVIEW

In this section, we provide a technical overview for our distributed protocols, for both the blackboard model and the coordinator/message-passing model.

**Previous approaches and why they do not work.** A natural question is whether simple approaches could achieve similar communication bounds. One might hope, for instance, that combining the space upper bounds of Zhu et al. (2025) with existing distributed clustering algorithms would suffice. However, it is not immediately clear how to translate space bounds into better distributed clustering bounds. For instance, as an application of their space bounds, Zhu et al. (2025) achieved a distributed protocol with total communication $\tilde{\mathcal{O}}\left(\frac{sdk}{\varepsilon^4}\right)$, for appropriate ranges of $k$ and $\varepsilon$, which is actually worse than the third row of Figure 1 and thus worse than our bounds of $\tilde{\mathcal{O}}\left(sk + \frac{dk}{\varepsilon^4} + dk \log n\right)$ for the same regime.

Another approach might be that each server computes a local coreset, e.g., Huang & Vishnoi (2020); Cohen-Addad et al. (2021; 2022a;b); Bansal et al. (2024), possibly with the incorporation of dimensionality reduction, e.g., Sohler & Woodruff (2018); Makarychev et al. (2019); Izzo et al. (2021),

and then either broadcasting these coresets in the blackboard model or sending these coresets to the coordinator in the message-passing model. However, this uses communication $\mathcal{O}\left(\frac{sdk}{\varepsilon^4}\log n\right)$ bits, compared to our bounds of $\tilde{\mathcal{O}}\left(sk + \frac{dk}{\varepsilon^4} + dk\log n\right)$. Similarly, any approach based on locality sensitivity hashing, possibly with the aid of dimensionality reduction or other compressions, e.g., Kashin representations (Lyubarskii & Vershynin, 2010; Asi et al., 2024), would require at least each server computing a local coreset and thus succumbing to the same pitfalls. These limitations highlight that overcoming the inherent overheads in naïve strategies is far from straightforward, and motivate the need for new techniques. Our framework is designed precisely to address these challenges by simulating centralized algorithms through communication-optimal primitives, thereby breaking the bottlenecks that defeat such strawman approaches.

$(1+\varepsilon)$-**coreset for the blackboard model.** Our starting point is the distributed protocol of Chen et al. (2016), which first adapts the central RAM algorithm of Mettu & Plaxton (2004) to publish a weighted set $S$ of $\mathcal{O}\left(k\log n\right)$ centers that is a constant-factor bicriteria approximation for $(k,z)$-clustering. It can be shown that a constant-factor approximation to $(k,z)$-clustering on $S$ achieves a constant-factor approximation $C$ to the optimal clustering on the input set $X$. Since $S$ is already available on the blackboard, this step is immediate. The total communication cost of their algorithm is $\mathcal{O}\left((s+kd)\log^2 n\right)$.

We observe that the sites can easily calculate the number of points in $X$ served by each center $c \in C$. We recall that using this information as well as $\mathrm{cost}(X, C)$ can be used to find constant-factor approximations of the sensitivity $s(x)$ of each point $x$, c.f. (Cohen-Addad et al., 2023b). Thus, the sites can subsequently sample $\tilde{\mathcal{O}}\left(\min\left(\frac{k^2}{\varepsilon^2}, \frac{k}{\min(\varepsilon^4, \varepsilon^{2+z})}\right)\right)$ points of $X$ using sensitivity sampling to achieve a $(1+\varepsilon)$-coreset. We can then use an efficient encoding scheme to express each sampled point in $\mathcal{O}\left(\log k + d\log\left(\frac{1}{\varepsilon}, d, \log(n\Delta)\right)\right)$ bits. This achieves the desired dependency for the points acquired through sensitivity sampling, but the communication from the constant-factor approximation is sub-optimal. Thus, it remains to find a more communication-efficient protocol for the constant-factor approximation.

**Constant-factor approximation in the blackboard model.** Although there are many options for the constant-factor approximation in the blackboard model, they all seem to have their various shortcomings. For example, the aforementioned Mettu-Plaxton protocol (Mettu & Plaxton, 2004) requires $\mathcal{O}\left(\log n\right)$ rounds of sampling, which results in $\mathcal{O}\left(k\log n\right)$ points. A natural approach would be to form coresets locally at each site so that the total number of weighted points is $\mathcal{O}\left(sk\right)$ rather than $n$. Unfortunately, when generalized to weighted points, Mettu-Plaxton requires $\mathcal{O}\left(\log W\right)$ rounds, where $W$ is total weight of the points, which is $\mathrm{poly}(n)$ for our purposes.

Another well-known prototype is the adaptive sampling framework (Aggarwal et al., 2009; Balcan et al., 2013), which can be used to implement the well-known kmeans++ algorithm (Arthur & Vassilvitskii, 2007) in the distributed setting (Bahmani et al., 2012). This approach iteratively samples a fixed number of points with probability proportional to the $z$-th power of the distances from the previously sampled points; adaptive sampling samples $\mathcal{O}\left(k\right)$ points for $\mathcal{O}\left(1\right)$-approximation while kmeans++ samples $k$ points for $\mathcal{O}\left(\log k\right)$-approximation. However, since all sites $i \in [s]$ must report the sum $D_i$ of the $z$-th power of the distances of their points to the sampled points after each iteration, these approaches naïvely require $\mathcal{O}\left(sk\log n\right)$ bits of communication, which is prohibitive for our goal.

Instead, we perform lazy updating of the sum $D_i$ of the $z$-th power of the distances of their points to the sampled points. The blackboard only holds estimates $\widehat{D_i}$ for $D_i$, based on the last time the site $i$ reported its value. The sites then attempt adaptive sampling, where the site $i$ is sampled with a probability proportional to $\widehat{D_i}$. When sampled, site $i$ uses $\widehat{D_i}$ to attempt to sample a point locally, but this can fail because $\widehat{D_i} \geq D_i$, which means that a randomly chosen integer in $\widehat{D_i}$ may not correspond to an integer in $D_i$. Fortunately, the failure probability would be constant if $\widehat{D_i}$ is a constant approximation of $D_i$, so the total number of rounds of sampling is still $\mathcal{O}\left(k\right)$ by the Markov inequality. If the site $i$ fails to sample a point, we expect $D_i$ to be a constant fraction smaller than $\widehat{D_i}$, prompting site $i$ to update its value $D_i$ on the blackboard. This can happen at most $\mathcal{O}\left(s\log n\right)$ times. If each site $i$ rounds $D_i$ to a power of 2, then $D_i$ can be approximated within a

factor of 2 by transmitting only the exponent of the rounding, using $\mathcal{O}\left(\log \log n\right)$ bits. Thus, the total communication for the constant-factor approximation is $\tilde{\mathcal{O}}\left(s \log n + kd \log n\right)$.

The primary downside is that the total number of communication rounds could reach $\mathcal{O}\left(s \log n\right)$, since each of the $s$ sites may update its $D_i$ value up to $\mathcal{O}\left(\log n\right)$ times.

**Communication round reduction in the blackboard model.** A natural approach to round reduction would be to have all $s$ sites report their updated $D_i$ values when we fail to sample a point. However, because we sample $\mathcal{O}\left(k\right)$ points, this results in communication complexity $\mathcal{O}\left(sk\right)$.

Instead, we check whether the global weight $\sum_{i \in [s]} D_i$ has decreased by $\frac{1}{64}$ compared to the amount $\sum_{i \in [s]} \widehat{D_i}$ on the blackboard. To do this, we first perform the $L_1$ sampling in each round. That is, the blackboard picks the site $i$ with probability $p_i = \frac{\widehat{D_i}}{\sum_{i \in [s]} \widehat{D_i}}$ and computes $\frac{D_i}{p_i}$. Observe that in expectation this quantity is $\sum_{i \in [s]} D_i$ and its variance is at most $(\sum_{i \in [s]} D_i)^2$, up to a constant factor. Thus, repeating $\mathcal{O}\left(1\right)$ times and taking the average, we get a $\sum_{i \in [s]} D_i$ additive approximation to $\sum_{i \in [s]} D_i$, which is enough to identify whether the sum has decreased by a factor of $\frac{1}{64}$ from $\sum_{i \in [s]} \widehat{D_i}$. We can then take $\mathcal{O}\left(\log \log n\right)$ instances to union bound over $\mathcal{O}\left(\log n \log k\right)$ iterations.

Now the other observation is that as long as our sampling probabilities have not decreased by $\frac{1}{64}$, then we can simultaneously take many samples at the same time. Thus, we start at $i = 0$ and collect $2^i$ samples from the $s$ sites using the adaptive sampling distribution. We then check if the total weight $\sum_{i \in [s]} D_i$ is less than $\frac{1}{64} \sum_{i \in [s]} \widehat{D_i}$, and if not, we increment $i$ and sample $2^i$ samples, and repeat until either $2^i > k$ or the total weight $\sum_{i \in [s]} D_i$ is less than $\frac{1}{64} \sum_{i \in [s]} \widehat{D_i}$.

Note that this takes $\mathcal{O}\left(\log n \log k\right)$ rounds of sampling, which is enough for the failure probability of the $L_1$ sampling procedure. Moreover, since we can use $\mathcal{O}\left(\log \log n\right)$ bits of communication to approximate $L_1$ sampling by rounding the values of $D_i$ to a power of 2 and returning the exponent. Since each time the total weight $\sum_{i \in [s]} D_i$ is less than $\frac{1}{64} \sum_{i \in [s]} \widehat{D_i}$, we require all $s$ sites to update their weights, then the communication is $\tilde{\mathcal{O}}\left(s \log n + kd \log n\right)$ across the $\mathcal{O}\left(\log n \log k\right)$ rounds of communication.

**Applying sensitivity sampling in the blackboard model.** To apply the sensitivity sampling framework of Bansal et al. (2024), a constant approximation of the number of each cluster $|C_j|$ and the cost of each cluster $\mathrm{cost}(C_j, S)$ is necessary, where $S$ is a $(\mathcal{O}\left(1\right), \mathcal{O}\left(1\right))$-bicriteria approximation of the optimal solution. Each site needs $\mathcal{O}\left(\log \log n\right)$ bits to upload such a constant approximation for a cluster. Since there are $s$ sites and $\mathcal{O}\left(k\right)$ clusters, it would lead to $\mathcal{O}\left(sk \log \log n\right)$ bits of communication, which is prohibitive for our goal. Instead, we adapt Morris counters, which are used for approximate counting in the streaming model, to the distributed setting. We use a counter $r$ to store the logarithm of the number/cost of the cluster, and increase the counter by 1 with probability $\frac{1}{2^r}$ for every time we count the number/cost of the cluster. Each site applies the Morris counters sequentially and only uploads the increment on the blackboard. Similar to Morris counters, such a subroutine can return a constant approximation of the number/cost of the cluster. If all counters remain the same after counted by a site, that site needs to use $\mathcal{O}\left(1\right)$ bits to tell the next site that nothing needs to be updated, which would use at most $\mathcal{O}\left(1\right)$. Otherwise, $\mathcal{O}\left(\log \log n\right)$ bits are needed to update a changed counter. Since each counter can increase at most $\mathcal{O}\left(\log n\right)$ times and there are $\mathcal{O}\left(k\right)$ numbers of number/cost of the cluster to be counted, it would use $\mathcal{O}\left(k \log n\right)$ bits to upload all the updates. Therefore, at most $\tilde{\mathcal{O}}\left(s + k \log n\right)$ bits are needed to upload a constant approximation of the number of each cluster $|C_j|$ and the cost of each cluster $\mathrm{cost}(C_j, S)$.

Each site can apply the sensitivity sampling locally after knowing a constant approximation of the number of each cluster $|C_j|$ and the cost of each cluster $\mathrm{cost}(C_j, S)$. Combined with the efficient encoding for the $(1+\varepsilon)$-coreset, we can upload the coreset using $\mathcal{O}\left(\frac{(d \log \log n + \log k)k}{\min(\varepsilon^4, \varepsilon^{2+z})}\right)$ bits, which achieves our desired bounds for the blackboard model.

**Coordinator/message-passing model.** For the message-passing/coordinator model, our starting point is once again adaptive sampling, which can be performed over $\mathcal{O}\left(k\right)$ rounds to achieve a

constant-factor approximation $C$ using $\mathcal{O}\left(skd \log n\right)$ total bits of communication (Balcan et al., 2013). We can again use sensitivity sampling on $C$ along with our efficient encoding scheme to achieve the desired communication for the subsequent $(1 + \varepsilon)$-coreset. Thus, the main question again is how to improve the constant-factor protocol.

To that end, we first introduce a subroutine EFFICIENTCOMMUNICATION that can send the location of the point using low communication cost. The intuition behind EFFICIENTCOMMUNICATION is similar to the efficient encoding, which uses the points a site owns to encode the point we want to send. The only difference is that we use different points to efficiently encode the coordinates for each dimension, rather than just using a single point to encode the point we want to send.

Recall that there exists GREATERTHAN protocol, which given two $\log n$ bit integers, figures out which one is greater with constant probability, using $\tilde{\mathcal{O}}\left(\log \log n\right)$ bits of communication. Moreover, this procedure can be repeated $\mathcal{O}\left(\log(sk)\right)$ times to achieve a failure probability $\frac{1}{\text{poly}(sk)}$ to union bound over $\text{poly}(sk)$ possible steps. With the help of GREATERTHAN, we can compare the coordinates of the points in the $i$-th dimension $x_j^{(i)}$ a site owns with the coordinate of the point to receive $y^{(i)}$. By a binary search, we can find the closest coordinate $x_j^{(i)}$ to $y^{(i)}$, which requires $\mathcal{O}\left(\log n\right)$ search times, since the site owns at most $n$ points. By another binary search, we can further find $\Delta y^{(i)}$ that is a $(1 + \varepsilon)$-approximation of $|x_j^{(i)} - y^{(i)}|$. Since $|x_j^{(i)} - y^{(i)}| = \text{poly}(n)$, we need $\mathcal{O}\left(\log \log n\right)$ searches to find $\Delta y^{(i)}$. Since the coordinator needs to send the approximate coordinates of the points sampled by adaptive sampling to each site, the total communication cost for adaptive sampling is $\tilde{\mathcal{O}}\left(dsk \log n\right)$ bits, which is still prohibitive for our goal.

To address this, we first generate a $(1 + \mathcal{O}\left(\varepsilon\right))$-coreset for each site, so that the size of the dataset of each site is $\mathcal{O}\left(k\right)$, which leads $\mathcal{O}\left(\log n\right)$ searches to find the closest coordinate $x_j^{(i)}$ to $y^{(i)}$. Furthermore, we observe that we only need the approximate cost for each point for adaptive sampling. Hence, we can apply Johnson-Lindenstrauss to project the dataset down to $\mathcal{O}\left(\log(sk)\right)$ dimensions. Therefore, the total communication cost for adaptive sampling is $sk\,\text{polylog}(\log n, s, k)$ bits.

After achieving an $(\mathcal{O}\left(1\right), \mathcal{O}\left(1\right))$-bicriteria approximation for the optimal solution, we can communicate a constant approximation of the number and cost of each cluster using a total $\mathcal{O}\left(sk \log \log n\right)$ bits of communication. However, each site may assign the points to an incorrect cluster since it only has an approximate location of $S$. Fortunately, the sensitivity sampling procedure of Bansal et al. (2024) still returns a $(1 + \varepsilon)$-coreset if we assign each point to a center so that the distance between them is close to the distance between the point and the center closest to it. Therefore, combined with EFFICIENTCOMMUNICATION, we can upload the $(1 + \varepsilon)$-coreset using $\frac{dk}{\min(\varepsilon^4, \varepsilon^{2+z})}\,\text{polylog}(\log n, k, \frac{1}{\varepsilon})$ bits, which achieves our desired bounds for the blackboard model.

## B  PRELIMINARIES

For a positive integer $n$, we denote by $[n]$ the set $\{1, \ldots, n\}$. We use $\text{poly}(n)$ to represent an arbitrary polynomial function in $n$, and $\text{polylog}(n)$ to denote a polynomial function in $\log n$. An event is said to occur with high probability if it holds with probability at least $1 - 1/\text{poly}(n)$.

Throughout this paper, we focus on Euclidean $(k, z)$-clustering. Given vectors $x, y \in \mathbb{R}^d$, we let $\text{dist}(x, y)$ denote their Euclidean distance, defined as $\|x - y\|_2$, where $\|x - y\|_2^2 = \sum_{i=1}^d (x_i - y_i)^2$. For a point $x$ and a set $S \subset \mathbb{R}^d$, we extend the notation $\text{dist}(x, S) := \min_{s \in S} \text{dist}(x, s)$. We also use $\|x\|_z$ to denote the $L_z$ norm of $x$, given by $\|x\|_z^z = \sum_{i=1}^d x_i^z$. Given a fixed exponent $z \geq 1$ and finite sets $X, C \subset \mathbb{R}^d$ with $X = \{x_1, \ldots, x_n\}$, we define the clustering cost $\text{cost}(X, C)$ as

$$\text{cost}(X, C) := \sum_{i=1}^n \text{dist}(x_i, C)^z.$$

We now recall generalized versions of the triangle inequality.

**Fact B.1** (Generalized triangle inequality). *For any $z \geq 1$ and any points $x, w, y \in \mathbb{R}^d$, it holds that*

$$\text{dist}(x, y)^z \leq 2^{z-1}(\text{dist}(x, w)^z + \text{dist}(w, y)^z).$$

**Fact B.2** (Claim 5 in Sohler & Woodruff (2018)). *Suppose $z \geq 1$, $x, y \geq 0$, and $\varepsilon \in (0, 1]$. Then*

$$(x + y)^z \leq (1 + \varepsilon) \cdot x^z + \left(1 + \frac{2z}{\varepsilon}\right)^z \cdot y^z.$$

Next, we define the notion of a strong coreset for $(k, z)$-clustering.

**Definition B.3** (Coreset). *Let $\varepsilon > 0$ be an approximation parameter, and let $X = \{x_1, \ldots, x_n\} \subset \mathbb{R}^d$ be a set of points. A* coreset *for $(k, z)$-clustering consists of a weighted set $(S, w)$ such that for every set $C \subset \mathbb{R}^d$ of $k$ centers,*

$$(1 - \varepsilon) \sum_{t=1}^{n} dist(x_t, C)^z \leq \sum_{q \in S} w(q) dist(q, C)^z \leq (1 + \varepsilon) \sum_{t=1}^{n} dist(x_t, C)^z.$$

We will use the following known coreset construction for $(k, z)$-clustering:

**Theorem B.4.** *(Cohen-Addad et al., 2022a; Huang et al., 2024; Cohen-Addad et al., 2022b) For any $\varepsilon \in (0, 1)$, there exists a coreset construction for Euclidean $(k, z)$-clustering that samples $\tilde{\mathcal{O}}\left(\min\left(\frac{1}{\varepsilon^2} k^{2-\frac{z}{z+2}}, \frac{k}{\min(\varepsilon^4, \varepsilon^{2+z})}\right)\right)$ weighted points from the input dataset.*

We also recall the classical Johnson-Lindenstrauss lemma, which enables dimensionality reduction while approximately preserving pairwise distances:

**Theorem B.5** (Johnson-Lindenstrauss lemma). *(Johnson & Lindenstrauss, 1984) Let $\varepsilon \in (0, 1/2)$ and $m = \mathcal{O}\left(\frac{1}{\varepsilon^2} \log n\right)$. Given a set $X \subset \mathbb{R}^d$ of $n$ points, there exists a family of random linear maps $\Pi : \mathbb{R}^d \to \mathbb{R}^m$ such that with high probability over the choice of $\pi \sim \Pi$, for all $x, y \in X$,*

$$(1 - \varepsilon)\|x - y\|_2 \leq \|\pi x - \pi y\|_2 \leq (1 + \varepsilon)\|x - y\|_2.$$

We remark that exist more efficient dimensionality reduction techniques for $(k, z)$-clustering (Makarychev et al., 2019; Izzo et al., 2021) though for the purposes of our guarantees, Johnson-Lindenstrauss suffices.

Finally, we recall Hoeffding's inequality, a standard concentration bound:

**Theorem B.6** (Hoeffding's inequality). *Let $X_1, \ldots, X_n$ be independent random variables with $a_i \leq X_i \leq b_i$ for each $i$. Let $S_n = \sum_{i=1}^{n} X_i$. Then for any $t > 0$,*

$$\mathbb{P}\left(|S_n - \mathbb{E}[S_n]| > t\right) \leq 2 \exp\left(-\frac{t^2}{\sum_{i=1}^{n}(b_i - a_i)^2}\right).$$

We next recall the following formulation of sensitivity sampling paradigm, given in Algorithm 2.

---

**Algorithm 2** SENSITIVITYSAMPLING

---

**Input:** Dataset $X = \{x_1, \ldots, x_n\} \subset [\Delta]^d$ and integer $k$
**Output:** Weighted set $A = \{(a_i, w_{a_i})\}$
 1: Compute an $\mathcal{O}(1)$-approximation $S = \{s_1, s_2, \cdots, s_k\}$. Let $C_j \subset X$ be the cluster centered at $s_j$. For a point $x \in C_j$, let $\Delta_p := cost(C_j, S)/|C_j|$ denote the average cost of $C_j$
 2: For $x \in C_j$, let

$$\mu(x) := \frac{1}{4} \cdot \left(\frac{1}{k|C_j|} + \frac{cost(x, S)}{k\, cost(C_j, S)} + \frac{cost(x, S)}{cost(X, S)} + \frac{\Delta_x}{cost(X, S)}\right).$$

 3: $m \leftarrow \tilde{\mathcal{O}}\left(k/\varepsilon^2 \cdot \min(\varepsilon^{-2}, \varepsilon^{-z})\right)$, $A \leftarrow \emptyset$
 4: **for** $i \leftarrow 1$ to $m$ **do**
 5:     Sample point $a_i$ independently from the distribution $\mu$, $A \leftarrow A \cup \{(a_i, \frac{1}{m \cdot \mu(a_i)})\}$
 6: **end for**
 7: **return** $A$

---

**Theorem B.7.** *(Bansal et al., 2024) Sensitivity sampling, c.f., Algorithm 2, outputs a $(1+\varepsilon)$-coreset of size $\tilde{\mathcal{O}}\left(\frac{k}{\min(\varepsilon^4, \varepsilon^{2+z})}\right)$ for Euclidean $(k,z)$-clustering with probability at least $0.99$.*

We next recall the following distributed protocol for determining the larger of two integers.

**Theorem B.8.** *(Nisan, 1993) Given two $\mathcal{O}(\log n)$ bit integers $X$ and $Y$, there exists a protocol* GREATERTHAN *that uses $\tilde{\mathcal{O}}(\log \log n)$ total bits of communication and determines whether $X = Y$, $X < Y$ or $X > Y$.*

We recall the following protocol for applying the Johnson-Lindenstrauss transformation using low communication cost.

**Theorem B.9.** *(Kane & Nelson, 2010) For any integer $d > 0$, and any $0 < \varepsilon, \delta < \frac{1}{2}$, there exists a family $\mathcal{A}$ of matrices in $\mathbb{R}^{\ell \times d}$ with $\ell = \Theta\left(\varepsilon^{-2} \log \frac{1}{\delta}\right)$ such that for any $x \in \mathbb{R}^d$,*

$$\Pr_{A \in \mathcal{A}} \left[\|Ax\|_2 \notin [(1-\varepsilon)\|x\|_2, (1+\varepsilon)\|x\|_2]\right] < \delta.$$

*Moreover, $A \in \mathcal{A}$ can be sampled using $\mathcal{O}\left(\log\left(\frac{l}{\delta}\right) \log d\right)$ random bits and every matrix $A \in \mathcal{A}$ has at most $\alpha = \Theta\left(\varepsilon^{-1} \log\left(\frac{1}{\delta}\right) \log\left(\frac{l}{\delta}\right)\right)$ non-zero entries per column, and thus $Ax$ can be evaluated in $\mathcal{O}\left(\alpha \cdot \|x\|_0\right)$ time if $A$ is written explicitly in memory.*

**Adaptive sampling.** We now introduce the adaptive sampling algorithm from Aggarwal et al. (2009) and generalize the analysis to $(k,z)$-clustering.

---

**Algorithm 3** ADAPTIVESAMPLING

---

**Input:** Dataset $X = \{x_1, \ldots, x_n\} \subset [\Delta]^d$; approximation parameter $\gamma \geq 1$
**Output:** Bicriteria approximation $S$ for $(k,z)$-clustering on $X$
1: $S \leftarrow \emptyset, N \leftarrow \mathcal{O}(k), \gamma \leftarrow \Theta(1)$
2: **for** $t \leftarrow 1$ to $N$ **do**
3:     Choose $s_t$ to be $x_i \in X$ with probability $\frac{d_i}{\sum d_i}$, where $\frac{1}{\gamma} \cdot d_i \leq \text{cost}(x_i, S) \leq \gamma \cdot d_i$
4:     $S \leftarrow S \cup \{s_t\}$
5: **end for**
6: **return** $S$

---

(Aggarwal et al., 2009) showed that adaptive sampling achieves a bicriteria approximation for $k$-means, i.e., $z = 2$. For completeness, we shall extend the proof to show that it works for weighted case and any $z \geq 1$, though we remark that the techniques are standard.

**Theorem B.10.** *There exists an algorithm, c.f., Algorithm 3 that outputs a set $S$ of $\mathcal{O}(k)$ points such that with probability $0.99$, $\text{cost}(S, X) \leq \mathcal{O}(1) \cdot \text{cost}(C_{OPT}, X)$, where $C_{OPT}$ is an optimal $(k,z)$-clustering of $X$.*

As stated, the version of adaptive sampling in Algorithm 3 requires updating $\text{cost}(x, S)$ after each update, leading to a high cost of communication. We will introduce the lazy sampling algorithm. The combination of our updated adaptive sampling algorithm and the lazy sampling algorithm can provide a lower number of updates required, and thus a lower overall communication cost, while still guaranteeing the result of the $(\mathcal{O}(1), \mathcal{O}(1))$-bicriteria approximation.

Suppose that we have $s$ sites, every site has a local dataset $X_i$ and every data point $x$ has a sampling weight $d_x$. Let $D_i = \sum_{x \in X_i} d_x$ and $D = \sum_{i=1}^s D_i$. Our goal is to sample a point $x$ with probability $p_x = \frac{d_x}{D}$. In the case of adaptive sampling, $d_x$ is just $\text{cost}(x, S)$. To avoid a high cost of communication, our strategy is that the coordinator only holds $\widetilde{D_i}$, an approximation of real $D_i$ that $D_i \leq \widetilde{D_i} < \lambda D_i$, and only updates $\widetilde{D_i}$ when it deviates far from $D_i$. Then we can use $\frac{d_x}{\widetilde{D}}$ to sample the point, where $\widetilde{D} = \sum_{i=1}^s \widetilde{D_i}$.

Since $\widetilde{D}$ may be larger than $D$, the site $i$ will send $\perp$ with probability $\frac{\widetilde{D_i} - D_i}{\widetilde{D_i}}$. Fortunately, the probability of returning $\perp$ is constant, and the probability that we sample $x$ conditioned on not returning $\perp$ is just $\frac{d_x}{D}$. Therefore, by Markov's inequality, we can sample at least $N$ points under

---

**Algorithm 4** LAZYSAMPLING

---

**Input:** $\{d_x\}_{x \in X}$, the sampling weight; $\{\widetilde{D_i}\}_{i \in [s]}$, a constant approximation to $\{D_i\}_{i \in [s]}$ that $D_i \leq \widetilde{D_i} \leq \lambda D_i$, where $D_i = \sum_{x \in X_i} d_x$ for every site $i$

**Output:** Either $\perp$ or a point $x$ sampled under distribution $\frac{d_x}{D}$, where $D = \sum_{x \in X} d_x$

  1: $\widetilde{D} \leftarrow \sum_{i=1}^{s} \widetilde{D_i}, x \leftarrow \perp$
  2: The coordinator generates an index $i$ with probability $= \frac{\widetilde{D_i}}{\widetilde{D}}$
  3: The coordinator sends sampling request to site $i$
  4: $x \leftarrow y$ with probability $\frac{d_y}{\widetilde{D_i}}$
  5: **return** $x$

---

the distribution $\frac{d_x}{D}$ with probability at least $0.99$ after repeating LAZYSAMPLING a total of $\mathcal{O}(N)$ times.

**Lemma B.11.** *Let $x$ be the result returned by Algorithm 4 and $\mathcal{E}$ be the event that Algorithm 4 does not return $\perp$. If $\sum D_j \leq \sum \widetilde{D_j} < \lambda \sum D_j$, $\mathbf{Pr}[\mathcal{E}] \geq \frac{1}{\lambda}$ and $\mathbf{Pr}[x = y | \mathcal{E}] = \frac{d_y}{D}$. Furthermore, there exists some constant $\gamma \geq 1$ such that after running Algorithm 4 a total of $\gamma N$ times, it will return at least $N$ points sampled by the distribution $\frac{d_x}{D}$ with probability at least $0.99$.*

*Proof.* Let $\mathcal{F}_i$ be the event that the result is sampled by the site $i$. Since $\mathbf{Pr}[\mathcal{F}_i] = \frac{\widetilde{D_i}}{\widetilde{D}}$ and $\mathbf{Pr}[\mathcal{E} | \mathcal{F}_i] = \sum_{x \in X_i} \frac{d_x}{\widetilde{D_i}} = \frac{D_i}{\widetilde{D_i}}$, then by the law of total probability,

$$\mathbf{Pr}[\mathcal{E}] = \sum_{i=1}^{s} \mathbf{Pr}[\mathcal{E} | \mathcal{F}_i] \cdot \mathbf{Pr}[\mathcal{F}_i] = \sum_{i=1}^{s} \frac{D_i}{\widetilde{D_i}} \cdot \frac{\widetilde{D_i}}{\widetilde{D}} = \frac{D}{\widetilde{D}} \geq \frac{D}{\lambda D} = \frac{1}{\lambda}.$$

For any $y \in X_i$, we have

$$\mathbf{Pr}[x = y | \mathcal{E}] = \frac{\mathbf{Pr}[x = y]}{\mathbf{Pr}[\mathcal{E}]} = \frac{1}{\mathbf{Pr}[\mathcal{E}]} \cdot \frac{\widetilde{D_i}}{\widetilde{D}} \cdot \frac{d_y}{\widetilde{D_i}} = \frac{\widetilde{D}}{D} \cdot \frac{\widetilde{D_i}}{\widetilde{D}} \cdot \frac{d_y}{\widetilde{D_i}} = \frac{d_y}{D}.$$

Furthermore, let

$$Z_i = \begin{cases} 0, \text{if LAZYSAMPLING returns } \perp \\ 1, \text{otherwise}. \end{cases}$$

After running LAZYSAMPLING for $\gamma N$ times, the number of times LAZYSAMPLING does not return $\perp$ is just $\sum_{i=1}^{\gamma N} Z_i$. Since $\mathbb{E}[Z_i] = \mathbf{Pr}[\mathcal{E}] \geq \frac{1}{\gamma}$, we have $\mathbb{E}\left[\sum_{i=1}^{\gamma N} Z_i\right] \geq \frac{\gamma}{\lambda} N$. By Markov's inequality, there exists some $\gamma \geq 1$ such that $\mathbf{Pr}\left[\sum_{i=1}^{\gamma N} Z_i < N\right] \leq 0.01$. Hence, LAZYSAMPLING will sample at least $N$ points with probability at least $0.99$. $\qquad\square$

Finally, in Algorithm 5 we recall the standard algorithm that provides an efficient approximate encoding of a number by storing the exponent of the number after rounding to a power of a fixed base $\lambda$.

---

**Algorithm 5** POWERAPPROX

---

**Input:** Number to be encoded $m$, base $\lambda > 1$
**Output:** An integer $i$ that $\lambda^i$ is a $\lambda$ approximation of $m$
  1: Let $i$ be the integer such that $m \leq \lambda^i < \lambda m$
  2: **return** $i$

---

**Theorem B.12.** *The output from Algorithm 5 can be used to compute a number $\widehat{m}$ such that $m \leq \widehat{m} < \lambda m$.*

**Efficient encoding for coreset construction for $(k, z)$-clustering.** We recall the following efficient encoding for a given coreset for $(k, z)$-clustering given by Cohen-Addad et al. (2025). Given a dataset $X$, which may represent either the original inputs or a weighted coreset derived from some larger dataset, we begin by computing a constant-factor approximation $C'$ for the $(k, z)$-clustering problem on $X$. For every $x \in X$, let $\pi_{C'}(x)$ denote the nearest center in $C'$ to $x$. Each point $x$ can then be decomposed as $x = \pi_{C'}(x) + (x - \pi_{C'}(x))$, separating it into its closest center and a residual vector.

To obtain a $(1 + \varepsilon)$-approximation for $(k, z)$-clustering, it suffices to round each coordinate of the offset vector $x - \pi_{C'}(x)$ to the nearest power of $(1 + \varepsilon')$, where $\varepsilon' = \text{poly}\left(\varepsilon, \frac{1}{d}, \frac{1}{\log(n\Delta)}\right)$. Let $y'$ denote this rounded vector.

We then encode each point as $x' = \pi_{C'}(x) + y'$, storing both the index of the center $\pi_{C'}(x)$ and the exponent values of the rounded offset. This representation uses $\mathcal{O}\left(\log k + d \log\left(\frac{1}{\varepsilon}, d, \log(n\Delta)\right)\right)$ bits per point. The full algorithm is detailed in Algorithm 6.

---

**Algorithm 6** Compact Encoding for Coreset Generation in $(k, z)$-Clustering

---

**Input:** Dataset $X \subset [\Delta]^d$ with weights $w(\cdot)$, accuracy parameter $\varepsilon \in (0, 1)$, number of clusters $k$, parameter $z \geq 1$, failure probability $\delta \in (0, 1)$
**Output:** $(1 + \varepsilon)$-approximate coreset for $(k, z)$-clustering
  1: $\varepsilon' \leftarrow \frac{\text{poly}(\varepsilon^z)}{\text{poly}(k, \log(nd\Delta))}$
  2: Compute a constant-factor $(k, z)$-clustering solution $C'$ on $X$
  3: **for** each $x \in X$ **do**
  4:     Set $c'(x)$ as the nearest center to $x$ in $C'$
  5:     Compute the residual vector $y' = x - c'(x)$
  6:     Round each coordinate of $y'$ to the nearest power of $(1 + \varepsilon')$ to obtain $y$
  7:     Define $x' = (c'(x), y)$, where $y$ stores the exponent for each coordinate
  8:     Add $x'$ to the new set: $X' \leftarrow X' \cup \{x'\}$
  9: **end for**
 10: **return** $(C', X')$

---

We have the following guarantees for the efficient encoding from Cohen-Addad et al. (2025):

**Lemma B.13.** *Cohen-Addad et al. (2025) Let $\varepsilon \in \left(0, \frac{1}{2}\right)$, and let $X'$ be the weighted dataset $S$, constructed using the offsets from the center set $C'$ as defined in Algorithm 6. Then, for any set of centers $C \subset [\Delta]^d$ with $|C| \leq k$, the following holds:*

$$(1 - \varepsilon) \cdot \text{cost}(C, X) \leq \text{cost}(C, X') \leq (1 + \varepsilon) \cdot \text{cost}(C, X).$$

**Lemma B.14.** *Cohen-Addad et al. (2025) Let $X$ be a coreset where the point weights lie within the range $[1, \text{poly}(nd\Delta)]$. Then the transformed set $X'$ forms a $(1 + \varepsilon)$-strong coreset for $X$, and its total space requirement is $\mathcal{O}\left(dk \log(n\Delta)\right) + |X| \cdot \text{polylog}\left(k, \frac{1}{\varepsilon}, \log(nd\Delta), \log \frac{1}{\delta}\right)$ bits.*

## C  ADAPTIVE SAMPLING FOR $(k, z)$-CLUSTERING

We first give a high-level overview of why ADAPTIVESAMPLING returns a bicriteria approximation. Let $\{A_j\}_{j=1}^k$ be the clusters that correspond to an optimal $(k, z)$-clustering. Then we can achieve an $\mathcal{O}(1)$-approximation to the optimal cost if the cost for every cluster $A_j$ induced by $S$ is already an $\mathcal{O}(1)$-approximation for $\text{cost}(A_j, C_{\text{OPT}})$, where $C_{\text{OPT}}$ is the set of centers for the optimal solution. We define a cluster $A_j$ as a good cluster if its cost is an $\mathcal{O}(1)$-approximation to $\text{cost}(A_j, C_{\text{OPT}})$, and otherwise we call $A_j$ a bad cluster. We can show that with constant probability, ADAPTIVESAMPLING samples a center that can transform a bad cluster to a good one. Then, by Markov's inequality, it follows that we can eliminate all bad clusters in $\mathcal{O}(k)$ rounds of sampling, resulting in a constant-factor approximation.

We use the following definition of good and bad clusters.

**Definition C.1.** *Let $S_i$ be the sampled set $S$ in the $i$-th round in Algorithm 3 and let $C_{OPT} = \{c_1, c_2, \cdots, c_k\}$ be an optimal solution for $(k, z)$-clustering.*

*For $j \in [k]$, let $A_j$ be the points in $X$ assigned to $c_j$ in the optimal clustering, breaking ties arbitrarily, i.e., $A_j = \{x \in X : dist(x, c_j) \leq dist(x, c_\ell), \forall \ell \in [k]\}$. We define*

$$\mathbf{Good}_i = \{A_j : \text{cost}(A_j, S_i) \leq \gamma_z \cdot \text{cost}(A_j, C_{OPT})\}$$
$$\mathbf{Bad}_i = \{A_1, A_2, \cdots, A_k\} \setminus \mathbf{Good}_i$$

*where $\gamma_z = 2 + (3 + 6z)^z$.*

By definition, if every cluster is a good cluster, we can guarantee that $S$ is an $(\mathcal{O}(1), \mathcal{O}(1))$-bicriteria approximation. We will prove that either $S$ is already an $(\mathcal{O}(1), \mathcal{O}(1))$-bicriteria approximation, or else we will reduce the number of bad clusters after every sampling with some constant probability, which would imply that we can achieve an $\mathcal{O}(1)$-approximation after sampling $\mathcal{O}(k)$ points.

**Lemma C.2.** *Suppose $\text{cost}(X, S_i) > 2\gamma^2 \cdot \gamma_z \text{cost}(X, C_{OPT})$, then*

$$\mathbf{Pr}\left[|\mathbf{Bad}_{i+1}| < |\mathbf{Bad}_i|\right] \geq \delta$$

*for some constant $\delta > 0$.*

Lemma C.2 is the generalization of Lemma 5 in Aggarwal et al. (2009). We will prove Lemma C.2 by breaking the proof into several lemmas. We first consider a specific parameterization of the generalized triangle inequality.

**Lemma C.3.** *For any $x, y, \mu \in \mathbb{R}^d$,*

$$dist(x, y)^z \leq 2 \cdot dist(x, \mu)^z + (1 + 2z)^z \cdot dist(y, \mu)^z.$$

*Proof.* The claim follows from applying Fact B.2 with $\varepsilon = 1$, so that by the triangle inequality, $\text{dist}(x, y)^z \leq (\text{dist}(x, \mu) + \text{dist}(y, \mu))^z \leq 2 \cdot \text{dist}(x, \mu)^z + (1 + 2z)^z \cdot \text{dist}(y, \mu)^z.$ □

We first show that we will sample a point from a bad cluster with constant probability every round. This statement is analogous to Lemma 1 in Aggarwal et al. (2009).

**Lemma C.4.** *In the $i$-th round of our algorithm, either $\text{cost}(X, S_i) \leq 2\gamma^2 \cdot \gamma_z \text{cost}(X, C_{OPT})$ or else the probability of picking a point from some cluster in $\mathbf{Bad}_i$ is at least $\frac{1}{2}$.*

*Proof.* Suppose $\text{cost}(X, S_{i-1}) > 2\gamma^2 \cdot \gamma_z \text{cost}(X, C_{OPT})$. Then the probability of picking $x$ from some bad cluster is

$$\mathbf{Pr}\left[x \in \mathbf{Bad}_i\right] = \frac{\sum_{A_j \in \mathbf{Bad}_i} \sum_{x_l \in A_j} d_l}{\sum_{x_l \in X} d_l} = 1 - \frac{\sum_{A_j \in \mathbf{Good}_i} \sum_{x_l \in A_j} d_l}{\sum_{x_l \in X} d_l}.$$

Here, we remark that $d_l$ is the distance from $x_l$ to $S_{i-1}$ as defined in Algorithm 3. Since $\text{cost}(A_j, S_{i-1}) \leq \gamma_z \text{cost}(A_j, C_{OPT})$ if $A_j \in \mathbf{Good}_i$, and $\text{cost}(X, S_{i-1}) > 2\gamma^2 \cdot \gamma_z \text{cost}(X, C_{OPT})$ by the condition we hold,

$$\mathbf{Pr}\left[x \in \mathbf{Bad}_i\right] \geq 1 - \frac{\gamma^2 \cdot \gamma_z \sum_{A_j \in \mathbf{Good}_i} \text{cost}(A_j, C_{OPT})}{2\gamma^2 \cdot \gamma_z \text{cost}(X, C_{OPT})} \geq 1 - \frac{1}{2} = \frac{1}{2}.$$

□

Consider a fixed bad cluster $A \in \mathbf{Bad}_i$. Let $m = w(A) = \sum_{x \in A} w(x)$, where $w(x)$ is the weight of $x$, and define $r = \left(\frac{\text{cost}(A, C_{OPT})}{m}\right)^{1/z}$. Denote $\mu$ as the center in $C_{OPT}$ corresponding to $A$. Let $y$ be the point closest to $\mu$ in $S_{i-1}$. We will show that $y$ (and thus every sampled point in $S_{i-1}$) is far from $\mu$.

**Lemma C.5.** *$dist(y, \mu) \geq 3r$.*

*Proof.* Because $A$ is a fixed bad cluster,

$$\gamma_z \, \text{cost}(A, C_{\text{OPT}}) \leq \text{cost}(A, S_{i-1}) = \sum_{x \in A} w(x) \min_{c \in S_{i-1}} \text{dist}(x, c)^z \leq \sum_{x \in A} w(x) \text{dist}(x, y)^z.$$

By Lemma C.3,

$$\sum_{x \in A} w(x) \text{dist}(x, y)^z \leq \sum_{x \in A} \left( 2 \cdot w(x) \cdot \text{dist}(x, \mu)^z + (1 + 2z)^z \cdot w(x) \cdot \text{dist}(y, \mu)^z \right)$$

$$= 2 \, \text{cost}(A, C_{\text{OPT}}) + (1 + 2z)^z \cdot m \cdot \text{dist}(y, \mu)^z.$$

Hence

$$(1 + 2z)^z \cdot m \cdot \text{dist}(y, \mu)^z \geq (\gamma_z - 2) \, \text{cost}(A, C_{\text{OPT}}).$$

Since $\gamma_z = 3^z(1 + 2z)^z + 2$,

$$3r = \left( \frac{\gamma_z - 2}{(1 + 2z)^z} \cdot \frac{\text{cost}(A, C_{\text{OPT}})}{m} \right)^{1/z} \leq \text{dist}(y, \mu).$$

$\square$

Define $B(\alpha) = \{x \in A : \text{dist}(x, \mu) \leq \alpha r\}$ as the points in the fixed bad cluster $A$ that are within distance $\alpha r$ from $\mu$. We will show that the bad cluster $A$ can be transformed into a good cluster if we sample some point $x$ close to the center $\mu$. This statement is analogous to Lemma 2 in Aggarwal et al. (2009).

**Lemma C.6.** *Let $A$ be any fixed bad cluster defined by $C_{OPT}$ and let $b \in B(\alpha)$, for $0 \leq \alpha \leq 3$. Then*

$$\text{cost}(A, S_{i-1} \cup \{b\}) \leq \gamma_z \cdot \text{cost}(A, C_{OPT}).$$

*Proof.*

$$\text{cost}(A, S_{i-1} \cup \{b\}) = \sum_{x \in A} w(x) \min_{c \in S_{i-1} \cup \{b\}} \text{dist}(x, c)^z \leq \sum_{x \in A} w(x) \cdot \text{dist}(x, b)^z.$$

By Lemma C.3,

$$\sum_{x \in A} w(x) \cdot \text{dist}(x, b)^z \leq \sum_{x \in A} \left( 2 \cdot w(x) \cdot \text{dist}(x, \mu)^z + (1 + 2z)^z \cdot w(x) \cdot \text{dist}(b, \mu)^z \right).$$

Since $b \in B(\alpha)$ and $0 \leq \alpha \leq 3$, $\text{dist}(b, \mu) \leq \alpha r \leq 3r$. Hence

$$\text{cost}(A, S_{i-1} \cup \{b\}) \leq 2 \, \text{cost}(A, C_{\text{OPT}}) + m(1 + 2z)^z(3r)^z$$

$$= 2 \, \text{cost}(A, C_{\text{OPT}}) + 3^z(1 + 2z)^z \, \text{cost}(A, C_{\text{OPT}})$$

$$= \gamma_z \, \text{cost}(A, C_{\text{OPT}}).$$

$\square$

We next show that most of the weights of the points of the fixed bad cluster $A$ fall into $B(\alpha)$. Recall that we define $m = w(A) = \sum_{x \in A} w(x)$. Let $w(B(\alpha)) = \sum_{x \in B(\alpha)} w(x)$. This statement is analogous to Lemma 3 in Aggarwal et al. (2009).

**Lemma C.7.**

$$w(B(\alpha)) \geq m \left( 1 - \frac{1}{\alpha^z} \right), \text{ for } 1 \leq \alpha \leq 3.$$

*Proof.*

$$\text{cost}(A, C_{\text{OPT}}) \geq \text{cost}(A \backslash B(\alpha), C_{\text{OPT}}) = \sum_{x \in A \backslash B(\alpha)} w(x) \min_{c \in C_{\text{OPT}}} \text{dist}(x, c)^z.$$

Since for any $x \in A$, $\mu$ is the nearest center to $x$ in $C_{\text{OPT}}$, then

$$\sum_{x \in A \backslash B(\alpha)} w(x) \min_{c \in C_{\text{OPT}}} \text{dist}(x, c)^z = \sum_{x \in A \backslash B(\alpha)} w(x) \text{dist}(x, \mu)^z.$$

Since for any $x \in A \backslash B(\alpha)$, $\text{dist}(x, \mu) \geq \alpha r$, then

$$\sum_{x \in A \backslash B(\alpha)} w(x)\text{dist}(x,\mu)^z \geq w(A \backslash B(\alpha)) \cdot (\alpha r)^z = \left(1 - \frac{w(B(\alpha))}{m}\right)m(\alpha r)^z.$$

Since $r = \left(\frac{\text{cost}(A, C_{\text{OPT}})}{m}\right)^{1/z}$,

$$\text{cost}(A, C_{\text{OPT}}) \geq \left(1 - \frac{w(B(\alpha))}{m}\right)m(\alpha r)^z = \left(1 - \frac{w(B(\alpha))}{m}\right)\alpha^z \, \text{cost}(A, C_{\text{OPT}}).$$

Therefore,

$$w(B(\alpha)) \geq m\left(1 - \frac{1}{\alpha^z}\right).$$

$\square$

We next show that the cost of points in $B(\alpha)$ is at least a constant fraction of the cost of the bad cluster $A$ and thus we will sample a point near $\mu$ with constant probability if we sample a point from $A$. This statement is analogous to Lemma 4 in Aggarwal et al. (2009).

**Lemma C.8.**

$$\mathbf{Pr}\left[x \in B(\alpha) | x \in A \text{ and } A \in \textbf{Bad}_i\right] \geq \frac{1}{\gamma^2}\left(1 - \frac{1}{\alpha^z}\right)\frac{(3-\alpha)^z}{\gamma_z}.$$

*Proof.* Recall that $y$ is the point closest to $\mu$ in $S_{i-1}$. We have

$$\text{cost}(A, S_{i-1}) = \sum_{x \in A} w(x) \min_{c \in S_{i-1}} \text{dist}(x, c)^z \leq \sum_{x \in A} w(x)\text{dist}(x, y)^z.$$

Let $d = \text{dist}(y, \mu)$. By Lemma C.3,

$$\text{cost}(A, S_{i-1}) \leq \sum_{x \in A} \left(2w(x)\text{dist}(x, \mu)^z + (1+2z)^z w(x)\text{dist}(y, \mu)^z\right) = 2\,\text{cost}(A, C_{\text{OPT}}) + m(1+2z)^z d^z.$$

Since $r = \left(\frac{\text{cost}(A, C_{\text{OPT}})}{m}\right)^{1/z}$, then $\text{cost}(A, S_{i-1}) \leq 2mr^z + m(1+2z)^z d^z$.

On the other hand, $\text{cost}(B(\alpha), S_{i-1}) = \sum_{x \in B(\alpha)} w(x) \min_{c \in S_{i-1}} \text{dist}(x, c)^z$. By triangle inequality, $\text{dist}(x, c) \geq \text{dist}(c, \mu) - \text{dist}(x, \mu) \geq \text{dist}(y, \mu) - \text{dist}(x, \mu)$. Since $x \in A$ and $A$ is a bad cluster, then $\text{dist}(y, \mu) - \text{dist}(x, \mu) \geq 0$, so that

$$\text{cost}(B(\alpha), S_{i-1}) \geq \sum_{x \in B(\alpha)} w(x)(\text{dist}(c, \mu) - \text{dist}(x, \mu))^z \geq \sum_{x \in B(\alpha)} w(x)(\text{dist}(y, \mu) - \text{dist}(x, \mu))^z.$$

For $x \in B(\alpha)$, we have $\text{dist}(x, \mu) \leq \alpha r$, so that $\text{cost}(B(\alpha), S_{i-1}) \geq \sum_{x \in B(\alpha)} w(x)(\text{dist}(y, \mu) - \alpha r)^z$. By Lemma C.7, $w(B(\alpha)) \geq m\left(1 - \frac{1}{\alpha^z}\right)$, so $\text{cost}(B(\alpha), S_{i-1}) \geq m\left(1 - \frac{1}{\alpha^z}\right)(\text{dist}(y, \mu) - \alpha r)^z$.

Since $\frac{1}{\gamma} \cdot d_i \leq \text{cost}(x_i, S) \leq \gamma \cdot d_i$, cf. Algorithm 3, therefore

$$\mathbf{Pr}\left[x \in B(\alpha) | x \in A \text{ and } A \in \textbf{Bad}_i\right] = \frac{\sum_{x_i \in B(\alpha)} d_i}{\sum_{x_i \in A} d_i} \geq \frac{\frac{1}{\gamma}\text{cost}(B(\alpha), S_{i-1})}{\gamma\,\text{cost}(A, S_{i-1})}$$

$$\geq \frac{m}{\gamma^2}\left(1 - \frac{1}{\alpha^z}\right) \cdot \frac{(\text{dist}(y, \mu) - \alpha r)^z}{m(2r^z + (1+2z)^z\text{dist}(y, \mu)^z)}.$$

By computing the derivative, we can observe that

$$\frac{(\text{dist}(y, \mu) - \alpha r)^z}{2r^z + m(1+2z)^z\text{dist}(y, \mu)^z}$$

is an increasing function of $\text{dist}(y, \mu)$ for $\text{dist}(y, \mu) \geq 3r \geq \alpha r$. Hence

$$\mathbf{Pr}\left[x \in B(\alpha) | x \in A \text{ and } A \in \mathbf{Bad}_i\right] \geq \frac{m}{\gamma^2}\left(1 - \frac{1}{\alpha^z}\right) \cdot \frac{(3r - \alpha r)^z}{m(2r^z + (1 + 2z)^z(3r)^z)}$$
$$= \frac{1}{\gamma^2}\left(1 - \frac{1}{\alpha^z}\right)\frac{(3 - \alpha)^z}{\gamma_z}.$$

$\square$

Therefore, we will transform a bad cluster $A$ into a good cluster if we sample a point in $B(\alpha) \subset A$ for $a \leq 3$, and we will sample such a point in $A$ with constant probability, which means that we will reduce the number of bad clusters with constant probability.

**Lemma C.9.** *Suppose the point $x$ picked by our algorithm in the $i$-th round is from $A \in \mathbf{Bad}_i$ and $S_i = S_{i-1} \cup \{x\}$. Then*

$$\mathbf{Pr}\left[\text{cost}(A, S_i) \leq \gamma_z \cdot \text{cost}(A, C_{OPT}) | x \in A \text{ and } A \in \mathbf{Bad}_i)\right] \geq \delta,$$

*where*

$$\delta = \max \frac{1}{\gamma^2}\left(1 - \frac{1}{\alpha^z}\right)\frac{(3 - \alpha)^z}{\gamma_z}.$$

*Proof.* The claim follows from Lemma C.6 and Lemma C.8. $\square$

If $\text{cost}(X, S_i) > 2\gamma^2 \cdot \gamma_z \text{cost}(X, C_{OPT})$, we will sample a point from a bad cluster with constant probability, and a point sampled from a bad cluster will transform it to a good one with probability. Hence, we will reduce the number of bad clusters with probability.

We have proven that we will sample a point from a bad cluster with constant probability. We also proved that a sampled center from a bad cluster will transfer that bad cluster to a good one with constant probability. These two facts lead us to the conclusion that we will transfer a bad cluster to a good one with constant probability for every sampling, which is what Lemma C.2 claimed.

**Lemma C.2.** *Suppose $\text{cost}(X, S_i) > 2\gamma^2 \cdot \gamma_z \text{cost}(X, C_{OPT})$, then*

$$\mathbf{Pr}\left[|\mathbf{Bad}_{i+1}| < |\mathbf{Bad}_i|\right] \geq \delta$$

*for some constant $\delta > 0$.*

*Proof.* If $\text{cost}(X, S_i) > 2\gamma^2 \cdot \gamma_z \text{cost}(X, C_{OPT})$, by Lemma C.4, we will sample a point $x$ from some bad cluster with probability at least $\frac{1}{2}$. By Lemma C.9, if $x \in A$ is from some bad cluster, $\text{cost}(A, S_i) \leq \gamma_z \cdot \text{cost}(A, C_{OPT})$ with probability at least $\delta$. Hence $A$ will become a good cluster in $\mathbf{Good}_{i+1}$ with probability at least $\frac{\delta}{2}$. Therefore, $\mathbf{Pr}\left[|\mathbf{Bad}_{i+1}| < |\mathbf{Bad}_i|\right] \geq \frac{\delta}{2}$. $\square$

Now we justify the correctness of adaptive sampling for general $z \geq 1$.

**Theorem B.10.** *There exists an algorithm, c.f., Algorithm 3 that outputs a set $S$ of $\mathcal{O}(k)$ points such that with probability 0.99, $\text{cost}(S, X) \leq \mathcal{O}(1) \cdot \text{cost}(C_{OPT}, X)$, where $C_{OPT}$ is an optimal $(k, z)$-clustering of $X$.*

*Proof.* By Lemma C.3, the number of bad clusters will decrease with constant probability every round, unless $\text{cost}(X, S_i) \leq 2\gamma^2 \cdot \gamma_z \text{cost}(X, C_{OPT})$. Thus, by Markov's inequality, with probability 0.99, the number of bad clusters will be reduced to 0 after $\mathcal{O}(k)$ rounds unless $\text{cost}(X, S_i) \leq 2\gamma^2 \cdot \gamma_z \text{cost}(X, C_{OPT})$. Consequently, either there are no bad clusters or $\text{cost}(X, S_i) \leq 2\gamma^2 \cdot \gamma_z \text{cost}(X, C_{OPT})$. Both of these cases mean that we have $\text{cost}(A_j, S) \leq 2\gamma^2 \cdot \gamma_z \cdot \text{cost}(A_j, C_{OPT})$ for all $j \in [k]$, so $S$ becomes a constant approximation to the optimal $(k, z)$-clustering. $\square$

---

**Algorithm 7** INITIALIZATION

---

**Input:** Dataset $X_i$ given to each site $i \in [s]$

**Output:** A set $S$ with one sampled point $x$, approximate cost $\widetilde{D_i}$ for every site on blackboard
1: The coordinator samples a point $x$ and upload it on blackboard, $S \leftarrow \{x\}$
2: **for** $i \leftarrow 1$ to $s$ **do**
3:     $r_i \leftarrow$ POWERAPPROX$(\text{cost}(X_i, S), 2)$
4:     Write $r_i$ on the blackboard
5: **end for**
6: $\widetilde{D_i} \leftarrow 2^{r_i}$
7: **return** $S, \widetilde{D_i}$

---

# D  BLACKBOARD MODEL OF COMMUNICATION

## D.1  BICRITERIA APPROXIMATION IN THE BLACKBOARD MODEL

We first give the initialization algorithm for $(k, z)$-clustering of blackboard model in Algorithm 7, which generates a point $x$ and uploads an 2-approximation $\widetilde{D_i}$ of the total cost $D_i$ of every site on the blackboard.

After INITIALIZATION, we use LAZYSAMPLING to sample $\mathcal{O}(k)$ points to get an $(\mathcal{O}(1), \mathcal{O}(1))$-bicriteria approximation for the $(k, z)$-clustering.

---

**Algorithm 8** Lazy Adaptive $(k, z)$-Sampling for $k = \mathcal{O}(\log n)$

---

**Input:** The dataset $X_i$ every site $i$ owns, $i \in [s]$
**Output:** A set $S$ that is an $(\mathcal{O}(1), \mathcal{O}(1))$-bicriteria approximation for the $(k, z)$-clustering
1: Run INITIALIZATION to get $S$ and $\widetilde{D_i}$
2: $N \leftarrow \mathcal{O}(k)$
3: **for** $i \leftarrow 1$ to $N$ **do**
4:     $x \leftarrow$ LAZYSAMPLING$(\{\text{cost}(x, S)\}, \{\widetilde{D_j}\})$, and upload $x$ on blackboard
5:     $S \leftarrow S \cup \{x\}$
6:     Every site $j$ computes $D_j = \text{cost}(X_j, S)$
7:     $r_j \leftarrow$ POWERAPPROX$(D_j, 2)$, every site $j$ update $r_j$ on the blackboard if $r_j \neq \log_2 \widetilde{D_j}$
8:     $\widetilde{D_j} \leftarrow 2^{r_j}$
9: **end for**
10: **return** $S$

---

We first show that Algorithm 8 returns an $(\mathcal{O}(1), \mathcal{O}(1))$-bicriteria approximation for the $(k, z)$-clustering with $\mathcal{O}(sk \log \log n + kd \log n)$ bits of communication and $\mathcal{O}(k)$ rounds of communication.

**Lemma D.1.** *Algorithm 8 outputs a set $S$ with size $\mathcal{O}(k)$ using $\tilde{\mathcal{O}}(sk + kd \log n)$ bits of communication and $\mathcal{O}(k)$ rounds of communication, such that with probability at least $0.98$, $\text{cost}(S, X) \leq \mathcal{O}(1) \cdot \text{cost}(C_{OPT}, X)$, where $C_{OPT}$ is the optimal $(k, z)$-clustering of $X$.*

*Proof.* We show the correctness of the algorithm, i.e., that the clustering cost induced by $S$ is a constant-factor approximation of the optimal $(k, z)$-clustering cost. We also upper bound the number of points in $S$.

**Bicriteria approximation guarantee.** Since we update $\widetilde{D_j}$ every round, we meet the condition $D_j \leq \widetilde{D_j} < 2D_i$ every time we run LAZYSAMPLING$(\{\text{cost}(x, S)\}, \{\widetilde{D_j}\})$ in the algorithm. Thus, by Lemma B.11, we sample a point $x$ with probability $\frac{\text{cost}(x, S)}{\text{cost}(X, S)}$ if LAZYSAMPLING does not return $\bot$, and we can get $\mathcal{O}(k)$ points which are not $\bot$ with probability at least $0.99$ after $N = \mathcal{O}(k)$ rounds of lazy sampling. Therefore, we will get $\mathcal{O}(k)$ sampled points that satisfy the required conditions for adaptive sampling in ADAPTIVESAMPLING (Algorithm 3). Therefore, by Lemma C.2,

we reduce the size of $\mathbf{Bad}_i$ with some constant probability $p$ every round. It follows from Theorem B.10 that Algorithm 8 produces a constant approximation for $(k, z)$-clustering with probability at least 0.99.

**Communication complexity of Algorithm 8.** INITIALIZATION uses $\mathcal{O}\left(s \log \log n + d \log n\right)$ bits of communication. Uploading the location of a sampled point $x$ requires $\mathcal{O}\left(d \log n\right)$ bits because there are $d$ coordinates that need to be uploaded, and each needs $\mathcal{O}\left(\log n\right)$ bits. Every site needs $\mathcal{O}\left(1\right)$ bits to return $\perp$ to the coordinator. Then, all $s$ sites need to write $r_i$ on the blackboard. Since $r_i = \mathcal{O}\left(\log n\right)$, and encoding $r_i$ requires $\mathcal{O}\left(\log \log n\right)$ bits, it leads to $\mathcal{O}\left(s \log \log n\right)$ bits of communication. Since we will repeat the iteration for $\mathcal{O}\left(k\right)$ times, the total communication cost would be $\mathcal{O}\left(sk \log \log n + kd \log n\right)$ bits. Finally, we remark that since we will use two rounds of communication for each iteration of the $N = \mathcal{O}\left(k\right)$ rounds of sampling, the total number of rounds of communication is $\mathcal{O}\left(k\right)$. $\qquad\square$

## D.2 $L_1$ SAMPLING SUBROUTINE

In this section, we introduce an $L_1$ sampling algorithm in Algorithm 9. The main purpose of the $L_1$ sampling algorithm is to ultimately improve the communication complexity and the round complexity of the adaptive sampling approach for $k = \Omega(\log n)$, c.f., Appendix D.3.

The sampling algorithm L1SAMPLING can detect whether $\widetilde{D}$ is a good approximation of $D$. When $\mu^2 D \le \widetilde{D}$, which means that $\widetilde{D}$ is far from a good approximation, L1SAMPLING will return True to notify us to update the value of $\widetilde{D}$. On the other hand, when $\mu D > \widetilde{D}$, which means $\widetilde{D}$ is a good enough approximation of $D$, L1SAMPLING will return False, so we can save communication cost by avoiding unnecessary updates.

---

**Algorithm 9** $L_1$-Sampling: L1SAMPLING($\{\widetilde{D_j}\}, \{D_j\}, \mu, \delta$)

---

**Input:** $D_i = \text{cost}(X_i, S)$, the cost of points in site $i$; total cost $D = \sum_{i=1}^{s} D_i$; $\{\widetilde{D_i}\}_{i=1}^{s}$ and $\widetilde{D}$ written on blackboard, which are approximations to $\{D_i\}_{i=1}^{s}$ and $D$; $\mu > 1$, the distortion parameter; $\delta > 0$, the failure probability

**Output:** A boolean indicator for whether $\widetilde{D}$ is a $\mu^2$-approximation of $D$

1: $N \leftarrow \mathcal{O}\left(\log \frac{1}{\delta}\right), T \leftarrow 0, \lambda \leftarrow \frac{\mu}{4}, \alpha \leftarrow 2\mu$
2: **for** $i \leftarrow 1$ to $N$ **do**
3:     Sample a site $j$ with probability $\mathbf{Pr}\left[j\right] = \frac{\widetilde{D_j}}{\widetilde{D}}$, update $r_j \leftarrow$ POWERAPPROX($D_j, \lambda$) on blackboard
4:     $T \leftarrow T + \frac{1}{\mathbf{Pr}[j]} \cdot \lambda^{r_j}$
5: **end for**
6: **if** $\alpha \cdot T \le N \cdot \widetilde{D}$ **then**
7:     **return** True
8: **else**
9:     **return** False
10: **end if**

---

We now justify the correctness of Algorithm 9, i.e., we show that if the total mass has decreased significantly, then the algorithm will return True with probability $1 - \delta$ and similarly if the total mass has not decreased significantly, then the algorithm will return False with probability $1 - \delta$.

**Lemma D.2.** *Suppose $\widetilde{D_j} \ge D_j$ for all $j \in [s]$, and $\widetilde{D} \ge D$. If $\mu^2 D \le \widetilde{D}$, Algorithm 9* L1SAMPLING($\{\widetilde{D_j}\}, \{D_j\}, \mu, \delta$) *will return True with probability at least $1 - \delta$. If $\mu D > \widetilde{D}$,* L1SAMPLING($\{\widetilde{D_j}\}, \{D_j\}, \mu, \delta$) *will return False with probability at least $1 - \delta$.*

*Proof.* Let $\widehat{D_j} = \lambda^{r_j}$. Since $r_j$ is returned by POWERAPPROX($D_j, \lambda$), then by Theorem B.12, we have $D_j \le \lambda^{r_j} = \widehat{D_j} < \lambda D_j$.

Define the random variable

$$Z_i = \frac{1}{p_j} \widehat{D_j}, \text{ with probability } p_j = \frac{\widetilde{D_j}}{\widetilde{\widetilde{D}}}.$$

We will evaluate the range of $Z_i$ and $\mathbb{E}[Z_i]$ so that we can apply Hoeffding's inequality later. We have $\widetilde{D_j} \geq D_j$ and $\widehat{D_j} < \lambda D_j$. Hence,

$$Z_i = \frac{1}{p_j} \widehat{D_j} = \frac{\widetilde{\widetilde{D}}}{\widetilde{D_j}} \widehat{D_j} \leq \frac{\widetilde{\widetilde{D}}}{D_j} \lambda D_j = \lambda \widetilde{D}.$$

On the other hand, $Z_i$ must be non-negative by definition, so $Z_i \in [0, \lambda \widetilde{D}]$.

For the range of $\mathbb{E}[Z_i]$, we have

$$\mathbb{E}[Z_i] = \sum_{j=1}^{s} \frac{1}{p_j} \widehat{D_j} \cdot p_j = \sum_{j=1}^{s} \widehat{D_j}.$$

Since $\widehat{D_j} \in [D_j, \lambda D_j]$, then we have $\mathbb{E}[Z_i] \in [D, \lambda D]$. Similarly by linearity of expectation, since $T = \sum_{i=1}^{N} Z_i$, then we have $\mathbb{E}[T] = N \cdot \mathbb{E}[Z_i] \in [ND, N\lambda D]$. We perform casework on whether $\mu^2 D \leq \widetilde{D}$ or $\mu^2 D > \widetilde{D}$, corresponding to each of the two cases in the stated guarantee.

**Case 1:** $\mu^2 D \leq \widetilde{D}$. We first analyze the case $\mu^2 D \leq \widetilde{D}$. We define the event $\mathcal{N}$ that L1SAMPLING($\{\widetilde{D_j}\}, \{D_j\}, \mu, \delta$) returns False when $\mu^2 D \leq \widetilde{D}$ as the false negative event. Observe that this occurs in the algorithm if and only if $\alpha T > N\widetilde{D}$ when $\mu^2 D \leq \widetilde{D}$.

Notice that $\alpha T > N\widetilde{D}$ is impossible if $|T - \mathbb{E}[T]| \leq \lambda ND$ since $\mu^2 D \leq \widetilde{D}$ and $|T - \mathbb{E}[T]| \leq \lambda ND$ imply

$$\alpha T \leq \alpha(\mathbb{E}[T] + |T - \mathbb{E}[T]|) \leq \alpha(\lambda ND + \lambda ND) \leq \frac{2\alpha\lambda}{\mu^2} N\widetilde{D} = \frac{1}{\mu} N\widetilde{D} < N\widetilde{D}.$$

The first inequality is due to the triangle inequality. The second inequality comes from the fact that $\mathbb{E}[T] \in [ND, N\lambda D]$ and $|T - \mathbb{E}[T]| \leq \lambda ND$. The third inequality is due to the condition $\mu^2 D \leq \widetilde{D}$. The equality is because of $\alpha = 2\mu$ and $\lambda = \frac{\mu}{2}$. The last inequality is due to $\mu > 1$.

Hence under the condition that $\mu^2 D \leq \widetilde{D}$, the false negative $\mathcal{N}$ can only occur if $|T - \mathbb{E}[T]| > \lambda ND$. Therefore,

$$\mathbf{Pr}\left[\mathcal{N}\right] = \mathbf{Pr}\left[\alpha T > N\widetilde{D} | \mu^2 D \leq \widetilde{D}\right] \leq \mathbf{Pr}\left[|T - \mathbb{E}[T]| > \lambda ND | \mu^2 D \leq \widetilde{D}\right].$$

Then by Hoeffding's inequality, c.f., Theorem B.6,

$$\mathbf{Pr}\left[|T - \mathbb{E}[T]| > \lambda ND | \mu^2 D \leq \widetilde{D}\right] \leq 2\exp\left(-\frac{(\lambda ND)^2}{\sum_{i=1}^{N}(\lambda D)^2}\right) = 2\exp\left(-N\right) \leq \delta,$$

because we have $N_i \in [0, \lambda D]$ and $N = \mathcal{O}\left(\log \frac{1}{\delta}\right)$. Since we have shown the probability of the event $\mathcal{N}$ of getting a false negative is no more than $\delta$, then L1SAMPLING($\{\widetilde{D_j}\}, \{D_j\}, \mu, \delta$) will return True when $\mu^2 D \leq \widetilde{D}$ with probability at least $1 - \delta$.

We next analyze the case $\mu D > \widetilde{D}$.

**Case 2:** $\mu D > \widetilde{D}$. We define the event $\mathcal{P}$ that L1SAMPLING($\{\widetilde{D_j}\}, \{D_j\}, \mu, \delta$) returns True when $\mu D > \widetilde{D}$, i.e., a false positive event. Observe that this can happen if and only if $\alpha T \leq N\widetilde{D}$ when $\mu D > \widetilde{D}$.

Notice that $\alpha T \leq N\widetilde{D}$ is impossible if $|T - \mathbb{E}[T]| \leq \frac{1}{2}ND$. In fact, if $\mu D > \widetilde{D}$ and $|T - \mathbb{E}[T]| \leq \frac{1}{2}ND$, then

$$\alpha T \geq \alpha(\mathbb{E}[T] - |T - \mathbb{E}[T]|) \geq \alpha(ND - \frac{1}{2}ND) = \frac{\alpha}{2}ND > \frac{\alpha}{2\mu}N\widetilde{D} = N\widetilde{D}.$$

The first inequality is due to the triangle inequality. The second inequality comes from the fact that $\mathbb{E}[T] \in [ND, N\lambda D]$ and $|T - \mathbb{E}[T]| \leq \frac{1}{2}ND$. The third inequality is due to the condition $\mu D > \widetilde{D}$. The second equality is because of $\alpha = 2\mu$.

Hence under the condition that $\mu D > \widetilde{D}$, a false positive $\mathcal{P}$ can only occur if $|T - \mathbb{E}[T]| > \frac{1}{2}ND$. Therefore,

$$\mathbf{Pr}\left[\mathcal{P}\right] = \mathbf{Pr}\left[\alpha T \leq N\widetilde{D} | \mu D > \widetilde{D}\right] \leq \mathbf{Pr}\left[|T - \mathbb{E}[T]| > \frac{1}{2}ND | \mu D > \widetilde{D}\right].$$

Then by Hoeffding's inequality, c.f., Theorem B.6,

$$\mathbf{Pr}\left[|T - \mathbb{E}[T]| > \frac{1}{2}ND | \mu D > \widetilde{D}\right] \leq 2\exp\left(-\frac{(\frac{1}{2}ND)^2}{\sum_{i=1}^{N}(\lambda D)^2}\right) = 2\exp\left(-\frac{N}{4\lambda^2}\right) \leq \delta,$$

because $N_i \in [0, \lambda D]$ and $N = \mathcal{O}\left(\log \frac{1}{\delta}\right)$. Since we have shown the probability of the event $\mathcal{P}$ is no more than $\delta$, then it follows that $\mathrm{L1SAMPLING}(\{\widetilde{D_j}\}, \{D_j\}, \mu, \delta)$ will return False if $\mu D > \widetilde{D}$ with probability at least $1 - \delta$. $\qquad\square$

### D.3 BICRITERIA APPROXIMATION WITH COMMUNICATION/ROUND REDUCTION

In this section, we will introduce another protocol that uses $\mathcal{O}\left(\log n \log k\right)$ rounds of communication and $\tilde{\mathcal{O}}\left(s \log n + kd \log n\right)$ bits of communication cost. The protocol will use fewer rounds of communication and total communication than Algorithm 8 when $k = \Omega(\log n)$.

We first apply the $L_1$ sampling subroutine L1SAMPLING to estimate $D$, which uses a low communication cost. We then update $\widetilde{D}$ if the estimation of $D$ from the $L_1$ sampling procedure has decreased significantly. Using such a strategy, we only need to update $\widetilde{D_j}$ in very few rounds, saving the communication cost of updating $\widetilde{D_j}$.

However, the communication rounds are still $\mathcal{O}\left(k\right)$, which would be too expensive in some settings. To further decrease the rounds of communication, we use the batch sampling strategy. We double the number of points to be sampled in every round until the total cost $D$ drops significantly. $D$ can only decrease significantly for at most $\mathcal{O}\left(\log n\right)$ rounds, so we can reduce the rounds of communication to $\mathcal{O}\left(\log n \log k\right)$.

Another issue that must be addressed is that we may sample some invalid points because the total cost $D$ may decrease significantly during a round of batch sampling. A sampled point is valid only if it is sampled under the condition that $\widetilde{D}$ is a $\mathcal{O}\left(1\right)$-approximation of $D$, so some sampled points may be invalid if $D$ drops significantly during that sampling round. To ensure that we sample enough valid points, we need to count the number of valid samples during the sampling procedure. Rather than setting a fixed number $N = \mathcal{O}\left(k\right)$ of points to sample, we count the number of valid samples and only terminate sampling after we get at least $N$ valid samples. Fortunately, although we may sample more than $N$ points in such a strategy, we can prove that the total number of points we sample is still $\mathcal{O}\left(k\right)$.

The algorithm appears in full in Algorithm 10.

Algorithm 10 uses L1SAMPLING to detect whether $D$ decreases significantly and needs to be updated. Under the condition that $\widetilde{D}$ is a $\mathcal{O}\left(1\right)$-approximation of $D$, it uses batch sampling to sample points, and verify whether $\widetilde{D}$ is still a $\mathcal{O}\left(1\right)$-approximation of $D$ after batch sampling. If $\widetilde{D}$ is still a $\mathcal{O}\left(1\right)$-approximation of $D$, it means that all the sampled points are valid, and we count them. If $\widetilde{D}$ is no longer a $\mathcal{O}\left(1\right)$-approximation of $D$, it means the $\mathcal{O}\left(1\right)$-approximation condition breaks during the batch sampling, which leads some sampled points invalid. However, the first sampled point must be valid, since we sample it under the condition that $\widetilde{D}$ is an $\mathcal{O}\left(1\right)$-approximation. Hence, we can add 1 to our valid sampled points count. We repeat the sampling until we have counted at least $N = \mathcal{O}\left(k\right)$ valid sampled points. Then the returned set $S$ will have $\mathcal{O}\left(k\right)$ points and is an $\mathcal{O}\left(1\right)$-approximation. Furthermore, by using L1SAMPLING and batch sampling, we can guarantee a low round complexity and total communication for Algorithm 10.

---

**Algorithm 10** Bicriteria approximation algorithm for $(k, z)$-clustering

---

**Input:** Dataset $X_i$ for each site $i \in [s]$
**Output:** A set $S$ that is an $(\mathcal{O}(1), \mathcal{O}(1))$-bicriteria approximation for the $(k, z)$-clustering
1: Uniformly sample a point into $S$ and compute constant-factor approximations $\{\widetilde{D_j}\}$ to $D_j = \text{cost}(X_j, S)$
2: $N \leftarrow \mathcal{O}(k), M \leftarrow 0, \mu \leftarrow 8, \delta \leftarrow \mathcal{O}\left(\frac{1}{\log^2 n}\right)$
3: **while** $M < N$ \\ Sample roughly $k$ points **do**
4:  $bool \leftarrow \text{False}, i \leftarrow 1$
5:  **while** $bool = \text{False}$ \\ Approximate distances $\{\widetilde{D_j}\}$ are accurate **do**
6:   **for** $l \leftarrow 1$ to $2^i$ \\ Attempt to sample points into $S$ **do**
7:    $x_l \leftarrow \text{LAZYSAMPLING}(\{\text{cost}(x, S)\}, \{\widetilde{D_j}\})$, upload $x_l$ on blackboard
8:   **end for**
9:   $S \leftarrow S \cup (\cup_{l=1}^{2^i} \{x_l\}), L \leftarrow \#\{x_l \neq \bot, 1 \leq l \leq 2^i\}$      ▷Add successful samples to $S$
10:   $M \leftarrow M + L$
11:   Every site $j$ computes $D_j = \text{cost}(X_j, S)$       ▷Update distances to closest center
12:   $bool \leftarrow \text{L1SAMPLING}(\{\widetilde{D_j}\}, \{D_j\}, \mu, \delta)$      ▷Check approximate distances $\{\widetilde{D_j}\}$
13:   **if** $bool = \text{False}$ **then**
14:    $i \leftarrow i + 1$       ▷More aggressive number of samples to reduce number of rounds
15:   **else**
16:    $r_j \leftarrow \text{POWERAPPROX}(D_j, 2)$, every site $j$ update $r_j$ on the blackboard if $r_j \neq \log_2 \widetilde{D_j}$
17:    $\widetilde{D_j} \leftarrow 2^{r_j}$       ▷All sites update approximate distances
18:   **end if**
19:  **end while**
20: **end while**
21: **return** $S$

---

We split the proof into three parts. First we will prove that $S$ is an $\mathcal{O}(1)$-approximation and $|S| = \mathcal{O}(k)$. Secondly, we will prove that the algorithm uses $\mathcal{O}(\log n \log k)$ rounds of communication. Lastly, we will prove that the algorithm uses $\tilde{\mathcal{O}}(s \log n + kd \log n)$ bits of communication with probability at least $0.99$.

For the purposes of analysis, we define the concept of valid sample. Suppose $x$ is a point sampled by $\text{LAZYSAMPLING}(\{\text{cost}(x, S)\}, \{\widetilde{D_i}\})$. Then we define $x$ to be a valid sample if it is sampled under the condition $D \leq \widetilde{D} < \gamma \cdot D_i$ for some constant $\gamma$.

**Lemma D.3.** *Algorithm 10 will return $S$ such that $|S| = \mathcal{O}(k)$ and $\text{cost}(S, X) \leq \mathcal{O}(1) \cdot \text{cost}(C_{OPT}, X)$ with probability at least $0.98$, where $C_{OPT}$ is the optimal $(k, z)$-clustering of $X$.*

*Proof.* Suppose $x$ is a point sampled by $\text{LAZYSAMPLING}(\{\text{cost}(x, S)\}, \{\widetilde{D_i}\})$. If $x$ is a valid sample, it will reduce the size of **Bad**$_i$ with constant probability by Lemma C.2. If we sample $N = \mathcal{O}(k)$ valid points by our algorithm, by Theorem B.10, we will get a constant approximation $S$ with probability at least $0.99$. We will show that our algorithm indeed samples at least $N$ valid samples for $\gamma = \mu^2$.

According to the algorithm, $\widetilde{D_j}$ is always a 2-approximation of $D_j$ that $D_j \leq \widetilde{D_j} < 2D_j$ after we update $\widetilde{D_j}$. Between two updates of $\widetilde{D_j}$, $\widetilde{D_j}$ is fixed, but $D_j$ may only decrease. This means that $D_j \leq \widetilde{D_j}$ always holds in Algorithm 10. Since $D_j \leq \widetilde{D_j}$ always holds, $D \leq \widetilde{D}$ always holds, too.

If $\widetilde{D} < \mu^2 D$ after we sample $L$ points, it means that all these $L$ points are valid. We will either add $L$ to the counter $M$ if L1SAMPLING returns False, or add 1 to the counter $M$ if L1SAMPLING returns True. In either case, the number of valid points we sample is at least the number we add to the counter $M$.

If $\widetilde{D} \geq \mu^2 D$ after we sample $L$ points, it means that some points we sample are invalid. However, the first point we sampled must be valid, as it is sampled under the condition $\widetilde{D} < \mu^2 D$. If

L1SAMPLING returns True, we will only add 1 to the counter $M$. Then the number of new valid samples in this round is at least the number we add to the counter $M$.

Therefore, the number of valid points we sample is at least the number we add to the counter $M$ if L1SAMPLING returns True every time $\widetilde{D} \geq \mu^2 D$. Since $D = \mathrm{poly}(n)$ when we have only one point in $S$, and every time we update $\widetilde{D}$ in Algorithm 10, it holds that $D \leq \widetilde{D} < 2D$, $\widetilde{D} \geq \mu^2 D$ can only occur at most $\mathcal{O}(\log n)$ times. We know L1SAMPLING will return True if $\widetilde{D} \geq \mu^2 D$ with probability at least $1 - \delta$. Since $\delta = \mathcal{O}\left(\frac{1}{\log^2 n}\right)$, then by a union bound, L1SAMPLING returns True every time $\widetilde{D} \geq \mu^2 D$ with probability at least 0.99.

Therefore, with probability at least 0.98, we will sample at least $\mathcal{O}(k)$ valid points and they form an $\mathcal{O}(1)$-approximation for the optimal solution.

**Upper bound on the size of $S$.** We will evaluate the number of sampled points that are not counted as valid samples, and show the total number of such points is $\mathcal{O}(k)$. Suppose that we update $\widetilde{D_j}$ for $m$ times for total.

Let $N_p$ be the total number of sampled points that are not counted as valid samples, and $p_i$ as the number of points that are not counted as valid samples between the $(i-1)$-th and $i$-th update. Then $N_p = \sum_{i=1}^{m} p_i$. Let $N_q$ be the total number of sampled points that are counted as valid samples, and $q_i$ as the number of points that are counted as valid samples between the $(i-1)$-th and $i$-th update. Then $N_q = \sum_{i=1}^{m} q_i$. Since we will terminate the algorithm if we count for $N = \mathcal{O}(k)$ valid samples, $N_q = \mathcal{O}(k)$. Since the number of points sampled before the $i$-th update is $p_i + 1$, and the number of point sampled before these $p_i + 1$ points is $\frac{p_i+1}{2}$, therefore, $p_i \leq 2q_i$. Hence $N_p \leq 2N_q = \mathcal{O}(k)$. The points in $S$ are either counted as valid samples or not, so $|S| = N_p + N_q = \mathcal{O}(k)$.

$\square$

We next upper bound the round complexity of our algorithm.

**Lemma D.4.** *Algorithm 10 uses $\mathcal{O}(\log n \log k)$ rounds of communication with probability at least* 0.99.

*Proof.* We need $\mathcal{O}(1)$ communication rounds in the initial stage Algorithm 7. For the communication rounds in the remaining part of Algorithm 10, we evaluate how many rounds of batch sampling occurs between two update, and evaluate the times of update. Then the product is just the rounds of total sampling.

For the rounds of batch sampling that occurs between two update, it must be at most $\mathcal{O}(\log k)$ rounds since we double the number of points to be sampled for every other round if we do not update $\widetilde{D}$ and the total points to be sampled is $\mathcal{O}(k)$. For the times we update $\widetilde{D}$, there are two cases: we update an 'unnecessary update' under the condition that $\widetilde{D} < \mu D$, and we update a 'necessary update' under the condition that $\widetilde{D} \geq \mu D$. By Lemma D.2, L1SAMPLING will return True and we will have an 'unnecessary update' under the condition that $\widetilde{D} < \mu D$ with probability at most $\delta$. Since there are at most $\mathcal{O}(\log k)$ successive batch sampling without an update, we will make an 'unnecessary update' between two 'necessary update' with probability at most $\mathcal{O}(\delta \log k)$. For the 'necessary update', we will update $\widetilde{D}$ so that $D \leq \widetilde{D} < 2D$. Since $\mu = 8$, it means every time we make a 'necessary update' under the condition that $\widetilde{D} \geq \mu D$, $\widetilde{D}$ decreases by at least a factor of 4. Hence, such update can occur at most $\mathcal{O}(\log n)$ times.

Since the probability that we will make an 'unnecessary update' between two 'necessary updates' is at most $\mathcal{O}(\delta \log k)$, and the times of 'necessary update' is at most $\mathcal{O}(\log n)$, we will make an 'unnecessary update' in Algorithm 10 with probability at most $\mathcal{O}(\delta \log n \log k)$. Since $\delta = \mathcal{O}\left(\frac{1}{\log^2 n}\right)$, we will have no 'unnecessary update' in Algorithm 10 with probability at least 0.99. Since 'necessary update' will occur at most $\mathcal{O}(\log n)$ times, it means we will have at most $\mathcal{O}(\log n)$ updates of $\widetilde{D}$ with probability at least 0.99.

Therefore, with probability at least 0.99, Algorithm 10 has $\mathcal{O}\left(\log n \log k\right)$ rounds of batch sampling. Then we need to evaluate how many communication rounds we need for a round of batch sampling. For each round of batch sampling, the coordinator can send the request of $2^i$ lazy sampling at the same time, and every site can respond afterwards. Hence there will be two rounds of communication for each iteration of sampling. To get the result returned by L1SAMPLING, the $\mathcal{O}\left(\log \frac{1}{\delta}\right) = \mathcal{O}\left(\log \log n\right)$ requests can be made simultaneously, so that there are $\mathcal{O}\left(1\right)$ rounds of communication. Hence, the total rounds of communication in Algorithm 10 is $\mathcal{O}\left(\log n \log k\right)$.

Finally, we upload $r_j$ when we update $\widetilde{D_j}$. All the sites can update the values of $r_j$ in the same round. Since we will update at most $\mathcal{O}\left(\log n\right)$ times, we needs $\mathcal{O}\left(\log n\right)$ rounds of communication for total. Summing the rounds of communication across each part of our algorithm, it follows that the total rounds of communication is $\mathcal{O}\left(\log n \log k\right)$ with probability at least 0.99. $\qquad\square$

Finally, we analyze the total communication of our algorithm.

**Lemma D.5.** *Algorithm 10 uses $\tilde{\mathcal{O}}\left(s \log n + kd \log n\right)$ bits of communication with probability at least 0.99.*

*Proof.* The Algorithm 7 subroutine INITIALIZATION induces $\mathcal{O}\left(d \log n + s \log \log n\right)$ bits of communication because we only write the sample point $x$ and $r_i$ on the blackboard, and $r_i = \mathcal{O}\left(\log \log \mathrm{cost}(X_i, S)\right) = \mathcal{O}\left(\log \log n\right)$.

We will update $\mathcal{O}\left(k\right)$ samples, which will use $\mathcal{O}\left(kd \log n\right)$ total bits of communication.

Moreover, we will run L1SAMPLING at most $\mathcal{O}\left(\log n \log k\right)$ times. For each time it runs, the coordinator will choose $\mathcal{O}\left(\log \frac{1}{\delta}\right) = \mathcal{O}\left(\log \log n\right)$ sites. It uses $\mathcal{O}\left(\log s\right)$ bits to represent a site on the blackboard, so that site knows it is chosen. Each chosen site needs $\mathcal{O}\left(\log \log n\right)$ bits to reply. Hence the total communication cost for running L1SAMPLING in Algorithm 10 is $\mathcal{O}\left(\log n \log k(\log s + \log \log n)\right)$ bits.

Updating the values $\{r_j\}$ costs $\mathcal{O}\left(s \log \log n\right)$ bits for each iteration. Since the value will be updated $\mathcal{O}\left(\log n\right)$ times with probability at least 0.99, the total cost for updating the values $\{r_j\}$ is $\mathcal{O}\left(s \log n \log \log n\right)$ bits.

Therefore, by adding the communication cost for all parts of Algorithm 10 and ignore all the polylogarithm terms for $s, k, \log n$, the total communication is just $\tilde{\mathcal{O}}\left(s \log n + kd \log n\right)$. $\qquad\square$

We have the following full guarantees for Algorithm 10.

**Lemma D.6.** *Algorithm 10 will return $S$ such that $|S| = \mathcal{O}\left(k\right)$ and $\mathrm{cost}(S, X) \leq \mathcal{O}\left(1\right) \cdot \mathrm{cost}(C_{OPT}, X)$ with probability at least 0.98, where $C_{OPT}$ is the optimal $(k, z)$-clustering of $X$. The algorithm uses $\tilde{\mathcal{O}}\left(s \log n + kd \log n\right)$ bits of communication and $\mathcal{O}\left(\log n \log k\right)$ rounds of communication with probability at least 0.99.*

*Proof.* The proof follows immediately from Lemma D.3, Lemma D.4, and Lemma D.5. $\qquad\square$

### D.4 $(1 + \varepsilon)$-CORESET VIA SENSITIVITY SAMPLING IN THE BLACKBOARD MODEL

To achieve a $(1 + \varepsilon)$-coreset for $X$ with the $(\mathcal{O}\left(1\right), \mathcal{O}\left(1\right))$-bicriteria approximation $S$, we use sensitivity sampling. The challenge lies in determining $C_j$ and $\mathrm{cost}(C_j, S)$ to calculate $\mu(x)$, as communicating the exact values of $C_j$ and $\mathrm{cost}(C_j, S)$ requires $\mathcal{O}\left(sk \log n\right)$ bits, which is impractical for our purposes. However, a constant approximation of $C_j$ and $\mathrm{cost}(C_j, S)$ suffices to approximate $\mu(x)$ for the sensitivity sampling, and this can be achieved efficiently with minimal communication through Morris counters.

**Theorem D.7.** *(Morris, 1978) Let $X = \mathrm{MORRIS}(0, n)$. Then $\mathbb{E}\left[2^X - 1\right] = n$. Moreover, there exists a constant $\gamma > 0$ such that if $l \geq \frac{\gamma}{\varepsilon^2} \log \frac{1}{\delta}$ and $Y = \frac{1}{l} \sum_{i=1}^{l} \left(2^{X_i} - 1\right)$, where $X_1, X_2, \ldots, X_l$ are $l$ independent outputs of $\mathrm{MORRIS}(0, n)$, then*

$$Y \in [n - \varepsilon n, n + \varepsilon n],$$

*with probability at least $1 - \delta$.*

---

**Algorithm 11** MORRIS$(r, n)$

---

**Input:** Initial count index $r$, and elements number $n$
**Output:** New count index $m$
 1: $m \leftarrow r$
 2: **for** $i \leftarrow 1$ to $n$ **do**
 3:     With probability $\frac{1}{2^m}$, $m \leftarrow m + 1$
 4: **end for**
 5: **return** $r$

---

We can adapt MORRIS to a distributed version DISTMORRIS to reduce the communication cost.

---

**Algorithm 12** DISTMORRIS$(\varepsilon)$:

---

**Input:** Precision parameter $\varepsilon$, every site $S_i$ owns $k$ numbers $n_{i,j}$, $j \in [k]$
**Output:** Approximation $(\widetilde{N}_1, \widetilde{N}_2, \cdots, \widetilde{N}_k)$ for sums $N_j = \sum_{i=1}^{s} n_{i,j}$, $j \in [k]$
 1: $l \leftarrow \mathcal{O}\left(\frac{1}{\varepsilon^2} \log(100k)\right)$
 2: $m_{j,t} \leftarrow 0$ for all $j \in [k]$ and $t \in [l]$
 3: **for** $i \leftarrow 1$ to $s$ **do**
 4:     $m'_{j,t} \leftarrow$ MORRIS$(m_{j,t}, n_{i,j})$ for all $j \in [k]$ and $t \in [l]$
 5:     **if** $m'_{j,t} = m_{j,t}$ for all $j \in [k]$ and $t \in [l]$ **then**
 6:         Site $i$ uploads $\perp$ on blackboard
 7:     **else**
 8:         Site $i$ uploads $(m'_{j,t} - m_{j,t}, j, t)$ for all $j \in [k]$ and $t \in [l]$ that $m'_{j,t} \neq m_{j,t}$
 9:     **end if**
10:     $m_{j,t} \leftarrow m'_j$
11: **end for**
12: $\widetilde{N}_j \leftarrow \frac{1}{l} \sum_{t=1}^{l} \left(2^{m_{j,t}} - 1\right)$ for all $j \in [k]$
13: **return** $(\widetilde{N}_1, \widetilde{N}_2, \cdots, \widetilde{N}_k)$

---

**Lemma D.8.** *Let every site $i$ own $k$ numbers $n_{i,j}$, and $|N_j| = \sum_{i=1}^{s} n_{i,j} = \mathrm{poly}(n)$. With probability at least* 0.99, *DISTMORRIS$(\frac{1}{4})$ returns constant approximations $\widetilde{N}_j$ that $\widetilde{N}_j \in [\frac{3}{4}N_j, \frac{5}{4}N_j]$ for all $j \in [k]$ using $\tilde{\mathcal{O}}\left(s + k \log n\right)$ bits of communication.*

*Proof.* Although we split the process of MORRIS into a distributed version, its accuracy does not matter. This is because every step of MORRIS only needs to know the status of the previous step, and every time a site uses a step of MORRIS, it knows the status of the previous step either because both of these steps occur at this site, or because the status of previous step is accurately uploaded by the previous site. Hence, the statement of the original MORRIS in Theorem D.7 still holds for our distributed version.

By Theorem D.7, $\widetilde{N}_j \in [\frac{3}{4}N_j, \frac{5}{4}N_j]$ with probability at least $1 - \frac{1}{100k}$ for any $j \in [k]$ if we set $l = \mathcal{O}\left(\log(100k)\right)$. Then, by a union bound, $\widetilde{N}_j \in [\frac{3}{4}N_j, \frac{5}{4}N_j]$ for all $j \in [k]$ at the same time with probability at least 0.99.

To evaluate the total communication cost, we introduce $M_i$ and $M_{i,j}$ as follows to facilitate the analysis. We set $M_i$ as 0 if site $i$ uploads $\perp$ on the blackboard and $M_i$ as the cost of communication to update all $(m'_{j,t} - m_{j,t}, j, t)$ for all $j \in [k]$ and $t \in [l]$ that $m'_{j,t} \neq m_{j,t}$ otherwise. We set $M_{i,j,t}$ to be the communication cost required to update $(m'_{j,t} - m_{j,t}, j, t)$.

Every site with $M_i = 0$ only needs $\mathcal{O}\left(1\right)$ bits to upload $\perp$. Since there are $s$ sites in total, the number of sites with $M_i = 0$ is at most $s$. Therefore, the total communication for these sites is $\mathcal{O}\left(s\right)$.

For the sites with $M_i \neq 0$, the communication cost used is

$$\sum_{M_i \neq 0} M_i = \sum_{i=1}^{s} M_i = \sum_{j=1}^{k} \sum_{t=1}^{l} \sum_{i=1}^{s} M_{i,j,t}.$$

Note that the first equality holds because we only add more terms that all have value zero, whereas the second equality is obtained by dividing $M_i$ into $M_{i,j,t}$.

Since $\widetilde{N_j} = \frac{1}{l} \sum_{t=1}^{l} (2^{m_{j,t}} - 1) \in [\frac{3}{4} N_j, \frac{5}{4} N_j]$, then $2^{m_{j,t}}$ cannot be greater than $2l \cdot N_j = \mathcal{O}(\log k \operatorname{poly}(n))$. Therefore $m_{j,t} \leq \mathcal{O}(\log n \log \log k)$. Since $M_{i,j,t}$ must be at least $1$ if it is nonzero, there are at most $m_{j,t}$ nonzero $M_{i,j,t}$ for given $j, t$. For every non-zero $M_{i,j,t}$, since the site uploads $m'_{j,t} - m_{j,t}$ which is at most the final $m_{j,t}$, then the site will use no more than $\mathcal{O}(\log m_{j,t})$ bits to update $m'_{j,t} - m_{j,t}$, and $\mathcal{O}(\log k + \log l)$ bits to express $j$ and $t$. Thus,

$$\sum_{i=1} M_{i,j,t} = \sum_{i \in [s], M_{i,j,t} \neq 0} M_{i,j,t} \leq m_{j,t} \cdot \mathcal{O}(\log m_{j,t} + \log k + \log l).$$

Since $m_{j,t} \leq \mathcal{O}(\log n \log \log k)$ and $l = \mathcal{O}(\log k)$, then $\sum_{i=1} M_{i,j,t} \leq \tilde{\mathcal{O}}(\log n)$. Therefore,

$$\sum_{M_i \neq 0} M_i = \sum_{j=1}^{k} \sum_{t=1}^{l} \sum_{i=1} M_{i,j,t} \leq kl \cdot \tilde{\mathcal{O}}(\log n) = \tilde{\mathcal{O}}(k \log n).$$

By summing the communication cost of the sites that upload $\perp$ and the communication cost of the other sites, the total communication will be no more than $\tilde{\mathcal{O}}(s + k \log n)$. $\qquad \square$

Since $|C_j| = \sum_{i=1}^{s} |C_j \cap X_i|$ and $\operatorname{cost}(C_j, S) = \sum_{i=1}^{s} \operatorname{cost}(C_j \cap X_i, S)$ for any $j \in [k]$, we can use DISTMORRIS to approximately evaluate these terms using low communication cost. Since both $|C_j|$ and $\operatorname{cost}(C_j, S)$ are at most $\operatorname{poly}(n)$, the total communication is at most $\tilde{\mathcal{O}}(s + k \log n)$ bits. Every site $i$ can compute $\operatorname{cost}(x, S)$ for $x \in X_i$ locally. Then with the constant-factor approximations for $|C_j|$ and $\operatorname{cost}(C_j, S)$, each site can evaluate a constant-factor approximation of the sensitivity $\mu(x)$ locally.

Finally, we give an algorithm producing a $(1+\varepsilon)$-coreset for $X$ with $\mathcal{O}\left(\frac{dk}{\varepsilon^4}(\log k + \log \frac{1}{\varepsilon} \log \log n)\right)$ bits of communication, given our $\mathcal{O}(1)$-approximation with $\mathcal{O}(k)$ points and sensitivity $\mu(x)$.

---

**Algorithm 13** $(1 + \varepsilon)$-coreset for the blackboard model

---

**Input:** A constant approximation $S$, $|S| = \mathcal{O}(k)$
**Output:** A $(1 + \varepsilon)$-coreset $A$
1: Use DISTMORRIS to get $\mathcal{O}(1)$-approximation for $|C_j|$ and $\operatorname{cost}(C_j, S)$ for all $j \in [k]$ on the blackboard
2: $m \leftarrow \tilde{\mathcal{O}}\left(\frac{k}{\varepsilon^2} \min\{\varepsilon^{-2}, \varepsilon^{-z}\}\right)$
3: **for** $i \leftarrow 1$ to $s$ **do**
4: $\quad A_i \leftarrow \emptyset$
5: $\quad$ Compute $\widetilde{\mu}(x)$ as an $\mathcal{O}(1)$-approximation of $\mu(x)$ locally for all $x \in X_i$
6: $\quad$ Upload $\widetilde{\mu}(X_i) = \sum_{x \in X_i} \widetilde{\mu}(x)$
7: **end for**
8: Samples site $i$ with probability $\frac{\widetilde{\mu}(X_i)}{\sum_{i=1}^{s} \widetilde{\mu}(X_i)}$ independently for $m$ times. Let $m_i$ be the time site $i$ are sampled. Write $m_i$ on blackboard
9: **for** $i \leftarrow 1$ to $s$ **do**
10: $\quad A_i \leftarrow \emptyset$
11: $\quad$ **for** $j \in [m_i]$ **do**
12: $\quad\quad$ Sample $x$ with probability $p_x = \frac{\widetilde{\mu}(x)}{\widetilde{\mu}(X_i)}$
13: $\quad\quad$ **if** $x$ is sampled **then**
14: $\quad\quad\quad$ Let $x'$ be $x$ efficiently encoded by $S$ and accuracy $\varepsilon' = \operatorname{poly}(\varepsilon)$
15: $\quad\quad\quad A \leftarrow A_i \cup \{(x', \frac{1}{m\widehat{\mu}(x)})\}$, where $\widehat{\mu}(x)$ is a $(1 + \frac{\varepsilon}{2})$-approximation of $\widetilde{\mu}(x)$
16: $\quad\quad$ **end if**
17: $\quad$ **end for**
18: $\quad$ Upload $A_i$ to the blackboard
19: **end for**
20: $A \leftarrow \cup_{i=1}^{s} A_i$
21: **return** $A$

---

We now justify the correctness and complexity of the communication of Algorithm 13.

**Lemma D.9.** *Algorithm 13 returns $A$ such that $A$ is a $(1 + \varepsilon)$-coreset for $X$ with probability at least $0.98$ if we already have an $(\mathcal{O}(1), \mathcal{O}(1))$-bicriteria approximation $S$. The algorithm uses $\tilde{\mathcal{O}}\left(s + k \log n + \frac{dk}{\min(\varepsilon^4, \varepsilon^{2+z})}\right)$ bits of communication.*

*Proof.* By Lemma D.8, we can get an $\mathcal{O}(1)$-approximation for $|C_j|$ and $\text{cost}(C_j, S)$ with probability at least $0.99$. Every site can also get the precise value of $\text{cost}(x, S)$ locally. Since $\mu(x) := \frac{1}{4} \cdot \left(\frac{1}{k|C_j|} + \frac{\text{cost}(x,S)}{k \text{cost}(Cj,S)} + \frac{\text{cost}(x,S)}{\text{cost}(X,S)} + \frac{\Delta_x}{\text{cost}(X,S)}\right)$ and $\Delta_p := \text{cost}(C_j, S)/|C_j|$, we can evaluate an $\mathcal{O}(1)$-approximation $\widetilde{\mu}(x)$ of $\mu(x)$. Although Bansal et al. (2024) proves Theorem B.7 in the setting that $|S| = k$ and sample points with probability $\mu(x)$, they only need $|S| = \mathcal{O}(k)$ and $\frac{1}{\mu(x)} \leq \mathcal{O}(1) \cdot \max\{\frac{1}{k|C_j|}, \frac{\text{cost}(x,S)}{k \text{cost}(Cj,S)}, \frac{\text{cost}(x,S)}{\text{cost}(X,S)}, \frac{\Delta_x}{\text{cost}(X,S)}\}$ in their proof. Therefore, Theorem B.7 is still valid under the condition that $|S| = \mathcal{O}(k)$ and $\widetilde{\mu}(x) = \mathcal{O}(1) \cdot \mu(x)$. Let $B = \{(x, \frac{1}{m\widetilde{\mu}(x)}))\}$, then $B$ is an $(1 + \frac{\varepsilon}{2})$-coreset for $X$ with probability at least $0.99$ if we have $\mathcal{O}(1)$-approximation for $|C_j|$ and $\text{cost}(C_j, S)$.

We have $A = \{(x', \frac{1}{m\widehat{\mu}(x)})\}$. Since $B$ is a $(1 + \frac{\varepsilon}{2})$-coreset for $X$, by Lemma B.13, $A$ is a $(1 + \varepsilon)$-coreset for $X$. Therefore, Algorithm 13 returns an $(1 + \varepsilon)$-coreset for $X$ with probability at least $0.98$.

We need $\mathcal{O}(s + k \log n)$ bits to get the $\mathcal{O}(1)$-approximation for $|C_j|$ and $\text{cost}(C_j, S)$. We need $\mathcal{O}(s \log m) = \tilde{\mathcal{O}}(s)$ bits to write $m_i$ on the blackboard. We need $\mathcal{O}(\log k + d \log \log n)$ bits to upload $x'$, because we need to point out which center in $S$ is closest to $x$, which needs $\mathcal{O}(\log k)$ bits, and we need to upload the power index, which needs $\mathcal{O}(\log \frac{1}{\varepsilon} \log \log n)$ bits. We need $\mathcal{O}(\frac{1}{\varepsilon} \log \log n)$ bits to upload $\frac{1}{m\widehat{\mu}(x)}$ because we only need to upload $\widehat{\mu}(x)$, which is a $(1 + \frac{\varepsilon}{2})$-approximation of $\widetilde{\mu}(x)$. Since $|A| = \mathcal{O}\left(\frac{k}{\min(\varepsilon^4, \varepsilon^{2+z})}\right)$, we need $\tilde{\mathcal{O}}\left(s + k \log n + \frac{dk}{\min(\varepsilon^4, \varepsilon^{2+z})}\right)$ bits in total. $\square$

We complete this subsection by claiming that we can get a $(1+\varepsilon)$-strong coreset with communication cost no more than $\tilde{\mathcal{O}}\left(s \log(n\Delta) + dk \log(n\Delta) + \frac{dk}{\min(\varepsilon^4, \varepsilon^{2+z})}\right)$ bits.

**Theorem D.10.** *There exists a protocol on $n$ points distributed across $s$ sites that produces a $(1+\varepsilon)$-strong coreset for $(k, z)$-clustering with probability at least $0.97$ that uses*

$$\tilde{\mathcal{O}}\left(s \log(n) + dk \log(n) + \frac{dk}{\min(\varepsilon^4, \varepsilon^{2+z})}\right)$$

*total bits of communication in the blackboard model.*

*Proof.* We can use Algorithm 10 to generate an $(\mathcal{O}(1), \mathcal{O}(1))$-bicriteria approximation $S$, and then use Algorithm 13 to get an $(1 + \varepsilon)$-strong coreset. By Lemma D.6 and Lemma D.9, the returned set $A$ is a $(1 + \varepsilon)$-strong coreset with probability at least $0.97$ and uses a communication cost of $\tilde{\mathcal{O}}\left(s \log(n) + dk \log(n) + \frac{dk}{\min(\varepsilon^4, \varepsilon^{2+z})}\right)$ bits for total. $\square$

# E  COORDINATOR MODEL OF COMMUNICATION

In this section, we discuss our distributed algorithms for the coordinator model of communication. Note that if all servers implement the distributed algorithms from the blackboard setting, the resulting communication would have $\mathcal{O}(dsk \log n)$ terms simply from the $\mathcal{O}(k)$ rounds of adaptive sampling across the $s$ servers.

## E.1  EFFICIENT SAMPLING IN THE COORDINATOR MODEL

To avoid these $\mathcal{O}(dsk \log n)$ terms in communication cost, we need a more communication efficient method so that we can apply adaptive sampling and upload coreset with low communication cost. We propose an algorithm called EFFICIENTCOMMUNICATION, which can send the location of a point with high accuracy and low cost. Suppose that a fixed site has a set of points

$X = \{x_1, \cdots, x_l\}$, and another site has a point $y$. To simulate adaptive sampling while avoiding sending each point explicitly to all sites, our goal is to send a highly accurate approximate location of $y$ to the first site with low cost. To that end, EFFICIENTCOMMUNICATION approximates and sends the coordinate of a point in every dimension. For the $i$-th dimension, we use a subroutine HIGHPROBGREATERTHAN, which can detect whether $y^{(i)}$ is greater than $x_j^{(i)}$ with high probability, where $y^{(i)}$ is the coordinate of the $i$-th dimension of $y$ and $x_j^{(i)}$ is the coordinate of the $i$-th dimension of $x_j \in X$.

---

**Algorithm 14** HIGHPROBGREATERTHAN$(x, y, \delta)$

---

**Input:** Integer $x, y$, failure parameter $\delta$
**Output:** A boolean *bool* that shows whether $x$ is greater than $y$
1: $N \leftarrow \mathcal{O}\left(\log \frac{1}{\delta}\right), r \leftarrow 0$
2: **for** $i \leftarrow 1$ to $N$ **do**
3:     $r_i \leftarrow 0$ if GREATERTHAN$(x, y)$ tells us $x \leq y$, and $r_i \leftarrow 1$ otherwise
4:     $r \leftarrow r + r_i$
5: **end for**
6: **if** $\frac{r}{N} \leq \frac{1}{2}$ **then**
7:     **return** False
8: **else**
9:     **return** True
10: **end if**

---

The subroutine HIGHPROBGREATERTHAN is an adaptation of GREATERTHAN by (Nisan, 1993). The main purpose of the subroutine is that by using a binary search, we can find the relative location of $y^{(i)}$ to $\{x_1^{(i)}, \cdots, x_l^{(i)}\}$ within $\log l$ comparisons. Then, comparing $y^{(i)}$ with $x_j^{(i)} + (1 + \varepsilon)^m$, we can find the best $x_j^{(i)}$ and $m$ to approximate $y^{(i)}$. Our algorithm for the coordinator model appears in full in Algorithm 15.

---

**Algorithm 15** EFFICIENTCOMMUNICATION$(X, y, \varepsilon, \delta)$

---

**Input:** A set $X = \{x_1, \cdots, x_l\}$ owned by one site; a point $y$ owned by another site; $\varepsilon$, accuracy parameter; $\delta$, the failure probability
**Output:** The second site sends $\widetilde{y}$ to the first site, which will be an approximate location of $y$ that $\|y - \widetilde{y}\|_2 \leq \varepsilon \|x - y\|_2$ for any $x \in X$, with probability at least $1 - \delta$
1: **for** $i \leftarrow 1$ to d **do**
2:     Sort $X = \{x_{i_1}, \cdots, x_{i_l}\}$ such that $x_{i_1}^{(i)} \leq \cdots \leq x_{i_l}^{(i)}$, where $x_{i_j}^{(i)}$ is the $i$-th coordinate of $x_{i_j}$
3:     $x_{i_0}^{(i)} \leftarrow -\Delta, x_{i_{l+1}}^{(i)} \leftarrow \Delta, \delta' \leftarrow \mathcal{O}\left(\frac{\delta}{d\left(\log l + \log\log n + \log \frac{1}{\varepsilon}\right)}\right)$
4:     Use binary search and HIGHPROBGREATERTHAN$(y^{(i)}, x_{i_j}^{(i)}, \delta')$ to find $x_{i_s}^{(i)}$ that $|x_{i_s}^{(i)} - y^{(i)}| \leq |x_{i_j}^i - y^{(i)}|$ for any $i_j \in \{0, 1, 2, \cdots, l, l+1\}$
5:     Use HIGHPROBGREATERTHAN test whether $y^{(i)} = x_{i_s}^{(i)}$
6:     **if** $y^{(i)} = x_{i_s}^{(i)}$ **then**
7:         $\Delta y^{(i)} \leftarrow 0$
8:     **else**
9:         $\gamma \leftarrow \text{sign}(y^{(i)} - x_{i_s}^{(i)})$
10:         Use binary search and HIGHPROBGREATERTHAN$(y^{(i)}, x_{i_s}^{(i)} + \gamma \cdot (1 + \varepsilon)^t, \delta')$ to find $m$ that $|x_{i_s}^{(i)} + \gamma \cdot (1 + \varepsilon)^m - y^{(i)}| \leq |x_{i_s}^{(i)} + \gamma \cdot (1 + \varepsilon)^t - y^{(i)}|$ for any $t \in \mathbb{N}$
11:         $\Delta y^{(i)} \leftarrow \gamma \cdot (1 + \varepsilon)^m$
12:     **end if**
13:     $\widetilde{y}^{(i)} \leftarrow x_{i_s}^{(i)} + \Delta y^{(i)}$
14: **end forreturn** $\widetilde{y} = (\widetilde{y}^{(1)}, \widetilde{y}^{(2)}, \cdots, \widetilde{y}^{(d)})$

---

We first show correctness of HIGHPROBGREATERTHAN, running multiple times and taking the majority vote if necessary to boost the probability of correctness.

**Lemma E.1.** *If* HIGHPROBGREATERTHAN$(x, y, \delta)$ *returns False, then* $x \leq y$ *with probability at least* $1 - \delta$. *If* HIGHPROBGREATERTHAN$(x, y, \delta)$ *returns True, then* $x > y$ *with probability at least* $1 - \delta$. *Furthermore, the protocol uses* $\mathcal{O}\left(\log \log n \log \frac{1}{\delta}\right)$ *bits of communication provided that* $x, y \in [-\operatorname{poly}(n), \operatorname{poly}(n)]$.

*Proof.* We define random variable $E_i = r_i$. By Theorem B.8, GREATERTHAN will give a wrong answer with probability at most $p < \frac{1}{2}$, so

$$\mathbb{E}[E_i | x \leq y] = 0 \cdot \mathbf{Pr}\left[E_i = 0 | x \leq y\right] + 1 \cdot \mathbf{Pr}\left[E_i = 1 | x \leq y\right] \leq p,$$
$$\mathbb{E}[E_i | x > y] = 0 \cdot \mathbf{Pr}\left[E_i = 0 | x > y\right] + 1 \cdot \mathbf{Pr}\left[E_i = 1 | x > y\right] \geq 1 - p.$$

Since $E_i \in [0, 1]$, by Hoeffding's inequality, c.f., Theorem B.6,

$$\mathbf{Pr}\left[\sum_{i=1}^{N} E_i > \frac{N}{2} \middle| x \leq y\right] = \mathbf{Pr}\left[\sum_{i=1}^{N} E_i > Np + \frac{N}{2} - Np \middle| x \leq y\right]$$

$$\leq \mathbf{Pr}\left[\left|\sum_{i=1}^{N} E_i - N\mathbb{E}[E_i]\right| > \frac{N}{2} - Np \middle| x \leq y\right]$$

$$\leq 2\exp\left(-\frac{\left(\frac{N}{2} - Np\right)^2}{N \cdot 1^2}\right)$$

$$= \delta,$$

and

$$\mathbf{Pr}\left[\sum_{i=1}^{N} E_i \leq \frac{N}{2} \middle| x > y\right] = \mathbf{Pr}\left[\sum_{i=1}^{N} E_i \leq N(1-p) + \frac{N}{2} - N(1-p) \middle| x > y\right]$$

$$\leq \mathbf{Pr}\left[\left|\sum_{i=1}^{N} E_i - N \cdot \mathbb{E}[E_i]\right| > \frac{N}{2} - Np \middle| x > y\right]$$

$$\leq 2\exp\left(-\frac{\left(\frac{N}{2} - Np\right)^2}{N \cdot 1^2}\right)$$

$$= \delta.$$

Hence HIGHPROBGREATERTHAN$(x, y, \delta)$ is correct with probability at least $1 - \delta$.

For the communication cost, since GREATERTHAN uses $\mathcal{O}\left(\log \log n\right)$ bits of communication, provided that $x, y \in [-\operatorname{poly}(n), \operatorname{poly}(n)]$, and we run GREATERTHAN for $N = \mathcal{O}\left(\log \frac{1}{\delta}\right)$ times, then the total communication cost is $\mathcal{O}\left(\log \log n \log \frac{1}{\delta}\right)$ bits in total. $\qquad\square$

EFFICIENTCOMMUNICATION$(X, y, \varepsilon, \delta)$ in Algorithm 15 will send $\widetilde{y}$, a good approximation location of $y$ with probability at least $1 - \delta$. In addition, it only uses $d \log l \operatorname{polylog}(\log n, \log l, \frac{1}{\varepsilon}, \frac{1}{\delta})$ bits. To formally prove these guarantees, we first show $\|y - \widetilde{y}\|_2 \leq \varepsilon \|x - y\|_2$ with probability $1 - \delta$ and then upper bound the total communication of the protocol.

**Lemma E.2.** EFFICIENTCOMMUNICATION$(X, y, \varepsilon, \delta)$ *will send* $\widetilde{y}$ *to the first site such that* $\|y - \widetilde{y}\|_2 \leq \varepsilon \|x - y\|_2$ *with probability at least* $1 - \delta$.

*Proof.* We condition on the correctness of HIGHPROBGREATERTHAN. We will prove that $\|y - \widetilde{y}\|_2 \leq \varepsilon \|x - y\|_2$. First, we will prove $|y^{(i)} - \widetilde{y}^{(i)}| \leq \varepsilon |x_j^{(i)} - y^{(i)}|$ for any $j \in [d]$ and $x_j \in X$.

Let $X^{(i)} = \{-\Delta, x_1^{(i)}, x_2^{(i)}, \cdots, x_l^{(i)}, \Delta\}$. Assume $x_s^{(i)} \in X^{(i)}$ that $|x_s^{(i)} - y^{(i)}| \leq |x_j^{(i)} - y^{(i)}|$ for any $x_j^{(i)} \in X^{(i)}$. If $y^{(i)} = x_s^{(i)}$, $\widetilde{y}^{(i)}$ is just $y^{(i)}$, which means $|y^{(i)} - \widetilde{y}^{(i)}| = |x_s^{(i)} - y^{(i)}| = 0$.

If $y^{(i)} \neq x_s^{(i)}$, we have $\widetilde{y}^{(i)} = x_s^{(i)} + \Delta y^{(i)}$ and $\Delta y^{(i)} = \gamma \cdot (1 + \varepsilon)^m$, where $\gamma = \operatorname{sign}(y^{(i)} - x_s^{(i)})$. Assume $y^{(i)} - x_s^{(i)} = \gamma \cdot (1 + \varepsilon)^{m+m'}$. Then

$$|y^{(i)} - \widetilde{y}^{(i)}| = |(y^{(i)} - x_s^{(i)}) - (\widetilde{y}^{(i)} - x_s^{(i)})| = |(1 + \varepsilon)^{m+m'} - (1 + \varepsilon)^m|.$$

Since $|x_s^{(i)} + \gamma \cdot (1+\varepsilon)^m - y^{(i)}| \leq |x_s^{(i)} + \gamma \cdot (1+\varepsilon)^t - y^{(i)}|$ for any $t \in \mathbb{N}$, $|m'|$ must be less than 1. Hence

$$|y^{(i)} - \widetilde{y}^{(i)}| = (1+\varepsilon)^{m+m'} \cdot |1 - (1+\varepsilon)^{-m'}| \leq \varepsilon(1+\varepsilon)^{m+m'} = \varepsilon|x_s^{(i)} - y^{(i)}|.$$

Since $|x_s^{(i)} - y^{(i)}| \leq |x_j^{(i)} - y^{(i)}|$ for any $j$, and $|y^{(i)} - \widetilde{y}^{(i)}| \leq \varepsilon|x_s^{(i)} - y^{(i)}|$, thus $|y^{(i)} - \widetilde{y}^{(i)}| \leq \varepsilon|x_j^{(i)} - y^{(i)}|$. Therefore, for any $x_j \in X$,

$$\|y - \widetilde{y}\|_2^2 = \sum_{i=1}^{d} |y^{(i)} - \widetilde{y}^{(i)}|^2 \leq \sum_{i=1}^{d} \varepsilon^2 |x_j^{(i)} - y^{(i)}|^2 = \varepsilon^2 \cdot \|x_j - y\|_2.$$

Hence $\|y - \widetilde{y}\|_2 \leq \varepsilon\|x - y\|_2$ for any $x \in X$.

**Analysis of the failure probability.** It remains to upper bound the failure probability by counting how many times HIGHPROBGREATERTHAN$(\cdot, \cdot, \delta')$ is run in the algorithm. For every dimension $i$, we run HIGHPROBGREATERTHAN$(\cdot, \cdot, \delta')$ to find the closest $x_{i_s}^{(i)}$ to $y^{(i)}$ and $m$ to approximate $y^{(i)} - x_{i_s}^{(i)}$. To find the closest $x_{i_s}^{(i)}$ to $y^{(i)}$, we can use binary search to find $x_{i_p}^{(i)} \leq y^{(i)} \leq x_{i_{p+1}}^{(i)}$, and then compare their midpoint $\frac{x_{i_p}^{(i)} + x_{i_{p+1}}^{(i)}}{2}$ with $y^{(i)}$ to determine which one is closer to $y^{(i)}$. Since there are $l + 2$ elements in $\{x_{i_0}^i, x_{i_1}^i, \cdots, x_{i_l}^i, x_{i_{l+1}}^i\}$, we need to run HIGHPROBGREATERTHAN$(\cdot, \cdot, \delta')$ for $\log(l+2) + 1$ times in this stage.

To find $m$ that $|x_{i_s}^{(i)} + \gamma \cdot (1+\varepsilon)^m - y^{(i)}| \leq |x_{i_s}^{(i)} + \gamma \cdot (1+\varepsilon)^t - y^{(i)}|$ for any $t \in \mathbb{N}$, we can use binary search to find $(1+\varepsilon)^q \leq |y^{(i)} - x_{i_s}^{(i)}| \leq (1+\varepsilon)^{q+1}$, and then compare their midpoint with $|y^{(i)} - x_{i_s}^{(i)}|$ to determine which one is closer to $|y^{(i)} - x_{i_s}^{(i)}|$. Since we already have $x_{i_p}^{(i)} \leq y^{(i)} \leq x_{i_{p+1}}^{(i)}$ and $|x_{i_j}^{(i)}| \leq \Delta$ for all $j \in \{0, 1, \cdots, l+1\}$, we only need to search $q$ among $\{0, 1, \cdots, \log_{1+\varepsilon} \Delta\}$. Since $\Delta = \text{poly}(n)$, we can use binary search to find $q$ in at most $\mathcal{O}\left(\log \frac{\log n}{\varepsilon}\right)$ rounds. Thus we need to run HIGHPROBGREATERTHAN$(\cdot, \cdot, \delta')$ for $\mathcal{O}\left(\log\log n + \log \frac{1}{\varepsilon}\right)$ times in this stage. Therefore, we will apply HIGHPROBGREATERTHAN for at most $\mathcal{O}\left(d\left(\log l + \log\log n + \log \frac{1}{\varepsilon}\right)\right)$ times. Since HIGHPROBGREATERTHAN$(\cdot, \cdot, \delta')$ will return correct result with failure probability at most $\delta'$ and $\delta' = \mathcal{O}\left(\frac{\delta}{d\left(\log l + \log\log n + \log \frac{1}{\varepsilon}\right)}\right)$, EFFICIENTCOMMUNICATION$(X, y, \varepsilon, \delta)$ has a failure probability at most $\delta$. $\square$

Next, we analyze the communication complexity of our algorithm.

**Lemma E.3.** EFFICIENTCOMMUNICATION$(X, y, \varepsilon, \delta)$ *uses* $d \log \ell \, \text{polylog}(\log n, \log \ell, \frac{1}{\varepsilon}, \frac{1}{\delta})$ *bits of communication for total, where* $\ell = |X|$ *is the number of points owned by the first site.*

*Proof.* We need to run HIGHPROBGREATERTHAN$(\cdot, \cdot, \delta')$ for at most $\mathcal{O}\left(d\left(\log \ell + \log\log n + \log \frac{1}{\varepsilon}\right)\right)$ times. Since HIGHPROBGREATERTHAN$(\cdot, \cdot, \delta')$ cost $\mathcal{O}\left(\log\log n \log \frac{1}{\delta'}\right)$ bits for a single running, it takes $d \log \ell \, \text{polylog}(\log n, \log \ell, \frac{1}{\varepsilon}, \frac{1}{\delta})$ bits to run HIGHPROBGREATERTHAN$(\cdot, \cdot, \delta')$ for total.

We also need communication to send $\widetilde{y} = (\widetilde{y}^{(1)}, \widetilde{y}^{(2)}, \cdots, \widetilde{y}^{(d)})$ to the first site. However, we only need to send $x_{i_s}^{(i)}$ and $\Delta y^{(i)}$ to the first site. Since the first site already has the location of all $x \in X$, we only need to send $i_s$ to identify $x_{i_s}^{(i)}$, which takes $\mathcal{O}(\log \ell)$ bits. Since $\Delta y^{(i)} = \gamma \cdot (1+\varepsilon)^m$, we only need to send $\gamma$ and $m$. Sending $\gamma$ requires $\mathcal{O}(1)$ bits since $\gamma \in \{-1, 0, 1\}$. Sending $m$ requires $\mathcal{O}\left(\log\log n + \log \frac{1}{\varepsilon}\right)$ bits since $(1+\varepsilon)^m = \text{poly}(n)$. Therefore, it takes at most $d \log \ell \, \text{polylog}(\log n, \frac{1}{\varepsilon})$ bits to send $\widetilde{y} = (\widetilde{y}^{(1)}, \widetilde{y}^{(2)}, \cdots, \widetilde{y}^{(d)})$.

Hence Algorithm 15 takes at most $d \log \ell \, \text{polylog}(\log n, \log \ell, \frac{1}{\varepsilon}, \frac{1}{\delta})$ to send $\widetilde{y}$ to the first site. $\square$

Putting together Lemma E.2 and Lemma E.3, we have:

**Lemma E.4.** EFFICIENTCOMMUNICATION$(X, y, \varepsilon, \delta)$ *will send* $\widetilde{y}$ *to the first site such that* $\|y - \widetilde{y}\|_2 \leq \varepsilon \|x - y\|_2$ *with probability at least* $1 - \delta$. *Furthermore, it only uses* $d \log \ell \operatorname{polylog}(\log n, \log \ell, \frac{1}{\varepsilon}, \frac{1}{\delta})$ *bits of communication for total, where* $\ell = |X|$ *is the number of points owned by the first site.*

## E.2 $(1 + \varepsilon)$-CORESET VIA SENSITIVITY SAMPLING IN THE COORDINATOR MODEL

Given the analysis in Appendix E.1, it follows that we can effectively perform adaptive sampling to achieve a bicriteria approximation at the coordinator. It remains to produce a $(1 + \varepsilon)$-coreset, for which we again use sensitivity sampling.

EFFICIENTCOMMUNICATION$(X, y, \varepsilon, \delta)$ can send the location of $y$ using low communication cost. However, its communication cost is $d \log \ell \operatorname{polylog}(\log n, \log \ell, \frac{1}{\varepsilon}, \frac{1}{\delta})$, where $\ell = |X|$. For the coordinator model, suppose that each site $i \in [s]$ has a dataset $X_i$. Since $|X_i| = \mathcal{O}(n)$, and the coordinator needs to send the approximate location of all $\mathcal{O}(k)$ samples to each site to apply adaptive sampling, which would still require $dsk \log n \operatorname{polylog}(\log n, \frac{1}{\varepsilon}, \frac{1}{\delta})$ bits to send the location of samples using EFFICIENTCOMMUNICATION$(X_i, y, \varepsilon, \delta)$. Fortunately, we can generate a $(1 + \frac{\varepsilon}{2})$-coreset $P_i$ for every $X_i$, which has a size of $\tilde{\mathcal{O}}\left(\frac{k}{\min\{\varepsilon^4, \varepsilon^{2+z}\}}\right)$. Since adaptive sampling also works for the weighted case, it is enough to generate an $(\mathcal{O}(1), \mathcal{O}(1))$-bicriteria approximation for the weighted coreset $P = \cup_{i \in [s]} P_i$. Therefore, only $dsk \operatorname{polylog}(\log n, k, \frac{1}{\varepsilon}, \frac{1}{\delta})$ bits are necessary to send the location of the samples to each site.

To further eliminate the multiple dependency of $d$, we notice that only the distance between the data point $x \in P_i$ and the center generated by adaptive sampling $s \in S$ is necessary to apply adaptive sampling and sensitivity sampling. Hence, we can use the Johnson-Lindenstrauss transformation to map $P$ to $\pi(P) \subset \mathbb{R}^{d'}$, where $d' = \mathcal{O}(\log(sk))$. As a result of the JL transformation, $\pi(P)$ is located in a lower-dimensional space, but the pairwise distance is still preserved by the mapping. Therefore, we can further reduce the communication cost needed to send the location of the samples to each site, which is now $sk \operatorname{polylog}(\log n, s, k, \frac{1}{\varepsilon}, \frac{1}{\delta})$ bits. Hence, we can apply adaptive sampling and send the exact location of the sampled point $s$ to the coordinator. The coordinator can then use EFFICIENTCOMMUNICATION$(\pi(P_i), \pi(s), \varepsilon, \delta)$ to send the approximate location of $\pi(s)$ to every site, which is accurate enough for every site to update a constant approximation for the cost of points. Thus, we can repeat adaptive sampling using a low cost of communication and get an $(\mathcal{O}(1), \mathcal{O}(1))$-bicriteria approximation $S$ for the optimal $(k, z)$-clustering.

Since every site has an approximate copy of $\pi(S)$, they can send a constant approximation of $|C_j \cap P_i|$ and $\operatorname{cost}(C_j \cap P_i, S)$ to the coordinator. However, the site may assign a point to a center in $S$ that is not nearest to it. This is because the site only has an approximate location of $\pi(S)$, and therefore can assign $x$ to another center $s'$ if $\operatorname{cost}(x, s')$ is very close to $\operatorname{cost}(x, s)$, where $s$ is the center closest to $x$. Fortunately, the proof of Theorem B.7 does not require that all points $x$ be assigned to its closet point. Theorem B.7 is still valid if $\operatorname{cost}(x, s') \leq \mathcal{O}(1) \cdot \operatorname{cost}(x, s)$. Hence, we can generate a $(1 + \frac{\varepsilon}{4})$-coreset for $P$ by sensitivity sampling.

After applying sensitivity sampling to sample the points, each site can send the sampled points to the coordinator using EFFICIENTCOMMUNICATION$(S, x, \varepsilon', \delta)$. By a similar discussion of efficient encoding in Lemma B.13, we can prove that the sampled points form a $(1 + \frac{\varepsilon}{2})$-coreset $A'$ for $P = \cup_{i \in [s]} P_i$. Therefore, $A'$ would be a $(1 + \varepsilon)$-coreset for $X = \cup_{i \in [s]} X_i$, and the coordinator can solve the $(k, z)$-clustering based on the coreset $A'$. Since $|P_i| = \tilde{\mathcal{O}}\left(\frac{k}{\min(\varepsilon^4, \varepsilon^{2+z})}\right)$, we can run EFFICIENTCOMMUNICATION$(P_i, y, \varepsilon, \delta)$ using $\operatorname{polylog}(\log n, k, \frac{1}{\varepsilon}, \frac{1}{\delta})$ bits of communication to send each sampled point. We give the algorithm in full in Algorithm 16.

We now show that Algorithm 16 will return a $(1 + \varepsilon)$-coreset of $X$ with constant probability and uses low communication cost. We will first show that $A'$ is a $(1 + \varepsilon)$-coreset of $X$.

**Lemma E.5.** *Algorithm 16 returns a* $(1 + \varepsilon)$ *coreset of* $X$ *with probability at least* 0.96 *in the coordinator model.*

---

**Algorithm 16** $(1 + \varepsilon)$-coreset for the coordinator model

---

**Input:** The dataset $X_i$ every site $i$ owns, $i \in [s]$
**Output:** A $(1 + \varepsilon)$-coreset $A'$ sent to the coordinator
1: Every site $i$ generates a $(1 + \frac{\varepsilon}{2})$-coreset $P_i$ of $X_i$
2: The coordinator send random seed to every site that generate Johnson-Lindenstrauss $\pi$, such that $\|\pi(x) - \pi(y)\|_2 \in [\frac{1}{2}\|x - y\|_2, \frac{3}{2}\|x - y\|_2]$ for any $x, y \in P = \cup_{i \in [s]} P_i$ and $\pi(x) \in R^{d'}$ where $d' = \mathcal{O}(\log(sk))$
3: $S \leftarrow \{s_0\}$, where $s_0$ is a point sampled by the coordinator
4: $S_j \leftarrow \emptyset$ for $j \in [s]$
5: $N \leftarrow \mathcal{O}(k), \varepsilon' \leftarrow \mathcal{O}(\varepsilon^z), m \leftarrow \tilde{\mathcal{O}}\left(\frac{k}{\min(\varepsilon^4, \varepsilon^{2+z})}\right), \delta \leftarrow \mathcal{O}\left(\frac{1}{sk+m}\right)$
6: **for** $i \leftarrow 1$ to $N$ **do**
7:    Use EFFICIENTCOMMUNICATION$(\pi(P_j), \pi(s_{i-1}), \varepsilon', \delta)$ to send $\tilde{s}_{i-1}^{(j)}$, an approximation of $\pi(s_{i-1})$ to every site $j$. $S_j \leftarrow S_j \cup \{\tilde{s}_{i-1}^{(j)}\}$
8:    Every site updates the cost $\mathrm{cost}(\pi(P_j), S_j)$ and sends $\widetilde{D}_j$, a constant approximation of $\mathrm{cost}(\pi(P_j), S_j)$ to the coordinator
9:    $s_i \leftarrow$ LAZYSAMPLING$(\{\mathrm{cost}(\pi(x), S_j)\}, \{\widetilde{D}_j\})$
10: **end for**
11: Every site $j$ computes $|C_l^{(j)}|$ and $\mathrm{cost}(\pi(C_l^{(j)}), S_j)$ for $l \le |S|$, where $C_l^{(j)} = \{x \in P_j : \mathrm{dist}(\pi(x), \tilde{s}_l^{(j)}) \le \mathrm{dist}(\pi(x), \tilde{s}_p^{(j)}), \forall p \le |S|\}$
12: Every site $j$ sends a constant approximation of $|C_l^{(j)}|$ and $\mathrm{cost}\left(\pi\left(C_l^{(j)}\right), S_j\right)$ to the coordinator
13: The coordinator computes $\sum_{l \le |S|} |C_l^{(j)}|$ and $\sum_{l \le |S|} \mathrm{cost}\left(\pi\left(C_l^{(j)}\right), S_j\right)$, and sends constant approximation of them to every site
14: Every site computes $\widetilde{\mu}(x)$ as an $\mathcal{O}(1)$-approximation of $\mu(x)$ locally for all $x \in P_j$, and send $\widehat{\mu}(P_i)$, a constant approximation of $\widetilde{\mu}(P_i) = \sum_{x \in P_i} \widetilde{\mu}(x)$ to the coordinator
15: The coordinator samples site $j$ with probability $\frac{\widehat{\mu}(P_j)}{\sum_{i=1}^s \widehat{\mu}(P_i)}$ independently for $m$ times. Let $m_j$ be the time site $j$ are sampled. Sends $m_j$ to site $j$
16: **for** $i \leftarrow 1$ to $s$ **do**
17:    $A_i \leftarrow \emptyset, A_i' \leftarrow \emptyset$
18:    **for** $j \in [m_i]$ **do**
19:       Sample $x$ with probability $p_x = \frac{\widetilde{\mu}(x)}{\widetilde{\mu}(P_i)}$
20:       **if** $x$ is sampled **then**
21:          $A_i \leftarrow A_i \cup \{x, \frac{1}{m\widetilde{\mu}(x)}\}, \widetilde{x} \leftarrow$ EFFICIENTCOMMUNICATION$(S, x, \varepsilon', \delta), A_i' \leftarrow A_i' \cup \{\widetilde{x}, \frac{1}{m\widehat{\mu}(x)}\}$, where $\widehat{\mu}(x)$ is a $(1 + \frac{\varepsilon}{2})$-approximation of $\widetilde{\mu}(x)$
22:       **end if**
23:    **end for**
24: **end for**
25: $A' \leftarrow \cup_{i=1}^s A_i'$
26: **return** $A'$

---

*Proof.* Since we use EFFICIENTCOMMUNICATION$(\pi(P_j), \pi(s_{i-1}), \varepsilon', \delta)$ to send $\tilde{s}_{i-1}^{(j)}$, by Lemma E.4, $\|\pi(s_{i-1}) - \tilde{s}_{i-1}^{(j)}\|_2 \le \varepsilon'\|\pi(x) - \pi(s_{i-1})\|_2$ for any $x \in P_j$. Hence $\|\pi(x) - \tilde{s}_{i-1}^{(j)}\|_2$ would be a $(1 + \varepsilon')$-approximation of $\|\pi(x) - \pi(s_{i-1})\|_2$.

Since $\pi$ is a Johnson-Lindenstrauss mapping, by Theorem B.9, $\|\pi(x) - \pi(y)\|_2 \in [\frac{1}{2}\|x-y\|_2, \frac{3}{2}\|x - y\|_2]$ with probability at least $1 - \frac{1}{\mathcal{O}(sk)}$. Therefore, $\|\pi(x) - \pi(y)\|_2 \in [\frac{1}{2}\|x - y\|_2, \frac{3}{2}\|x - y\|_2]$ holds for any $x, y \in P = \cup_{i \in [s]} P_i$ with probability at least 0.99. Since $\|\pi(x) - \pi(y)\|_2$ is a constant approximation of any $x, y \in P$, and $\|\pi(x) - \tilde{s}_{i-1}^{(j)}\|_2$ is a $(1 + \varepsilon')$-approximation of $\|\pi(x) - \pi(s_{i-1})\|_2$, thus $\|\pi(x) - \tilde{s}_{i-1}^{(j)}\|_2$ would be a 4-approximation of $\|x - s_{i-1}\|_2$.

By sending $\widetilde{D}_j$, an $\mathcal{O}(1)$-approximation of $\mathrm{cost}(\pi(P_j), S_j)$, the coordinator owns a $\mathcal{O}(1)$-approximation of $\mathrm{cost}(P_j, S)$. Thus, we can apply adaptive sampling, and by Theorem B.10, $S$ would be an $(\mathcal{O}(1), \mathcal{O}(1))$-bicriteria approximation of the optimal solution for $(k, z)$-clustering of $P = \cup_{j=1}^s P_j$ with probability at least $0.99$.

Since $C_l^{(j)} = \{x \in P_j : \mathrm{dist}(\pi(x), \widetilde{s}_l^{(j)}) \leq \mathrm{dist}(\pi(x), \widetilde{s}_p^{(j)}), \forall p \leq |S|\}$ and $\frac{1}{4} \cdot \mathrm{dist}(x, s_p) \leq \mathrm{dist}(\pi(x), \widetilde{s}_p^{(j)}) \leq 4 \cdot \mathrm{dist}(x, s_p)$, therefore, $\frac{1}{16} \cdot \mathrm{dist}(x, S) \leq \mathrm{dist}(\pi(x), \widetilde{s}_l^{(j)}) \leq 16 \cdot \mathrm{dist}(x, S)$ if $x \in C_l^{(j)}$. Hence, if we apply sensitivity sampling with

$$\mu(x) = \frac{1}{4} \cdot \left( \frac{1}{k \left| \bigcup_{j=1}^s C_l^{(j)} \right|} + \frac{\mathrm{cost}(\pi(x), S_j)}{k \sum_{l \leq |S|} D_l^{(j)}} + \frac{\mathrm{cost}(\pi(x), S_j)}{\sum_{l \leq |S|, j \in [s]} D_l^{(j)}} + \frac{\Delta_x}{\sum_{l \leq |S|, j \in [s]} D_l^{(j)}} \right),$$

where $D_l^{(j)} = \mathrm{cost}\left( \pi\left( C_l^{(j)} \right), S_j \right)$ and $\Delta_p = \frac{\sum_{l \leq |S|} D_l^{(j)}}{\left| \bigcup_{j=1}^s C_l^{(j)} \right|}$, it would return a $(1 + \frac{\varepsilon}{4})$-coreset for $P = \cup_{j=1}^s P_j$ with probability at least $0.99$.

Since we send $\widetilde{x}$ to the coordinator by EFFICIENTCOMMUNICATION$(S, x, \varepsilon', \delta)$, thus $\|\widetilde{x} - x\|_2 \leq \varepsilon'\|s - x\|_2$, where $s$ is the point in $S$ closest to $x$. Since $\|\widetilde{x} - x\|_2 \leq \varepsilon'\|s - x\|_2$, by the proof of Lemma B.13, $(1 - \frac{\varepsilon}{4}) \cdot \mathrm{cost}(C, A) \leq \mathrm{cost}(C, A') \leq (1 + \frac{\varepsilon}{4}) \cdot \mathrm{cost}(C, A)$ for any solution $|C| = k$.

Since $P$ is a $(1 + \frac{\varepsilon}{2})$-coreset of $X$, $A$ is a $(1 + \frac{\varepsilon}{4})$-coreset of $P$, and $A'$ is a $(1 + \frac{\varepsilon}{4})$-coreset of $A$, therefore, $A'$ is a $(1 + \varepsilon)$-coreset for $X$.

**Evaluation of the success probability.** To apply adaptive sampling, we need to apply EFFICIENTCOMMUNICATION$(\pi(P_j), \pi(s_{i-1}), \varepsilon', \delta)$ for every $P_j$ and $s_{i-1}$, which would be $\mathcal{O}(sk)$ times of running in total. Since we need to apply EFFICIENTCOMMUNICATION$(S, x, \varepsilon', \delta)$ for every point sampled by sensitivity sampling, it would be $m$ times of running in total. Since $\delta \leftarrow \mathcal{O}\left( \frac{1}{sk+m} \right)$, the probability that all instances of EFFICIENTCOMMUNICATION returns an approximate location accurate enough is at least $0.99$.

Since the Johnson-Lindenstrauss would preserve the pairwise distance up to a multiple constant factor with probability at least $0.99$, the probability that adaptive sampling returns an $(\mathcal{O}(1), \mathcal{O}(1))$-bicriteria approximation is at least $0.99$, and sensitivity sampling returns a coreset with probability at least $0.99$, therefore, the total probability that Algorithm 16 returns a coreset for $X$ is at least $0.96$. $\qquad\square$

**Lemma E.6.** *Algorithm 16 uses* $\tilde{\mathcal{O}}\left( sk + \frac{dk}{\min(\varepsilon^4, \varepsilon^{2+z})} + dk \log n \right)$ *bits of communication.*

*Proof.* Since we apply Johnson-Lindenstrauss to map $P$ to $\mathbb{R}^{d'}$ where $d' = \mathcal{O}(\log |P|) = \mathcal{O}(\log(sk))$, by Theorem B.9, the coordinator needs to send an $\mathcal{O}(\log(sk \log(sk)) \cdot \log d)$ bits random seed to each site so that every site can generate the map $\pi$. It uses $s \, \mathrm{polylog}(s, k, d)$ bits.

By Lemma E.4, since $|P_j| = \mathcal{O}(k)$ and $d' = \mathcal{O}(\log(sk))$, we need to use $\mathrm{polylog}(\log n, s, k, \frac{1}{\varepsilon'}, \frac{1}{\delta})$ bits to apply EFFICIENTCOMMUNICATION$(\pi(P_j), \pi(s_{i-1}), \varepsilon', \delta)$. Since $|S| = \mathcal{O}(k)$, we need to use $d \, \mathrm{polylog}(\log n, s, k, \frac{1}{\varepsilon'}, \frac{1}{\delta})$ bits to apply EFFICIENTCOMMUNICATION$(S, x, \varepsilon', \delta)$. Since we need to apply EFFICIENTCOMMUNICATION$(\pi(P_j), \pi(s_{i-1}), \varepsilon', \delta)$ for $\mathcal{O}(sk)$ times to send location of $\pi(s_{i-1})$ to each site, and apply EFFICIENTCOMMUNICATION$(S, x, \varepsilon', \delta)$ for $\mathcal{O}\left( \frac{k}{\min(\varepsilon^4, \varepsilon^{2+z})} \right)$ times to send location of points sampled by sensitivity sampling to the coordinator, then the total cost to apply EFFICIENTCOMMUNICATION, in bits, is

$$sk \, \mathrm{polylog}\left( \log n, s, k, \frac{1}{\varepsilon} \right) + \frac{dk}{\min(\varepsilon^4, \varepsilon^{2+z})} \, \mathrm{polylog}\left( \log n, s, k, \frac{1}{\varepsilon} \right).$$

We need to send the exact location of $s_{i-1}$ to the coordinator, which takes $\mathcal{O}(dk \log n)$ bits. Every site $j$ sends a constant approximation of $|C_l^{(j)}|$ and $\mathrm{cost}\left( \pi\left( C_l^{(j)} \right), S_j \right)$ to the coordinator, and the coordinator needs to send constant approximation of $\sum_{l \leq |S|} |C_l^{(j)}|$ and

$\sum_{l \leq |S|} \text{cost}\left(\pi\left(C_l^{(j)}\right), S_j\right)$ to every site, which costs $\mathcal{O}\left(sk \log \log n\right)$ bits for total. Every site needs to send $\widehat{\mu}(P_i)$ to the coordinator, which costs $\mathcal{O}\left(s \log \log n\right)$ bits. The coordinator needs to send $m_i$, the number of points to be sampled to every site, which costs $\mathcal{O}\left(s \log k\right)$ bits. We need $\mathcal{O}\left(\frac{1}{\varepsilon} \log \log n\right)$ bits to send the weight $\widehat{\mu}(x)$ to the coordinator.

Therefore, the total communication cost for Algorithm 16 is

$$\tilde{\mathcal{O}}\left(sk + \frac{dk}{\min(\varepsilon^4, \varepsilon^{2+z})} + dk \log n\right)$$

bits. $\qquad \square$

We now give our full guarantees of our algorithm for the coordinator model.

**Theorem E.7.** *There exists an algorithm that returns a* $(1+\varepsilon)$ *coreset of* $X$ *with probability at least* 0.96 *in the coordinator model and uses* $\tilde{\mathcal{O}}\left(sk + \frac{dk}{\min(\varepsilon^4, \varepsilon^{2+z})} + dk \log n\right)$ *bits of communication.*

*Proof.* The proof follows immediately from Lemma E.5 and Lemma E.6. $\qquad \square$

### E.3    CLUSTERING ON GENERAL TOPOLOGIES

We provide a formal definition of distributed clustering under general topologies. Let $V = \{v_1, \ldots, v_n\}$ be a set of $n$ nodes connected via an undirected graph $G = (V, E)$, where each edge $(v_i, v_j) \in E$ represents a direct communication link between sites $v_i$ and $v_j$. Each node $v_i$ holds a local dataset $X_i$, and the global dataset is $X = \bigcup_{i=1}^{n} X_i$. Communication is allowed only along the edges of $G$.

The objective is to compute a set of $k$ centers $\mathcal{C} = \{c_1, \ldots, c_k\}$ that minimizes the global clustering cost for a given norm parameter $z \geq 1$:

$$\text{cost}(X, \mathcal{C}) = \sum_{x \in X} d(x, \mathcal{C})^z,$$

where $d(x, \mathcal{C}) = \min_{c_j \in \mathcal{C}} d(x, c_j)$ is the distance from point $x$ to its closest center in $\mathcal{C}$. The goal is to compute an approximate set of centers $\hat{\mathcal{C}}$ while minimizing communication over the edges of $G$.

In Balcan et al. (2013), the clustering problem is studied under this general-topology setting. They assume $s$ sites are connected by an undirected graph $G = (V, E)$ with $m = |E|$ edges. Their algorithm constructs a $(1 + \varepsilon)$-coreset, with communication $\mathcal{O}\left(m\left(\frac{dk}{\varepsilon^4} + sk \log sk\right)\right)$ words for $k$-means and $\mathcal{O}\left(m\left(\frac{dk}{\varepsilon^2} + sk\right)\right)$ words for $k$-median. Since a word requires $\mathcal{O}\left(d \log n\right)$ bits in their model, the corresponding costs are $\mathcal{O}\left(mdk \log n \left(\frac{d}{\varepsilon^4} + s \log sk\right)\right)$ bits for $k$-means and $\mathcal{O}\left(mdk \log n \left(\frac{d}{\varepsilon^2} + s\right)\right)$ bits for $k$-median.

Our coordinator-model algorithm can be naturally extended to general topologies: each message to the coordinator traverses at most $m$ edges, multiplying total communication by $m$. The resulting communication cost, in bits, becomes

$$\tilde{\mathcal{O}}\left(m\left(sk + \frac{dk}{\min(\varepsilon^4, \varepsilon^{2+z})} + dk \log n\right)\right).$$

Compared to Balcan et al. (2013), our approach improves the dependency on $n$ by replacing a multiplicative $\log n$ factor with an additive term, while preserving the same approximation guarantees for general $(k, z)$-clustering.

## F    ADDITIONAL EMPIRICAL EVALUATIONS

In this section, we perform additional empirical evaluations on both synthetic and real-world datasets to further support our theoretical guarantees. Unlike the experiments in the blackboard model presented in Section 4, we now focus on assessing our algorithms in the *coordinator model*. As a baseline, we use the constant-factor approximation algorithm based on adaptive sampling, denoted

`AS`. The second applies a standard dimensionality reduction approach to each of the points, using shared randomness, which can be acquired from public randomness, corresponding with the more communication-efficient variant `AS-JL`. In particular, for a dataset with $d$ features, we generate a random matrix of size $d' \times d$, where each entry is drawn from the scaled normal distribution $\frac{1}{\sqrt{d'}} \cdot \mathcal{N}(0,1)$ and $d'$ is set to be a constant factor smaller than $d$, i.e., $d' = \frac{d}{2}$ or $d' = \frac{d}{4}$. Finally, we apply our compact encoding scheme to the sampled points, centers produced by adaptive sampling, representing our coordinator model algorithm, denoted by `EAS-JL`.

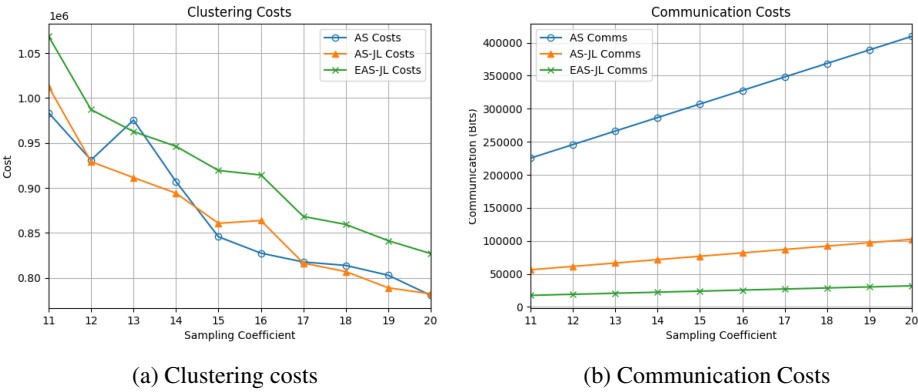

(a) Clustering costs            (b) Communication Costs

Fig. 7: Experiments for clustering costs and communication costs on DIGITS dataset

### F.1 REAL-WORLD DATASET

We first evaluated our algorithms on the DIGITS dataset (Alpaydin & Kaynak, 1998), as previously described in Section 4. As before, we use the parameters $n = 1797$, $d = 64$, and $k = 10$. We set $d' = \frac{d}{4} = 16$ for `AS-JL` and `EAS-JL`. We compare the clustering costs in Figure 5a and communication costs in Figure 5b for $c \in \{11, 12, \dots, 20\}$. Our results show that the clustering costs of the centers returned by our algorithm `EAS-JL` is consistently competitive with the centers returned by the other algorithms `AS` and `AS-JL`, while using significantly less communication across all settings of $c \in \{11, 12, \dots, 20\}$. For example Figure 7 shows that at $c = 11$, the clustering cost of `EAS-JL` is roughly 1.07 times the clustering cost of `AS` while using $9\times$ less communication. This trend seems to continue throughout the range of the sampling coefficient $c$, e.g., at $c = 20$, the clustering cost of `EAS-JL` remains roughly 1.07 times the clustering cost of `AS` while still using roughly $9\times$ less communication.

### F.2 SYNTHETIC DATASET

We next evaluated our algorithms on synthetic datasets consisting of Gaussian mixtures, as in Section 4. Specifically, we generated $k = 3$ clusters, each containing $640$ points for a total of $1920$ points, each with $8$ features, for a total dataset size of $15360$. Each cluster was drawn from a distinct Gaussian distribution whose mean was selected uniformly at random from the range $[-10, 10]^8$, and whose covariance matrix was generated as a random positive-definite matrix, producing clusters of varying orientations and shapes.

We compare clustering costs and communication costs across a sampling coefficient of $c \in \{1, 2, \dots, 10\}$, and find that `EAS-JL` consistently achieves clustering performance competitive with both `AS` and `AS-JL` while using substantially less communication. For instance, at $c = 5$, the clustering costs of `EAS-JL` and `AS` are almost equal, while the communication cost of `EAS-JL` is more than a factor of $4\times$ better. Similarly, at $c = 10$, the clustering costs of `EAS-JL` and `AS` are almost equal, while the communication cost of `EAS-JL` is a factor of almost $8\times$ better. These results parallel our findings on real-world data, demonstrating that our algorithm effectively preserves the clustering quality while significantly reducing communication in the coordinator model.

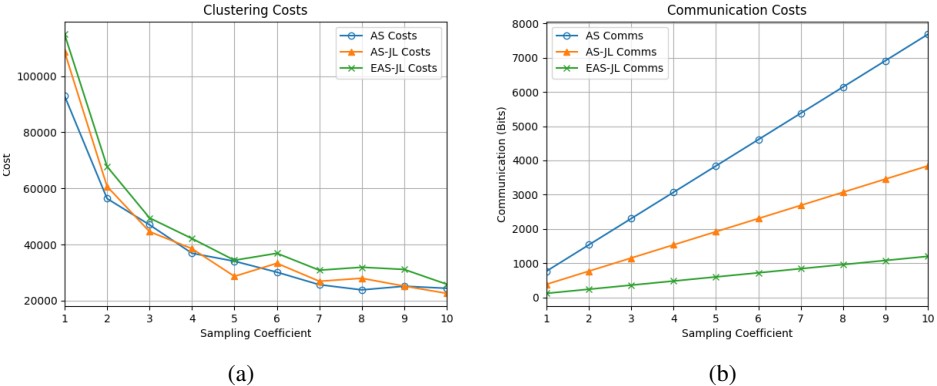

(a)                                                            (b)

Fig. 8: Experiments for clustering costs and communication costs on synthetic dataset

