# OpenReview forum: "Distributed Algorithms for Euclidean Clustering"
_ICLR.cc/2026/Conference — ICLR 2026 Poster_

### Official Review · Reviewer_WL6Z · 2025-10-26

**Soundness:** 3
**Presentation:** 3
**Contribution:** 4
**Rating:** 8
**Confidence:** 1

**Summary:**

The paper addresses the problem of efficiently constructing $(1+\varepsilon)$-coresets for Euclidean (k,z)-clustering in a distributed environment, focusing on two prominent and practically relevant communication models: the coordinator and the blackboard models. The authors propose novel distributed protocols that provide strong $(1+\varepsilon)$-approximation guarantees while simultaneously significantly reducing the overall communication complexity compared to prior best known results.

**Strengths:**

- The protocols presented almost match the lower bounds ie up to polylogarithmic for distributed $(1+\varepsilon)$ coreset construction for Euclidean (k,z)-clustering in both the coordinator and blackboard models. Further,

- The work introduces compact encoding schemes that enable coordinators/servers to exchange essential summary information rather than raw point coordinates.

**Weaknesses:**

- The protocols likely introduce implementation complexity relating to encoding/decoding, and requirements for synchronization among nodes.

**Questions:**

It will be helpful for readers if the authors can elaborate on the computational overhead incurred from the compact encoding and decoding schemes.

---

> ### Author Response · Authors · 2025-11-24
>
> > The protocols likely introduce implementation complexity relating to encoding/decoding, and requirements for synchronization among nodes.
>
> Thank you for the comment. While our protocols introduce some additional complexity compared to naive implementations, they are designed to be practical. The encoding and decoding schemes involve rounding values to powers of $(1+\varepsilon')$, which are standard numerical operations, and synchronization is carefully structured to minimize overhead. In the blackboard model, we reduce the number of rounds to $O(\log n \log k)$, and in the coordinator model, the protocol is again designed to minimize the number of communication rounds. We have also implemented these protocols in our empirical evaluations, demonstrating that they are indeed practical, at least as a proof-of-concept, and we believe that the substantial improvements in communication efficiency justify this moderate increase in implementation complexity.
>
> > It will be helpful for readers if the authors can elaborate on the computational overhead incurred from the compact encoding and decoding schemes.
>
> The computational overhead is relatively low.
>
> - **Encoding:** For a point $x$, encoding involves finding the nearest center $c$ in the bicriteria solution $S$ and then rounding each coordinate of the offset vector $x-c$ to the nearest power of $(1+\varepsilon')$. This takes $O(|S|d) = O(kd)$ time to find the nearest center and $O(d)$ time for the rounding.
> - **Decoding:** Decoding involves retrieving the center $c$ and reconstructing the rounded offset vector from the stored exponents, which takes $O(d)$ time.
>
> These operations are computationally efficient (linear in $d$ and $k$), and as noted above, our empirical evaluations implement these schemes, illustrating that the algorithms are practical in a proof-of-concept setting.

---

### Official Review · Reviewer_sPGF · 2025-10-28

**Soundness:** 1
**Presentation:** 2
**Contribution:** 3
**Rating:** 6
**Confidence:** 3

**Summary:**

The paper studies the construction of coresets for Euclidean $(k,z)$-clustering in distributed settings, where the $n$ input points are partitioned among $s$ sites. Two models are considered: the coordinator model, where all sites communicate only with a central coordinator over private channels using private randomness, and the blackboard model, where each site can write on a public blackboard visible to all, but still uses private randomness.

In the coordinator model, the authors design a $(1+\epsilon)$-coreset using roughly $O(sk + dk + dk \log(n\Delta))$ bits of communication (hiding dependencies on $z$ and $\epsilon$). In the blackboard model, they further reduce the communication complexity to $O(s \log(n\Delta) + dk \log(n\Delta))$ bits.

To achieve these results, the paper first presents a randomized bicriteria algorithm that uses only $O(s \log n + kd \log n)$ bits of communication. This is achieved via a lazy sampling procedure based on the adaptive sampling technique from prior literature. The key insight is to sample from a distribution close to the adaptive one, but requiring fewer communication bits. The resulting bicriteria solution is then used to construct the coresets through sensitivity sampling. Finally, the authors present experimental results supporting their theoretical claims. However, the paper lacks technical proofs, making it difficult to verify the correctness of the results (though they appear plausible).

**Strengths:**

1. The results are interesting and elegant — unlike prior approaches that required a communication term of $sdk \cdot \log(n\Delta)$ (due to each site transmitting point coordinates), the proposed approach eliminates this per-site dependence.

2. The techniques appear to involve clever ideas and nontrivial technical depth.

**Weaknesses:**

1. The main concern is presentation quality. The paper does not clearly explain the technical innovations. Moreover, the absence of proofs prevents the reader from understanding or validating the key arguments. This issue is aggravated by theorem statements (e.g., Lemmas 2.1 and 2.2) referring to algorithms that are not actually described in the main text. While the results seem promising, the lack of detail makes it impossible to assess the technical contributions on their merit.

2. Related to the above, several algorithmic components are mentioned (e.g., in Lemmas 2.1 and 2.2) without any accompanying description or pseudocode, making it hard to follow the logic or reconstruct the methodology.

**Questions:**

1. Could the authors elaborate on the main technical ideas that enable the reduction in communication complexity?

2. What is the size of the constructed coresets? Is there a particular reason why the coreset size is not explicitly stated in the technical results?

---

> ### Author Response · Authors · 2025-11-24
>
> > The main concern is presentation quality. The paper does not clearly explain the technical innovations. Moreover, the absence of proofs prevents the reader from understanding or validating the key arguments. This issue is aggravated by theorem statements (e.g., Lemmas 2.1 and 2.2) referring to algorithms that are not actually described in the main text. While the results seem promising, the lack of detail makes it impossible to assess the technical contributions on their merit.
>
> > The main concern is presentation quality. The paper does not clearly explain the technical innovations. Moreover, the absence of proofs prevents the reader from understanding or validating the key arguments. This issue is aggravated by theorem statements (e.g., Lemmas 2.1 and 2.2) referring to algorithms that are not actually described in the main text. While the results seem promising, the lack of detail makes it impossible to assess the technical contributions on their merit.
>
> We apologize for any confusion caused by the presentation in the original version. To address these concerns, we have made a substantial number of changes to improve clarity, exposition, and intuition throughout the paper. In particular:
>
> - We provide a full technical overview in Appendix A of the original submission and have now explicitly described the algorithmic components underlying Lemmas 2.1 and 2.2. This includes the subroutines `LazySampling`, `LOneSampling`, and `PowerApprox`, which are now explained in paragraph form with references to pseudocode (Alg. 4, Alg. 9, and Alg. 5 in Appendices B and D). These descriptions clarify how adaptive sampling, accuracy verification, and communication-efficient encoding work together in the blackboard and coordinator models.
>
> - We expanded the discussion of algorithmic and technical novelties at the end of Section 1 to highlight the key innovations, including:
>   - Why classical techniques like sensitivity sampling, JL transforms, and bicriteria approximations are insufficient in distributed settings.
>   - The lazy adaptive sampling protocol in the blackboard model and the use of $L_1$-sampling to reduce rounds and overall communication.
>   - Coordinate-wise sampling and distributed binary search in the coordinator model, which decouple communication cost from the dimension $d$.
>
> - We have clarified the size of the constructed $(1+\varepsilon)$-coresets (e.g., $\widetilde{O}(k/\varepsilon^4)$ for $k$-means and $k$-median) in the main technical results, which were previously only detailed in Appendix B.
>
> We hope that together, these changes make the technical contributions, algorithms, and proofs more accessible, and that the revised version clearly conveys both the algorithmic and technical novelties, along with the key subroutines behind our results.
>
> > Related to the above, several algorithmic components are mentioned (e.g., in Lemmas 2.1 and 2.2) without any accompanying description or pseudocode, making it hard to follow the logic or reconstruct the methodology.
>
> Thank you for the comment. To clarify, we provide explicit descriptions of the key algorithmic components underlying Lemmas 2.1 and 2.2 in the revised version. In particular, we describe the subroutine `LazySampling`, which efficiently implements adaptive sampling in the blackboard model using lazily updated approximate site weights, ensuring points are drawn close to the true distribution while drastically reducing communication. We also detail `LOneSampling`, which verifies that the aggregate approximate site costs remain within a constant factor of the true total, and `PowerApprox`, which encodes each site’s cost using only $O(\log\log n)$ bits. These subroutines are now described in paragraph form with accompanying pseudocode references (Alg. 4, Alg. 9, and Alg. 5) in Appendices B and D.

---

> > ### Author Response · Authors · 2025-11-24
> >
> > > Could the authors elaborate on the main technical ideas that enable the reduction in communication complexity?
> >
> > Thank you for the feedback. We provided a full technical overview in Appendix A of the original version and described the algorithmic and technical novelties at the end of Section 1. In the revised version, we have expanded the discussion at the end of Section 1 to clarify our contributions and highlight the key innovations, incorporating the following points:
> >
> > - Why standard techniques like sensitivity sampling, JL transforms, and bicriteria approximations are insufficient in distributed clustering.
> > - Our "lazy" adaptive sampling protocol in the blackboard model, which reduces communication by updating the blackboard only when local weights change significantly, combined with an $L_1$-sampling subroutine to further reduce rounds.
> > - Our coordinate-wise sampling and distributed binary search in the coordinator model, allowing transmission of only small offsets instead of full high-dimensional centers, decoupling communication from the dimension $d$.
> >
> > These methods show that classical techniques can be substantially optimized to efficiently support sensitivity sampling in distributed clustering. We believe our technical novelties, particularly efficient encoding which can be viewed as a form of quantization, also have potential applications in other distributed problems, such as regression and low-rank approximation, and thus may be of independent interest.
> >
> > > What is the size of the constructed coresets? Is there a particular reason why the coreset size is not explicitly stated in the technical results?
> >
> > Thanks for the question. Our algorithms utilize sensitivity sampling to output a $(1+\varepsilon)$-coreset of size $\widetilde{O}\Big(\frac{k}{\varepsilon^4}\Big)$ for Euclidean $k$-means and $k$-median and generalize to the optimal bounds for $(k,z)$-clustering as well. In the original submission, the coreset construction was described in Appendix B, but the size was not emphasized in the main text. In the revised version, we have made the coreset size explicit in the main technical results at the end of Section 1, clarifying the coreset size for both the blackboard model and the coordinator model.

---

### Official Review · Reviewer_DaWp · 2025-10-31

**Soundness:** 3
**Presentation:** 3
**Contribution:** 3
**Rating:** 6
**Confidence:** 4

**Summary:**

This paper addresses the problem of constructing coresets for Euclidean (k,z)-clustering in distributed settings, where n data points are partitioned across s sites. The authors study two communication models: the coordinator model and the blackboard model. For the coordinator model, they present a protocol that achieves a strong coreset with total communication complexity $\tilde{O}(sk+dk/\min${$\epsilon^4, \epsilon^{2+z}$}$+dklog(n\Delta))$ bits, which improves upon prior work (Chen et al., 2016) by avoiding directly to transmit all coordinates. In the blackboard model, they further achieve better bounds of communication cost than previous algorithms. The main contribution of this work is to provide the compact encoding schemes, and sampling operations, and efficient algorithm in message-passing.

**Strengths:**

1.The paper improves the communication complexity for distributed (k,z)-clustering coresets in both the coordinator and blackboard models as illustrated in Figure 1, eliminating the need to send raw coordinates.

2.The paper presents a solid theoretical contribution to the study of distributed coreset construction for Euclidean (k,z)-clustering.

3.The paper is well-written and well-structured, making the technical content accessible.

4.The combination of new coreset constructions, compact encoding strategies, and communication-efficient protocols results in practical algorithms that are easy to implement in real-world distributed systems.

**Weaknesses:**

1.While the paper proposes distributed algorithms for both the coordinator and blackboard models, the experimental evaluation only includes results for the blackboard model. No empirical comparison or validation is provided for the coordinator-based algorithm.

2.There is no evaluation of how the algorithm scales with the number of distributed machines, which is critical for understanding its practical applicability in real-world distributed systems.

3.The paper provides only communication complexity bounds, without reporting the actual running time and memory on the coordinator or client machines.

4.The paper lacks a clear and formal definition of clustering under general topologies, which is briefly mentioned but not elaborated upon.

**Questions:**

1.The authors should provide experimental comparisons with existing algorithms in the coordinator model to clearly demonstrate the advantages of their approach.

2.Could the authors include more details and comparative results on how the algorithm performs with varying numbers of machines, in order to assess its scalability in distributed settings?

3.In practical distributed applications, how should the parameters k and s be selected to balance clustering quality and communication efficiency in both the coordinator and blackboard models?

4.Regarding the blackboard model, is the per-machine computational complexity small enough to allow application on resource-constrained edge devices?

---

> ### Author Response · Authors · 2025-11-24
>
> > 1.While the paper proposes distributed algorithms for both the coordinator and blackboard models, the experimental evaluation only includes results for the blackboard model. No empirical comparison or validation is provided for the coordinator-based algorithm.
>
> Thank you for pointing this out. In response, we have included new experiments for the *coordinator model* on both real-world and synthetic datasets (see Appendix F in the revised version).
>
> On the DIGITS dataset, our coordinator-model algorithm `EAS-JL` delivers clustering costs that are nearly identical to those of `AS` and `AS-JL`, while achieving up to a $9\times$ reduction in communication across sampling coefficients $c \in \{11,\ldots,20\}$. Similarly, on synthetic Gaussian mixture datasets, `EAS-JL` maintains comparable clustering quality while lowering communication by a factor of up to $8\times$ over $c \in \{1,\ldots,10\}$.
>
> These new results provide empirical confirmation of the coordinator-model algorithm’s performance. We note, however, that the primary focus of our work remains theoretical, and the experiments are intended to complement and illustrate the theoretical guarantees rather than serve as the main contribution.
>
> > 2.There is no evaluation of how the algorithm scales with the number of distributed machines, which is critical for understanding its practical applicability in real-world distributed systems.
>
> From a theoretical standpoint, our bounds show a clear advantage as the number of sites $s$ increases. In the coordinator model, a naive approach would require $\widetilde{O}(skd) \cdot \mathrm{poly}(1/\varepsilon)$ communication, while our bound improves this to $\widetilde{O}(sk + dk) \cdot \mathrm{poly}(1/\varepsilon)$. This effectively separates the dependence on $s$ from the dimension $d$ and the approximation factor $\varepsilon$, resulting in substantially better scaling as the system grows—particularly in high-dimensional settings.
>
> > 3.The paper provides only communication complexity bounds, without reporting the actual running time and memory on the coordinator or client machines.
>
> We thank the reviewer for pointing this out. The running time and memory usage on the coordinator or client machines are indeed important in practice, but they reflect a different set of objectives that can be optimized with additional engineering. Our focus in this paper is on communication complexity, which is the primary bottleneck in distributed clustering settings. While implementation efficiency is important, it is orthogonal to the theoretical contributions we aim to highlight.
>
> > 4.The paper lacks a clear and formal definition of clustering under general topologies, which is briefly mentioned but not elaborated upon.
>
> Thank you for the comment. Formally, let $V = \{v_1, \ldots, v_n\}$ be nodes connected via an undirected graph $G=(V,E)$, where each edge allows communication between its endpoints. Each node $v_i$ holds a local dataset $X_i$, and the global dataset is $X = \bigcup_i X_i$. The goal is to compute $k$ centers $\mathcal{C} = \{c_1, \ldots, c_k\}$ minimizing the global $(k,z)$-clustering cost $\sum_{x \in X} d(x, \mathcal{C})^z$, where $d(x,\mathcal{C}) = \min_{c_j \in \mathcal{C}} d(x, c_j)$, while respecting the communication constraints of $G$.
>
> This makes explicit the communication restrictions and the general $(k,z)$-clustering objective. In [BEL13], a similar general-topology setting is studied, where $s$ sites connected by $G$ construct $(1+\varepsilon)$-coresets with communication proportional to the number of edges and dataset size.
>
> In the original version, we included a brief definition of distributed clustering under general topologies in Appendix E. In the revised version, we have expanded it to include the above formal definition.

---

> ### Author Response · Authors · 2025-11-24
>
> > 1.The authors should provide experimental comparisons with existing algorithms in the coordinator model to clearly demonstrate the advantages of their approach.
>
> We have added new experiments in Appendix F for the *coordinator model* on both real-world and synthetic datasets. On the DIGITS dataset, our coordinator-model algorithm `EAS-JL` achieves clustering costs nearly identical to `AS` and `AS-JL`, while reducing communication by up to $9\times$ across sampling coefficients $c \in \{11,\ldots,20\}$. Similarly, on synthetic Gaussian mixtures, `EAS-JL` maintains comparable clustering quality while lowering communication by a factor of $4\times$-$8\times$ for $c \in \{1,\ldots,10\}$. These results empirically confirm the advantages of our approach in the coordinator model, complementing our theoretical guarantees.
>
> > 2.Could the authors include more details and comparative results on how the algorithm performs with varying numbers of machines, in order to assess its scalability in distributed settings?
>
> From a theoretical perspective, our bounds clearly show improved scaling with the number of sites $s$. In the coordinator model, a naive approach would require $\widetilde{O}(skd)\cdot \mathrm{poly}(1/\varepsilon)$ communication, whereas our algorithm reduces this to $\widetilde{O}(sk + dk)\cdot \mathrm{poly}(1/\varepsilon)$, effectively separating the dependence on $s$ from the dimension $d$ and approximation factor $\varepsilon$. This ensures that communication scales efficiently as the number of machines increases, particularly in high-dimensional settings.
>
> > 3.In practical distributed applications, how should the parameters k and s be selected to balance clustering quality and communication efficiency in both the coordinator and blackboard models?
>
> The choice of $k$ typically depends on the application and the desired granularity of clusters. The number of sites $s$ is usually dictated by the infrastructure and natural partitioning of the data. Our theoretical bounds make explicit the trade-off between clustering quality and communication efficiency for given $k$ and $s$. In practice, one can tune $k$ to achieve the required clustering objective, while $s$ is fixed by the distributed setup; our algorithms remain communication-efficient across these settings in both coordinator and blackboard models.
>
> > 4. Regarding the blackboard model, is the per-machine computational complexity small enough to allow application on resource-constrained edge devices?
>
> Yes, the per-machine computation in the blackboard model involves computing distances to sampled centers and performing local aggregation, scaling with the local dataset size and the number of centers $k$. This makes it lightweight and easily handled by typical edge devices with standard CPUs, such as embedded computers or smartphones. However, the model assumes some local computation is possible, so extremely resource-constrained devices like simple sensors with minimal processing capability may not be suitable targets.

---

> ### Comment · Reviewer_DaWp · 2025-11-26
> **Response to the Authors**
>
> I thank the authors for their responses. My initial evaluation remains unchanged at this stage.

---

### Official Review · Reviewer_w3Z4 · 2025-11-05

**Soundness:** 3
**Presentation:** 2
**Contribution:** 2
**Rating:** 8
**Confidence:** 3

**Summary:**

The paper constructs coresets for the $(k,z)$ clustering problem in distributed setting for coordinator model where a coordinator facilitates the communication between nodes in rounds and also for the blackboard model where communication is using a shared 'blackboard'. For both models, by using the coresets, the paper improves the existing communication complexity for the $(k,z)$ clustering problem. The paper also demonstrates the effectiveness of the coresets with empirical evaluations on real and synthetic data.

**Strengths:**

1) The problem is an interesting one and will be of interest to the community.
2) To the best I could check, the claims in the paper appear sound. The paper is overall written well.
3) The paper improves on existing communication complexity bounds for both the coordinator and blackboard models.
4) The way, the usual tools and tricks of coreset literature like JL transform, bicriteria approximation, sensitivity sampling etc, have been modified to fit the requirements of the distributed setting will be of interest to the coreset community.

**Weaknesses:**

The two minor weaknesses that I see in the paper are:
1) The structure of the paper makes it a little hard to parse. Also, many ideas like JL transform, bicriteria approximation, sensitivity sampling which are well known are used in a clever way to get the results. However, that technical novelty and challenges are not sufficiently evident from the main body of the paper. The authors should try to highlight them. It may be a good idea to bring the discussion on why the usual techniques are not directly applicable from appendix to main body and also why coordinate wise sampling is required can be elaborated.

2) The experiments are of a proof-of-concept nature and not extensive. Detailed experiments and comparisons on some more datasets and different settings will strengthen the paper.

**Questions:**

See Weaknesses

---

> ### Author Response · Authors · 2025-11-24
>
> > The structure of the paper makes it a little hard to parse. Also, many ideas like JL transform, bicriteria approximation, sensitivity sampling which are well known are used in a clever way to get the results. However, that technical novelty and challenges are not sufficiently evident from the main body of the paper. The authors should try to highlight them. It may be a good idea to bring the discussion on why the usual techniques are not directly applicable from appendix to main body and also why coordinate wise sampling is required can be elaborated.
>
> Thanks for the suggestion. We introduce several novel techniques that circumvent standard communication bottlenecks in distributed Euclidean clustering. In the original version, we provided a technical overview in Appendix A and described the algorithmic and technical novelties at the end of Section 1. To clarify these contributions further, we expanded the discussion at the end of Section 1 in the revised version, incorporating the following details on our technical novelties.
>
> In the blackboard model, we first prove that the adaptive sampling process is robust to outdated data, thereby introducing a "lazy" adaptive sampling protocol, where sites only update the blackboard when their local weight estimates are significantly inaccurate. Our analysis proves that sampling from this outdated distribution suffices to guarantee a constant-factor approximation, so that all sites do not need to communicate in each round of adaptive sampling. To further improve the number of rounds (and thus the overall communication), we introduce a subroutine based on $L_1$ sampling to check whether the global weight has dropped significantly without querying all sites. Together, these techniques lead to a smaller number of sites transmitting information in each round, as well as a smaller number of overall rounds.
>
> Additionally, in the coordinator setting, we introduced a subroutine EfficientCommunication based on coordinate-wise sampling. Instead of broadcasting a high-dimensional center $y$, the coordinator and a site collaboratively perform a distributed binary search on the site's local sorted coordinates to find the closest match. This technique allows us to transmit only a small offset and efficiently decouple communication cost from the dimension $d$.
>
> Taken together, these components provide a unified message that well-known techniques can surprisingly be further optimized to efficiently support sensitivity sampling in distributed clustering. Moreover, our approach can be viewed as a form of quantization to reduce communication costs and thus we believe these techniques could be of independent interest, naturally lending itself to applications in other distributed problems such as regression, low-rank approximation, or general high-dimensional problems in computational geometry.
>
> > The experiments are of a proof-of-concept nature and not extensive. Detailed experiments and comparisons on some more datasets and different settings will strengthen the paper.
>
> Thank you for the suggestion. We have added new experiments in the *coordinator model* on both real-world and synthetic datasets (see Appendix F of the revised version).
>
> On the DIGITS dataset, our algorithm `EAS-JL` achieves clustering costs nearly identical to `AS` and `AS-JL` while reducing communication by up to $9\times$ across the range of sampling coefficients $c \in \{11,\ldots,20\}$. On synthetic Gaussian mixtures, `EAS-JL` similarly maintains clustering quality while achieving up to $8\times$ lower communication across $c \in \{1,\ldots,10\}$.
>
> We believe these results consistently demonstrate the effectiveness of our algorithm. Nevertheless, we emphasize that the main contribution of our paper is theoretical, and the experiments serve to complement and illustrate the theoretical guarantees rather than form the central focus.

---

### Author Response · Authors · 2025-11-21

We thank the reviewers for their insightful questions, thoughtful feedback, and positive outlook. We are currently finalizing additional experiments for our revised manuscript. Once these experiments are complete, we will provide full responses to all initial questions and upload the updated version of the document.

---

### Author Response · Authors · 2025-11-24

We thank the reviewers for their careful and thoughtful comments, as well as their positive feedback on our work, such as:
- The problem is an interesting one and will be of interest to the community.
- ...the claims in the paper appear sound. (Reviewer w3Z4)
- The paper is overall written well. (Reviewer w3Z4)
- The paper improves on existing communication complexity bounds for both the coordinator and blackboard models. (Reviewer w3Z4)
- The way, the usual tools and tricks of coreset literature like JL transform, bicriteria approximation, sensitivity sampling etc, have been modified to fit the requirements of the distributed setting will be of interest to the coreset community. (Reviewer w3Z4)
- The paper improves the communication complexity for distributed (k,z)-clustering coresets in both the coordinator and blackboard models as illustrated in Figure 1, eliminating the need to send raw coordinates. (Reviewer DaWp)
- The paper presents a solid theoretical contribution to the study of distributed coreset construction for Euclidean (k,z)-clustering. (Reviewer DaWp)
- The paper is well-written and well-structured, making the technical content accessible. (Reviewer DaWp)
- The combination of new coreset constructions, compact encoding strategies, and communication-efficient protocols results in practical algorithms that are easy to implement in real-world distributed systems. (Reviewer DaWp)
- The results are interesting and elegant — unlike prior approaches that required a communication term of $sdk\cdot\log(n\Delta)$ (due to each site transmitting point coordinates), the proposed approach eliminates this per-site dependence. (Reviewer sPGF)
- The techniques appear to involve clever ideas and nontrivial technical depth. (Reviewer sPGF)
- The protocols presented almost match the lower bounds ie up to polylogarithmic for distributed $(1+\varepsilon)$ coreset construction for Euclidean (k,z)-clustering in both the coordinator and blackboard models. (Reviewer WL6Z)
- The work introduces compact encoding schemes that enable coordinators/servers to exchange essential summary information rather than raw point coordinates. (Reviewer WL6Z)

In response to the initial revisions, we have made substantial revisions to improve presentation, clarity, and accessibility of the technical contributions, with the changes marked in blue in the revised manuscript.

In particular, we have:
- Expanded the discussion at the end of Section 1 to highlight the main technical innovations, including the limitations of classical techniques in distributed clustering, our lazy adaptive sampling protocol in the blackboard model with $L_1$-sampling for round reduction, and coordinate-wise sampling with distributed binary search in the coordinator model.
- Provided explicit descriptions of key algorithmic components underlying Lemmas 2.1 and 2.2, including the subroutines `LazySampling`, `LOneSampling`, and `PowerApprox`, described in paragraph form with references to pseudocode (Alg. 4, Alg. 9, and Alg. 5 in Appendices B and D).
- Clarified the size of the constructed $(1+\varepsilon)$-coresets in the main technical results, which were previously only detailed in Appendix B.
- Added new empirical evaluations in Appendix F for the coordinator model on both real-world and synthetic datasets to illustrate the practical feasibility of our algorithms and validate their communication efficiency, complementing the theoretical results.

We provide detailed responses to individual reviewer comments below.

---

### Meta-Review · Area_Chair_VRAe · 2026-01-05

**Summary:**

The paper studies the problem of constructing (1 + ε)-coresets for Euclidean (k, z)-clustering in the distributed settings (coordinator model and blackboard model). In particular, the proposed method significantly improves the previous communication costs, as shown in Fig 1.

All the reviewers are happy with the results, and agree that the proposed method is an important improvement upon distributed clustering problems. My only concern is that the topic, distributed clustering, slightly deviates from the core area of ICLR (which focuses on deep learning). Probably the algorithms and theory conferences, like SODA/ESA/SPAA, are more suitable.

**Reviewer Concerns:**

The reviewers raised some concerns on the presentation and experimental details. I think the authors have well addressed most of them.

**Reviewer Scores:**

The reviewers give 8, 6, 6, 8. I think the reviewers are happy with the responses from the authors, and likely to maintain their positive scores.

---

### Decision · Program_Chairs · 2026-01-26

Accept (Poster)